# Parameterization adaption needed to unlock the benefits of increased resolution for the ITCZ in ICON

Clarissa A. Kroll<sup>1</sup>, Robert C. Jnglin Wills<sup>1</sup>, Luis Kornblueh<sup>2</sup>, Ulrike Niemeier<sup>2</sup>, and Andrea Schneidereit<sup>3</sup>

Correspondence: Clarissa A. Kroll (clarissa.kroll@env.ethz.ch)

Abstract. The double Inter-Tropical Convergence Zone (double-ITCZ) bias is a persistent tropical precipitation bias over many climate model generations. This motivates investigations of whether increasing resolution and discarding parameterizations improves the representation of the large-scale atmospheric circulation and ITCZ. In this work, we study the double-ITCZ bias in an ICON XPP resolution hierarchy spanning from parameterized to explicitly described deep convection within a consistent framework. We demonstrate that the double-ITCZ persists across horizontal resolutions from 160 km to 5 km in specified seasurface temperature simulations, independent of deep-convective and non-orographic gravity wave parameterization. Changes in the treatment of near-surface wind speed within the turbulence parameterization can reduce the bias. However, we highlight that a key driver of the double-ITCZ bias in ICON seems to lie in the insufficient moisture transport from the subtropics to the inner tropics. The resulting low bias in tropical near-surface moisture reduces deep convection over the Warm Pool, leading to a weakened Walker circulation. These biases ultimately culminate in the double-ITCZ feature. Increasing the near-surface wind speed limiter improves tropical near-surface moisture but exacerbates the bias in the moisture source, increasing the inner tropical contribution at the expense of the subtropics. This degrades the representation of the global circulation, energy balance, and teleconnections. Additionally, we show that parameter adjustments at low resolution are informative of the response to the same parameter adjustments at high resolution. Our findings showcase the benefits of models supporting a range of resolutions and underline the importance of continuing the development of non-discardible parameterizations.

#### 1 Introduction

The hydrological cycle is an important manifestation of the global atmospheric circulation. Despite its importance, biases in the representation of precipitation within the Inter-Tropical Convergence Zone (ITCZ) have been a persistent challenge throughout many model generations in the Coupled Model Intercomparison Project (CMIP) (Tian and Dong, 2020). Among them, the double-ITCZ bias is the most prominent problem (Mechoso et al., 1995; Lin, 2007). It describes positive precipitation biases south of the equator in the eastern Pacific and Atlantic, as well as underestimated precipitation in the equatorial Pacific. While observations indicate that a double-ITCZ feature can occur naturally in these regions at certain times of the year, preferentially in March (Magnusdottir and Wang, 2008; Gonzalez et al., 2025), these features are much too strong and persistent in model

<sup>&</sup>lt;sup>1</sup>Institute for Atmospheric and Climate Science, ETH Zürich, Zürich, Switzerland

<sup>&</sup>lt;sup>2</sup>Max Planck Institute for Meteorology, Hamburg, Germany

<sup>&</sup>lt;sup>3</sup>Deutscher Wetterdienst, Offenbach am Main, Germany

simulations. In addition to its influence on regional precipitation biases, the double ITCZ can influence large-scale climate phenomena such as the El Niño-Southern Oscillation (Ham and Kug, 2014; Zhang et al., 2014). Therefore, identifying the cause of the double ITCZ is a key step towards improving climate models.

In order to understand the simulated double-ITCZ bias, it is helpful to consider the role of the ITCZ at the center of the large-scale atmospheric meridional circulation that imports moisture into the tropics and exports energy to higher latitudes (Holton, 2004; Schneider et al., 2014). Subtropical moisture is transported within the trade winds into the inner tropics, where its convergence sustains the ITCZ (Peixoto and Oort, 1992). The moist static energy content is greater in the poleward return flow in the upper troposphere, such that there is a net energy transport into the subtropics by the Hadley cells (Figure 1a). Since the position and structure of the ITCZ and the distribution of the net energy input to the atmosphere are interrelated (Bischoff and Schneider, 2014; Ren and Zhou, 2024), changes in the distribution of precipitation will be reflected in the atmospheric energy budget and vice versa. For example, the ITCZ resides in the Northern Hemisphere in the annual-mean due to the difference in the net energy input between the Northern and Southern Hemispheres (Philander et al., 1996; Frierson et al., 2013; Marshall et al., 2014). This underlines that the double-ITCZ problem cannot be investigated as an isolated tropical phenomenon: Suband extratropical biases in the energy budget can also be sources of the problem (Kang et al., 2008; Hwang and Frierson, 2013; Kang et al., 2019), and tropical biases can likewise cause biases in the sub- and extratropics (Henderson et al., 2017; Dong et al., 2022; Feng et al., 2023).

It is critical to recognize the coupling between the double-ITCZ problem and biases in the energy budget also in the context of model tuning, because the first step of model evaluation is in many cases the energy balance of the model (Wild, 2020). Most model evaluation workflows focus first on global mean top-of-the-atmosphere (TOA) fluxes and potentially surface energy fluxes (Mauritsen et al., 2012), and then proceed to tune additional atmospheric fields and processes, for example, as outlined in Hourdin et al. (2017). This is done with the knowledge that there can be trade-offs between these different optimizations, making dedicated approaches to optimize across multiple processes necessary, for example to focus on the precipitation distribution and TOA fluxes within one framework (Zhao et al., 2018a, b). However, in general, most published model evaluation workflows provide little information on regional energy budgets that are mechanistically important for the large-scale circulation and precipitation distribution. This poses a problem: Sometimes improvements in global mean energy fluxes introduce compensating errors in regional energy fluxes and lead to deterioration of the large-scale circulation and precipitation distribution.

Next to model tuning choices, an additional possible source of tropical precipitation biases is the incorrect representation of convective processes in coarse-scale models. Discarding potentially error-prone convective parameterizations (Arakawa, 2004; Jones and Randall, 2011; Sherwood et al., 2014; Hardiman et al., 2015) has therefore been one of the avenues explored to address the double-ITCZ problem (Song and Zhang, 2018; Zhou et al., 2022; Ma et al., 2023). In this context, it is important to note that even at a horizontal resolution of 5 km, the necessity of deep convective parameterizations is still disputed, with some

studies showing improvements in atmospheric representation with the elimination of parameterizations (Vergara-Temprado et al., 2020) and others showing deterioration and insufficiently resolved processes (Clark et al., 2024). Additionally, although it is sometimes stated that high-resolution simulations are less sensitive to parameterization choices than low-resolution simulations, a number of parameterizations still remain – such as the cloud microphysics, non-orographic gravity wave, turbulence, and in some cases shallow convective parameterizations – all of which are important for the model's atmospheric circulation and energy budget (Hu et al., 2021; Hájková and Šácha, 2023). High computational costs, in combination with the hope that the most relevant processes are now resolved, can however then lead to a shortening of the model tuning process in high-resolution simulations. A lack of systematic tuning could be one of the reasons why high-resolution convection-permitting simulations without parameterizations do not show improvements in skill for the tropical precipitation pattern compared to conventional CMIP models, while high-resolution simulations that still rely on parameterization can outperform both (Zhou et al., 2022; Schneider et al., 2024).

70

80

In this work, we investigate the precipitation distribution and the double-ITCZ bias in a resolution hierarchy of the ICON model spanning from parameterization-supported 160 km to deep-convection-permitting 5 km horizontal resolution. We discuss the influence of modeling parameter choices over the resolution hierarchy, using an otherwise identical setup, model code, and boundary conditions to allow for a robust comparison. This endeavor has only been recently enabled by the newly developed XPP configuration of the ICON climate model (Früh et al., 2022; Niemeier et al., 2023; Müller et al., 2024), which can be run efficiently employing GPUs at a wide range of horizontal resolutions and, in contrast to ICON sapphire (Hohenegger et al., 2022), supports the usage of convective parameterizations. We focus on the following questions:

- 1. **Resolution and parameterization dependence of the double-ITCZ bias**: Can increased horizontal resolution and switching off deep convective and gravity wave parameterization improve the double-ITCZ bias? Are there common biases across resolutions? Where can resolution-dependent improvements be found?

  (Addressed in Section 3.1 and 3.3)
- 2. **Resolution-(in)dependent bias corrections:** To the extent that there are common (double-ITCZ) biases, how can they be addressed and can the same adjustments be applied at various resolutions? (*Addressed in Section 3.2 and 3.3*)
- 3. **Underlying mechanisms:** What are the underlying mechanisms leading to the double-ITCZ bias in ICON and how do the chosen adjustments ameliorate it?

(Addressed in Section 3 and Section 4, summarized in Schematic Figure 1)

To address the second group of research questions, we focus on the choice of a minimum surface wind speed threshold  $(U_{min})$  in the vertical turbulence scheme of ICON XPP (see Section 2.1.2 for details). The minimum wind speed limiter  $U_{min}$  has been identified as a promising tuning candidate to address the double-ITCZ precipitation biases in low-wind regimes such as the Warm Pool (Segura et al., 2025). Additionally, it may also be resolution dependent, with generally increasing near-surface

wind speeds at increased horizontal resolution (Jeevanjee, 2017; Iles et al., 2020; Paccini et al., 2021; Morris et al., 2024), making it particularly interesting for a resolution sensitivity study.

In the following, we will show that precipitation biases are reduced with increases in resolution (Sec. 3.1) and physically motivated parameter adjustments (Sec. 3.2.1), both in terms of the large-scale precipitation field and a simplified ITCZ metric. Then we discuss the underlying physical mechanisms of this improvement, identifying potential trade-offs between the improved precipitation representation and other key quantities. We focus in particular on the impacts on the Hadley and Walker circulation (Sec. 3.2.2), global moisture sources (Sec. 3.2.3), moisture transport (Sec. 3.2.4) and the sensitivity of key global metrics to further tuning (Sec. 3.2.5). We also outline why our experiments hint that the actual origin of the precipitation biases does not reside in the tropics but in the subtropics (Sec. 3.2.6). For this purpose, we contrast the real world atmospheric representation, the one simulated by ICON expressing a double ITCZ and the one simulated by ICON with tuning adjustments. The identified large-scale differences in the three scenarios will be visualized building up on Fig. 1. In a last step, we test the transferability of the gained knowledge with respect to tuning choices to higher resolution (Sec. 3.3).

## 105 2 Methods

## 2.1 Model simulations

## 2.1.1 Model setup

We perform simulations with the atmosphere model of the Icosahedral Nonhydrostatic Weather and Climate Model for Numerical Weather Prediction, ICON-NWP (Zängl et al., 2015; Prill et al., 2023), in the eXtended Prediction and Projection, XPP-configuration - previously: Seamless configuration (Früh et al., 2022; Niemeier et al., 2023; Müller et al., 2025b). The setup of ICON XPP allows users to conduct weather and climate studies within one consistent framework (Müller et al., 2025a). ICON XPP uses parameterizations for radiation (Hogan and Bozzo, 2018), cloud microphysics (Seifert, 2008), vertical diffusion (Mauritsen et al., 2007), convection (Tiedtke, 1989; Bechtold et al., 2008), subgrid scale orographic drag (Lott and Miller, 1997) and non-orographic gravity wave drag (Orr et al., 2010). The atmosphere is coupled to the land model JSBACH (Reick et al., 2021).

The experiments cover a hierarchy of four horizontal resolutions: 160 km (R2B4), 80 km (R2B5), 40 km (R2B6), and 5 km (R2B9). This includes the horizontal resolution range of conventional CMIP-type models (160 km) up to the resolution of deep-convection-permitting models (5 km). For all resolutions, 150 vertical levels up to a model top of 75 km are employed with a constant layer thickness of 300 m from a height of 8.4 km to 19 km. The prescribed constant layer thickness ensures that the altitude range of deep convection is fully captured, even in the inner tropics (Schmidt et al., 2023). Two configurations are run at 5 km. The first configuration relies on the full suite of parameterizations. In the second configuration, the parameterization for subgrid non-orographic gravity waves, mid-level and deep convection are disabled. We will refer to this setup

Figure 1. Conceptual depiction of the main zonal and meridional circulations exchanging moisture and energy between the subtropics and the tropics as a) observed, b) in default ICON simulations, and c) in  $U_{min}$  adjusted ICON simulations. Observations show that the moisture within the innermost tropics originates from evaporation in tropical and subtropical areas. The moisture is imported into the innermost tropics by the trade winds, with small losses to the free troposphere. In the innermost tropics it is then transported to the main convective regions, i.e., the Warm Pool (WP) within the zonally anomalous Walker circulation. Energy is exported to the subtropics within the outward branches of the Hadley circulation. In conventional ICON simulations (b), a large portion of the subtropical moisture is lost due to too much vertical transport of moisture from the boundary layer to the free troposphere. This leads to a moisture deficit in the tropics, specifically over the WP, reducing deep convection, slowing down the Walker circulation and leading to spurious additional convective centers over the Eastern Pacific as too little moisture is transported to the WP within the boundary layer. In the  $U_{min}$  adjusted ICON simulations (c) increased evaporation leads to an additional moisture source in the tropics, which resolves the dry bias over the WP. However, the adjustments increase the positive free-tropospheric moisture bias between  $0^{\circ}$  to  $30^{\circ}$ , reduce the trade wind strength and change the balance between innermost-tropical and subtropical sources of tropical moisture content. Green colors indicate high moisture content. Brown colors low moisture content.

as the rParam simulation. In contrast to Hohenegger et al. (2022), the orographic gravity wave drag and shallow-convective

145

In all experiments, we use prescribed 6-hourly climatological sea ice and SST fields interpolated from the monthly climatological values of the CMIP6 Forcing Datasets (input4MIPs, 1978-2020) as boundary conditions (Durack and Taylor, 2018). Prescribed climatological SSTs reduce the impact of interannual variability, such as the influence of ENSO events on precipitation. In addition, they separate the effect of model biases in the SST representation and atmospheric processes. The simulations are initialized with data from the European Centre for Medium-Range Weather Forecasts ERA5 reanalysis (Hersbach et al., 2020) for January 1, 2004.

## 2.1.2 Experiments

## Default and $U_{min}$ adapted ICON

 $U_{min}$ , the near-surface wind speed limiter, has been suggested as a potential lever for precipitation biases over the Warm Pool region in the ICON sapphire model setup (Segura et al., 2025). In our study, we investigate potential consequences of using this limiter for the global circulation and climate in a resolution hierarchy of the ICON XPP setup.

The near-surface wind speed limiter  $U_{min}$  acts within the bulk flux formulation in the turbulence parameterization. In this framework, evaporation E is tied to wind speed U at the lowest atmospheric level via the bulk flux formula for evaporation

$$E = \rho_a C_E U(q_s - q_a), \tag{1}$$

where  $\rho_a$  is the atmospheric density,  $q_s$  surface specific humidity,  $q_a$  near-surface atmospheric specific humidity, and the bulk transfer coefficient for latent heat  $C_E$ , which is inversely proportional to the Richardson number. The bulk flux formulation shows that the latent heat flux is directly proportional to the near-surface wind speed. The fix suggested by Segura et al. (2025) does not change U directly, but adapts a hard-coded lower limit for the near-surface wind speeds. In ICON XPP's turbulence parameterization, the surface wind U used for the computation of the Richardson number has a lower prescribed threshold  $U_{min}$  with  $U = MAX(U_{min}, U)$ . This lower limiter is used to account for the influence of subgrid-scale turbulence with the goal of increasing the turbulent fluxes in low-wind regimes. Increasing the default value of  $U_{min}$  from  $1 \text{ m s}^{-1}$  decreases the Richardson number in low-wind regimes, e.g., in the Warm Pool, and leads to increased evaporation. The influence of changes in  $U_{min}$  at various resolutions, including in simulations with the full set of parameterizations, has not been tested before. It is important to note that changes in  $U_{min}$  can similarly impact turbulent fluxes of sensible heat and momentum in addition to the targeted latent heat flux. Potential consequences of the resulting changes in the momentum budget were not considered in Segura et al. (2025). Concretely, this means that the suggested increase in  $U_{min}$  will increase the drag on near-surface winds. It is our aim to investigate whether there are associated negative influences on the circulation next to the positive effect for precipitation over the Warm Pool.

## **Description of experiment configurations**

The simulations presented in this work are summarized in Table 1. We run five control simulations (CTL) in four different horizontal resolutions: 160 km, 80 km, 40 km and 5 km. The simulations are run for six years for all coarser resolutions (160 to 40 km). Due to the very high computation demand, the two 5 km resolution setups - with full parameterization set and reduced parameterization set - are run for three years each instead of six years.

Branching off at the fourth year of the 40-km simulations, two-year-long sensitivity (perturbation) studies are performed (hereafter labeled with PTB). In the perturbation experiments, we vary the minimum surface wind speed limiter,  $U_{min}$ , which is presented to the turbulence scheme (Table 1). The two  $U_{min}$  settings that perform the best in representing the large-scale annual-mean precipitation, PTB-5 and PTB-6, with  $U_{min} = 5 \text{ m s}^{-1}$  and 6 m s<sup>-1</sup>, are chosen for further optimization. First, the wind speed is adapted for land and ocean separately. In PTB-5\_1 and PTB-6\_1,  $U_{min}$  is set to = 1 m s<sup>-1</sup> over land to account for the slower near-surface wind speeds. Second, in PTB-5\_1t, PTB-6\_1t, the model is retuned to reestablish a similar top-of-the-atmosphere (TOA) imbalance to CTL.

Only the best performing  $U_{min}$  setting, PTB-5\_1 and the corresponding tuned settings PTB-5\_1t, are tested in 3-month branchoff runs of the CTL-5 km and CTL-5 km-rParam simulation. For this purpose, MAM is chosen because it is the season with the largest double-ITCZ biases. To allow the model time to equilibrate with the changed settings, a one-month spin-up is used.

As the investigated horizontal resolutions differ by up to a factor of 32, all data sets are remapped for a fair comparison in the subsequent analysis. For this purpose, a 1° horizontal resolution is chosen. This resolution is close to the tropical 1.4° horizontal resolution of the 160 km simulation.

## 2.2 Reanalysis and observational data sets

The model output is evaluated against reanalysis and observational data sets. For the radiative fluxes, data from the "Clouds and the Earth's Radiant Energy System" (CERES; Doelling et al. (2013, 2016)), an observational dataset for TOA energy fluxes, are used. The precipitation distribution is compared to the observational data sets "Integrated Multi-satellitE Retrievals for GPM" (IMERG; Huffman et al. (2010)) and "Global Precipitation Climatology Project" (GPCP; Huffman (2021)). In addition to these observational data sets, we show radiative fluxes from ERA5, (C3S, 2018; Simmons et al., 2020), noting that these values are not assimilated (Dee et al., 2011; Hersbach et al., 2017). This means that the reanalysis product might behave more like a conventional model for these variables and should only be used to complement the observational data. For latent heat flux, near-surface wind speeds, and near-surface specific humidity  $q_s$ , we refer to ERA5 as a primary reference, as it assimilates station-based humidity measurements (Simpson et al., 2024) and offers a homogeneous data set with values both over land and ocean. Whereas latent heat flux and near-surface wind speeds are direct outputs of ERA5, the values of  $q_s$  are computed (details in Appendix A). The ERA5 latent heat flux values are high biased (Martens et al., 2020; Song et al., 2021). Therefore, we also compare our data to the OAFlux dataset (Schneider et al., 2013; National Center for Atmospheric Research Staff, 2022), which integrates satellite retrievals and three atmospheric reanalysis.

**Table 1. Summary of Experiments**: Experiment acronyms, employed resolutions, simulated timeframe, setting for the minimum horizontal wind speed over ocean/land and tuning. The control simulation (CTL) run for all resolutions is marked in gray, the best-performing perturbation experiment (PTB) for the large-scale precipitation representation is marked in green. The runs with the reduced set of parameterizations are labeled as rParam. Resolution is given in terms of the ICON grid used and the corresponding horizontal grid spacing.

| Experiment      | Resolution                 | Simulated | Surface Wind Speed Limiter $U_{min}$ | Tuned        |
|-----------------|----------------------------|-----------|--------------------------------------|--------------|
|                 |                            | Timeframe | $(Ocean/Land) / m s^{-1}$            |              |
| PTB-0.5         | R2B6, 40 km                | 2 years   | 0.5 / 0.5                            |              |
| CTL             | R2B{4,5,6}, 160, 80, 40 km | 6 years   | 1.0 / 1.0                            | $\checkmark$ |
|                 | R2B9, 5 km                 | 3 years   | 1.0 / 1.0                            | $\checkmark$ |
| CTL-rParam      | R2B9, 5 km                 | 3 years   | 1.0 / 1.0                            | $\checkmark$ |
| PTB-1.5         | R2B6, 40 km                | 2 years   | 1.5 / 1.5                            |              |
| PTB-2           | R2B6, 40 km                | 2 years   | 2.0 / 2.0                            |              |
| PTB-4           | R2B6, 40 km                | 2 years   | 4.0 / 4.0                            |              |
| PTB-5           | R2B6, 40 km                | 2 years   | 5.0 / 5.0                            |              |
| PTB-5_1         | R2B6, 40 km                | 2 years   | 5.0 / 1.0                            |              |
| PTB-5_1         | R2B9, 5 km                 | 2 MAM     | 5.0 / 1.0                            |              |
| PTB-5_1-rParam  | R2B9, 5 km                 | 2 MAM     | 5.0 / 1.0                            |              |
| PTB-5_1t        | R2B6, 40 km                | 2 years   | 5.0 / 1.0                            | ✓            |
|                 | R2B9, 5 km                 | 2 MAM     | 5.0 / 1.0                            | $\checkmark$ |
| PTB-5_1t-rParam | R2B9, 5 km                 | 2 MAM     | 5.0 / 1.0                            | ✓            |
| PTB-6           | R2B6, 40 km                | 2 years   | 6.0 / 6.0                            |              |
| PTB-6_1         | R2B6, 40 km                | 2 years   | 6.0 / 1.0                            |              |
| PTB-6_1t        | R2B6, 40 km                | 2 years   | 6.0 / 1.0                            | $\checkmark$ |

When comparing model data to observational data or ERA5 statistical significance is tested with a two-sided z-test at  $\alpha = 0.1$ . For the humidity and temperature fields, we correct for autocorrelation via adjustment of the net sample size based on lag-1 autocorrelation values of years 2004-2010 from ERA5 and the 40 km simulations (cf. Trenberth (1984)).

## 195 2.3 ITCZ evaluation metrics

To evaluate the large-scale ITCZ features, we use an energetically motivated framework, where precipitation biases are separated into hemispherically symmetric (Adam et al., 2016) and antisymmetric components (Hwang and Frierson, 2013). The symmetric component

$$E_p = \frac{\overline{P}_{2^{\circ} S - 2^{\circ} N}}{\overline{P}_{20^{\circ} S - 20^{\circ} N}} - 1 \tag{2}$$

is related to the net energy input near the equator.  $E_p = 0$  would describe a state in which the precipitation average  $\overline{P}$  between  $2^{\circ}$  S and  $2^{\circ}$  N is equal to the precipitation average between  $20^{\circ}$  S and  $20^{\circ}$  N. The anti-symmetric component

$$A_{p} = \frac{\overline{P}_{0^{\circ} N - 20^{\circ} N} - \overline{P}_{0^{\circ} S - 20^{\circ} S}}{\overline{P}_{20^{\circ} S - 20^{\circ} N}},$$
(3)

is related to the cross-equatorial energy flux, which is influenced by asymmetries in the net energy input in the high latitudes of the two hemispheres. In both equations  $\overline{P}$  is the average precipitation in the region constrained by the two latitude boundaries referred to in the respective subscript.

### 2.4 Residence time in boundary layer based on moisture budget analysis

In order to assess biases in moisture transport, we evaluate the residence time of moisture within the boundary layer. Our analysis is based on a time-mean moisture budget (Peixóto and Oort, 1983; Peixoto and Oort, 1992) linking moisture flux convergence with precipitation and evaporation fluxes as

210 
$$\overline{P} - \overline{E} = -\frac{1}{g} \int_{0}^{p_s} \nabla \cdot \overline{\beta_{\rho} \mathbf{u} q} \, dp.$$
 (4)

In this equation  $\overline{(\cdot)}$  denotes the annual-mean climatology,  $\mathbf{u}$  the horizontal wind vector, q the specific humidity, p is the pressure and g the gravitational constant.  $\beta_{\rho}$  is a Heaviside function indicating whether values are above or below the surface (cf. Boer (1982)). Based on equation 4, we then estimate the residence time  $\tau_{R_{bl}}$  within the boundary layer by splitting the integral into a contribution from the boundary layer up to 850 hPa and the rest of atmosphere and rearranging:

215 
$$\frac{1}{\tau_{R_{bl}}} = \left| \frac{-\frac{1}{g} \int_{850 \ hPa}^{p_s} \nabla \cdot \overline{\beta_{\rho} \mathbf{u} q} \, dp}{\frac{1}{g} \int_{850 \ hPa}^{p_s} \overline{q} \, dp} \right| = \left| \frac{\overline{P} - \overline{E} + \frac{1}{g} \int_{0}^{850 \ hPa} \nabla \cdot \overline{\beta_{\rho} \mathbf{u} q} \, dp}{\frac{1}{g} \int_{850 \ hPa}^{p_s} \overline{q} \, dp} \right|.$$
 (5)

Dividing by the integrated moisture within the boundary layer helps control for the dependence of boundary-layer moisture flux on boundary-layer moisture.

#### 2.5 Circulation analysis

205

We calculate the velocity potential  $\chi$  at 200 hPa to assess the Walker and Hadley circulation strength following the approach by Tanaka et al. (2004) and Lu et al. (2007) (see Appendix B for derivation details).  $\chi_{max}$ , the maximum of the zonal-mean velocity potential at 200 hPa within the 30 °latitude band is used as a proxy for the Hadley circulation strength. As  $\chi$  can only be determined with a constant offset due to the inverse Laplacian, we choose this constant so that the global mean of  $\chi$  is zero for consistency between the individual experiments. The Walker circulation is the zonally anomalous part of the circulation. Here, we employ  $\chi_{max}^*$ , the maximum deviation of the annual mean  $\chi$  in the 10 °latitude band from its zonal mean, as a Walker circulation metric.

## 3 Results

## 3.1 Consistent large-scale precipitation biases across horizontal resolutions

To investigate the influence of horizontal resolution on the double-ITCZ bias, we focus on CTL experiments with horizontal grid resolutions of 5 to 160 km employing all parameterizations. The two-year mean precipitation shows qualitatively similar biases compared to IMERG in all four resolutions of ICON (Fig. 2). Quantitatively, there is a consistent improvement in the global root mean square error (RMSE) with respect to IMERG from 1.55 mm day<sup>-1</sup> (160 km) to 1.27 mm day<sup>-1</sup> (5 km). The large-scale patterns of precipitation bias persist with increasing resolution, but decrease in amplitude. Especially apparent in the tropics is the double-ITCZ feature over the Pacific and Atlantic, as well as the dry bias over the Maritime Continent. Areas of consistent improvement include coastal regions, for example, in the ITCZ on either side of South America, or surrounding the Maritime Continent.

The tropical dry bias over the Warm Pool is tied to a lack of near-surface specific humidity throughout the entire tropics (average global values in inlays of Fig. 2; spatial distribution in Appendix Fig. C1). The lack of near-surface specific humidity affects the build-up of larger deep convective systems in the inner tropics and with it the vertical moisture distribution (Fig. 3). Consistent for all resolutions, the regions between 20° S and 20° N are too dry below 1 km in altitude, although a clear improvement with increasing horizontal resolution is apparent. The lack of near-surface moisture helps to explain the lack of precipitation in convective areas close to the equator. In contrast to the inner tropics, the areas between 15° to 40° latitude exhibit a moist bias above 1 km in height, peaking at around 20° N and S at a height of 2 km. The vertical structure of humidity biases indicates too strong vertical moisture flux out of the boundary layer. The drying of the boundary layer reduces the amount of moisture transported into the inner tropics by the trade winds (Figure 1b). We will demonstrate this mechanism in more detail in Section 3.2.

The consequences of the biases in the specific humidity distribution are also apparent in the zonal-mean temperature biases (Fig. 4, cf. Fig. 3 and Appendix Fig. C1). Across the resolutions, a near-surface warm bias and a cold bias in the tropical free troposphere persist. The near-surface warm bias is most likely caused by the dry bias in the near-surface layer, which ties the atmosphere closer to a dry adiabatic lapse rate. This leads to a warming of the near-surface atmosphere as descending air masses heat up more easily. The free troposphere aloft cools: deep convective systems carry less moisture, this reduces latent heat release and ties the temperatures to the steeper dry adiabat. As temperatures in the tropics are quite homogeneous, as explained by the Weak Temperature Gradient theory (Charney, 1963; Held and Hoskins, 1985; Bretherton and Smolarkiewicz, 1989), this bias propagates through the tropical region. The warm air band at 20-40° N coincides with the region influenced by the local topography of the Himalayas, where the too warm surface air is advected upwards (Pan et al., 2016).

Figure 2. (a) Climatological IMERG precipitation distribution from 2004 to 2010 and (b)-(e) two-year mean precipitation bias with respect to IMERG for  $160 \,\mathrm{km}$  (b),  $80 \,\mathrm{km}$  (c),  $40 \,\mathrm{km}$  (d) and  $5 \,\mathrm{km}$  (e) setups. Statistically insignificant differences between IMERG and the experiments based on a two sided z-test at  $\alpha$ =0.1 are grayed out in (b)-(e). The corresponding global RMSE in precipitation is stated beneath the panel for each resolution experiment. The global RMSE and mean error in near-surface specific humidity, calculated with respect to values derived from ERA5 reanalysis, is depicted in the inlays.

**Figure 3.** (a) Climatological zonal-mean atmospheric specific humidity distribution from ERA5 for the years 2004 to 2010 and (b)-(e) two-year atmospheric specific humidity bias with respect to ERA5 for 160 km (b), 80 km (c), 40 km (d) and 5 km setups. Statistically significant differences between ERA5 and the model data are shown, and insignificant regions are dashed out.

Summarizing, we found that increased resolution alone does not resolve the double-ITCZ bias up to a resolution of 5 km. From the biases, we gain a first understanding of the underlying problem: It appears that the negative bias in near-surface specific

**Figure 4.** (a) Climatological zonal-mean temperature fields from ERA5 averaged over the timeframe from 2004 to 2010 and (b)-(e) two-year atmospheric temperature bias with respect to ERA5 for 160 km (b), 80 km (c), 40 km (d) and 5 km (e) setups. Statistically significant differences between ERA5 and the model data are shown, insignificant regions are dashed out.

humidity is one of the causes of the precipitation bias. A negative bias in near-surface moisture inhibits deep convection in the Warm Pool and weakens the Walker circulation. A too-weak Walker circulation reduces moisture transport from the East Pacific to the West Pacific and further weakens the Walker circulation. Furthermore, too-weak deep convection leads to a cold bias in the free troposphere, which reduces the temperature inversion also in distant regions such as the Eastern Pacific, potentially leading to an additional positive feedback on the circulation by reducing the cloud cover in the subsidence regions. These resolution-independent ICON CTL simulation biases are conceptually summarized in Figure 1b. In the following, we will explore an option to counteract against the biases found in ICON by increasing the near-surface moisture and use the resulting perturbation simulations as a test of the mechanistic understanding that we have discussed so far.

## 3.2 Sensitivity of large-scale precipitation biases to the surface wind threshold in the turbulence scheme

In this section, we investigate the efficacy of a parameter adjustment in addressing the double-ITCZ bias across resolutions in ICON. We test the  $U_{min}$  parameter adjustments (Sec. 2.1.2) to address the near-surface dry bias over the Warm Pool region with the goal of interrupting the circle of biases outlined in the previous paragraph. With an increase in  $U_{min}$ , we aim to increase evaporation and near-surface specific humidity. We expect that the increase in near-surface specific humidity will improve the Warm Pool precipitation and the double-ITCZ feature, through the mechanisms shown in Fig. 1.

#### 3.2.1 The choice of surface wind threshold

We use the  $40 \,\mathrm{km}$  resolution configuration to test the effects of varying the  $U_{min}$  threshold in the turbulence scheme. As the mean near-surface wind speed in the tropics is lower than in the extratropics, increases in  $U_{min}$  will have a stronger effect in the tropical region. Starting from the default  $1 \,\mathrm{m\,s^{-1}}$  setting in the CTL simulation, we change  $U_{min}$  to the values of  $\{0.5, 1.5, 2, 4, 5, 6\} \,\mathrm{m\,s^{-1}}$ . The resulting two-year mean precipitation biases against IMERG are shown in the first two rows of Fig. 5. Indeed, RMSEs in both precipitation and near-surface specific humidity decrease with an increase in  $U_{min}$  up to  $5 \,\mathrm{m\,s^{-1}}$  (PTB-5). Going beyond this value increases the RMSE in precipitation again and a positive precipitation bias develops over the Warm Pool in PTB-6.

Despite the decrease in the large-scale precipitation bias, precipitation biases over land remain or even worsen. Most apparent is the persistent dry bias over the islands in the Warm Pool region, but also an increase of the bias over South America and Africa, with increasing  $U_{min}$  (Fig. 5a-d). Observational data show an average wind speed of below  $2.5 \,\mathrm{m\,s^{-1}}$  at the surface over land, whereas the lower surface roughness over the ocean allows higher wind speeds with values of  $6.7 \,\mathrm{m\,s^{-1}}$  (Fig. 6). High values of  $U_{min}$  would therefore cut off more of the velocity distribution over land, leading to exaggerated evaporation over land and potentially drying of the soil. In the next experiments, we therefore distinguish between ocean and land when setting  $U_{min}$ , going back to the CTL setting of 1 m s<sup>-1</sup> for land in PTB-5\_1 and PTB-6\_1. This change reverses the increasing dry bias over South America and improves the RMSE precipitation score even further (Fig. 5 e, f).

Figure 5. Two-year mean large-scale precipitation bias with respect to IMERG averaged over 2004-2010 for different settings of the surface wind at 40 km resolution: PTB-0.5 (a), PTB-4 (b), PTB-5 (c), PTB-5\_1 (e), PTB-5\_1t (g), PTB-6 (d), PTB-6\_1 (f) and PTB-6\_1t (h). The corresponding global precipitation RMSE is stated beneath the panel for each sensitivity experiment. Statistically insignificant differences between IMERG and the experiments based on a two-sided z-test at  $\alpha = 0.1$  are grayed out. The global RMSE and mean error in near-surface specific humidity, calculated with respect to values derived from ERA5 reanalysis, is depicted in the inlays.

In a next step we address the concerns that the increased  $U_{min}$ , might adversely affect the velocity distribution by increasing the surface drag coefficient on the near-surface velocities and correspondingly slowing down the circulation. A comparison of the probability density function of monthly mean surface wind speeds in CTL, PTB-5 and PTB-5\_1 with ERA5 in Fig. 6 summarizes how the changes in  $U_{min}$  feed back on the wind distribution in the tropics. Compared to ERA5, CTL exhibits a skew to smaller surface wind speeds, both over ocean and land. Against expectations, the skew to smaller near-surface wind speeds over ocean is reduced instead of deteriorated via adjustment of  $U_{min}$  over the ocean in PTB-5 and PTB-5\_1 (Fig. 6b). For the values over land (Fig. 6c), which are mainly in the velocity bins from  $0 \text{ m s}^{-1}$  to  $2.5 \text{ m s}^{-1}$ , the main velocity peak remains essentially unchanged in location across all ICON simulations, although the probability of strong winds is reduced in PTB-5 relative to CTL.

305

300

295

In the following subsections, we will investigate the impacts of  $U_{min}$  changes on the global climate system through the mechanism described at the end of Section 3.1 and depicted in Figure 1. Specifically, we will analyze the impact on the global circulation in Section 3.2.2, explaining why the tropical circulation shows the unexpected strengthening. Section 3.2.3 explores how  $U_{min}$  leads to an unintended shift in origin of the inner tropical moisture. In Section 3.2.4, we will demonstrate that ICON exhibits an exaggerated moisture transport out of the boundary layer, which can lead to the deficit in near-surface moisture in the inner tropics. We discuss why this exaggerated vertical moisture transport necessitates high-biased evaporation (i.e. through increased  $U_{min}$ ) to maintain the global circulation. In Section 3.2.5, we discuss the the global impacts of the  $U_{min}$  changes and also how they change with retuning. Section 3.2.6 summarizes the impacts of  $U_{min}$  changes on the global dynamics and underlying mechanisms.

**Figure 6.** Normalized probability density functions for the monthly mean near-surface wind speed distributions between 10° S to 10° N in the experiments CTL, PTB-5 and PTB-5\_1. A comparison with ERA5 data is given for (a) the entire region, (b) over ocean only, and (c) over land only.

## 3.2.2 The impact of the surface wind threshold: Hadley and Walker circulation

An inspection of the spatially resolved near-surface wind fields and velocity potential of CTL, PTB-5 and PTB-5\_1 compared to ERA5 in Fig. 7 gives more nuanced insights into the impact of  $U_{min}$  changes on the atmospheric circulation. The choice of globally high  $U_{min}$  in PTB-5 (Fig. 6, Fig. 7e) leads to the expected overall reduction in wind speeds on land. Over land, this reduction in overall wind speed compared to CTL is reverted when separate  $U_{min}$  values are chosen for land and ocean in PTB-5\_1 (Fig. 7g). Over ocean, a general slowdown of the Trades in the Southern Hemisphere, e.g., off the West Coast of South America, can be noted in all PTB experiments (Fig. 7e,g). This not only impacts moisture transport within the Trades ranging from 30° to the inner tropics, but may also interfere with the wind-evaporation-SST feedback and related teleconnections. The inner tropics are an exception to the general slow-down of the circulation, which would be expected considering the increase in surface drag coefficient. In the next step, we will therefore investigate if the increased moisture supply fuels deep convection leading to an increased Walker circulation strength. This increased Walker circulation strength can then outweigh the influence of increased surface drag.

Indeed, changes in the velocity potential at 200 hPa confirm changes in the Walker circulation (Fig. 7 b, d, f, h and Tab. 2). In the ERA5 reanalysis (Fig. 7 b), the velocity potential field shows a peak in divergence over the Warm Pool and convergence near the coast of South America. For the CTL configuration, the Walker circulation strength is only 80 % of its strength in ERA5 (Fig. 7 c, Table 2). This is in line with the reduced divergence both over the Warm Pool and South America, the locations of rising branches of the Walker circulation. As already suspected from the near-surface wind plots, the  $U_{min}$  changes improve upon the Walker circulation strength, such that it exhibits the same strength as in ERA5. This is also apparent when comparing Fig. 7 d and f,h where the positive bias of CTL is reduced over the maritime continent. The reduced divergence over the Warm Pool area in CTL is almost entirely counterbalanced at the cost of an increased bias over South America, which shows a decreased divergence. In PTB5\_1, the increased convergence over South America decreases again compared to PTB5, further reducing the velocity potential bias (Fig. 7f, h).

The Hadley circulation is generally too weak in all experiments (Table 2). In CTL, the Hadley circulation strength is 70% of its strength in ERA5. In PTB5 and PTB5\_1 it drops to 60%. The differing response of the circulation in the innermost tropics and subtropical latitudes can be explained by the increased role of moisture in the inner tropics. More available near-surface moisture leads to increased deep convection strengthening the Walker circulation and outweighing the increased drag on the near-surface winds. In the higher latitudes this effect is less important, leading to the slow-down of the Trades and the Hadley circulation.

Figure 7. (a) ERA5 multi-year average near surface wind speeds for the years 2004-2010. Arrows visualizing the near-surface wind speed for ERA5 are superimposed to visualize the general circulation. (c, e, g) Two-year mean large-scale near-surface wind speed bias with respect to ERA5 for CTL and different settings of the near-surface wind speed limiter,  $U_{min}$ , in 40 km resolution: PTB-5, PTB-5\_1. (b) ERA5 multi-year average velocity potential  $\chi$  at 200 hPa for the years 2004-2010. (d,f,h) Two-year mean bias of the CTL, PTB-5 and PTB-5\_1 experiment with respect to the ERA5 velocity potential. Negative values of the velocity potential are found in regions of ascent and divergent motion; positive values in region of subsidence and convergence.

**Table 2.** Walker and Hadley circulation strengths for ERA5 as well as CTL, PTB-5 and PTB-5\_1. Indices are calculated following the approach by Tanaka et al. (2004) as described in Section 2.5.

| Experiment | Walker circulation Strength $\chi*_{max}$ / $10^7~{\rm m}^2~{\rm s}^{-1}$ | Hadley circulation Strength $\chi_{max}$ / $10^6~{\rm m^2~s^{-1}}$ |
|------------|---------------------------------------------------------------------------|--------------------------------------------------------------------|
| ERA5       | 1.16                                                                      | 3.23                                                               |
| CTL        | 0.94                                                                      | 2.17                                                               |
| PTB-5      | 1.27                                                                      | 1.98                                                               |
| PTB-5_1    | 1.22                                                                      | 1.98                                                               |

## 3.2.3 The impact of the surface wind threshold: Moisture sources

The changes in the spatial wind speed patterns (Fig. 7) have demonstrated that  $U_{min}$  has the potential to substantially impact the general atmospheric circulation. It increases the Walker circulation strength, but leads to a reduction of the Hadley circulation strength, especially noticeable in the region of the Southern Hemisphere trade winds. The trade winds transport a substantial amount of moisture into the inner tropics. Therefore, the change in  $U_{min}$  comes at the risk of changing the weighting of the inner-tropical versus the subtropical moisture sources. The analysis of the latent heat flux biases of CTL, PTB-5 and PTB-5\_1 with respect to ERA5 and OAFlux can give some indication of potential changes in the moisture sources (Fig. 8). In the bias plots, a positive evaporation bias with too much moisture release to the atmosphere is visualized in blue, whereas an insufficient evaporation is visualized in red. When interpreting these biases, it is important to keep in mind that the latent heat flux in ERA5 is high biased (cf. Martens et al. (2020); Song et al. (2021)) and larger than in OAFlux (Fig. 8 a, b). The biases against ERA5 will therefore tend to have a stronger bias towards more negative values. For consistency with the other plots with the other plots we are using ERA5 as reference, but also add in OAFlux.

CTL shows too little latent heat flux in the region of the ITCZ in the Pacific as well as in the Indian Ocean with respect to ERA5 (Fig. 8 c), whereas values compared to OAFlux almost perfectly capture the zonal-mean climatology. With the increase in  $U_{min}$  (Fig. 8e-h), the low bias in the Warm Pool region is reverted to a positive latent heat flux bias, leading to a too strong latent heat flux in the tropical region, even compared to the already high values simulated by ERA5. Although the spatial contrast in evaporation reduces, negative evaporation biases emerge in the trade wind regions at the coasts of Central- and South-America, key source regions for moisture transported into the tropical Pacific. These reductions are obscured in the zonal mean, because subtropical regions with climatologically lower wind speeds, such as the region South-East of Australia, develop positive evaporation biases. However, as this is a region of relatively little moisture-flux convergence, the moisture lingers in the region of evaporation. All in all, these findings indicate a reduction of the subtropical moisture contribution to moisture in the Pacific inner tropics.

**Figure 8.** Latent heat flux in reanalysis and 40 km simulations. (a) ERA5 multi-year average latent heat flux for the years 2004-2010. (b) OAFlux multi-year average latent heat flux for the years 2004-2010. **Two**-year mean large-scale latent heat flux change with respect to ERA for CTL, PTB-5 and PTB-5\_1 (c-h). The left column shows global biases, the right column the zonal mean bias with respect to ERA5 (land and ocean) and OAFlux (ocean only).

## 3.2.4 The impact of the surface wind threshold: Moisture transport

Next we investigate the potential role of vertical moisture transport out of the boundary layer at latitudes between 20° and 40° in reducing moisture transport by the trade winds into the inner tropics. This exaggerated vertical moisture transport out of the boundary layer could lead to the negative near-surface humidity bias in the inner tropics (cf. Fig. 3). If this is the case, then strong evaporation in the innermost tropics is necessary to compensate for the lack of moisture transported into the tropics.

However, this would also mean that the wind speed limiter fix addresses a downstream consequence and not the primary error in moisture transport.

Figure 9. Inverse residence time  $\tau_{R_{bl}}$  within the boundary layer up to 850 hPa: (a) ERA5 for the reference years 2004-2010, and differences of two-year average of CTL (d), PTB-5 (e) and PTB-5\_1 (f) compared to ERA5. The gray lines mark the 40° and 20° latitudes. (b) Difference of zonal distribution of  $\tau_{R_{bl}}$  of CTL, PTB-5 and PTB-5\_1 to ERA5 within the latitude band between 40° S and 40° N, restricted to ocean only. (c) Probability density function of  $\tau_{R_{bl}}$  of ERA5, CTL, PTB-5 and PTB-5\_1 within the latitude band between 40° S and 40° N, restricted to ocean only.

Figure 9 shows the inverse residence time  $\tau_{R_{bl}}$  of specific humidity within the boundary layer up to 850 hPa (Section 2.4). The reference values for ERA5 (Figure 9 a) exhibit shorter  $\tau_{R_{bl}}$  (larger  $\tau_{R_{bl}}^{-1}$ ) on land compared to ocean. On a large scale, the  $\tau_{R_{bl}}$  values simulated in CTL, PTB-5, and PTB-5\_1 are shorter compared to ERA5 (Figure 9d-f), demonstrating a too fast loss of boundary layer moisture to the free troposphere. This transport bias is well aligned with positive specific humidity biases in the free troposphere in the mid latitudes shown in Figure 3. The fast loss of boundary layer moisture is most pronounced in the higher latitudes around the poles, but tropical and midlatitude land as well as the trade wind region are also affected. In CTL, the weaker circulation manifests itself in a pronounced bias to longer residence times in the Northern ITCZ band over the Eastern Pacific, which is indicative of too weak moisture-flux divergence. With the  $U_{min}$  adaptation, this bias is corrected at the expense of an overall shortening of residence times in the trade wind region.

A quantitative analysis of  $\tau_{R_{bl}}$  over the ocean only demonstrates that the probability density function of  $\tau_{R_{bl}}$  is biased to shorter residence times without apparent improvements with the  $U_{min}$  patch (Figure 9c). The corresponding bias in the zonal distribution of  $\tau_{R_{bl}}$  over the ocean compared to ERA5 confirms that shorter residence times dominate over all latitudes in the ICON experiments (Figure 9b). The  $U_{min}$  adjustment does not correct the short  $\tau_{R_{bl}}$  values in the tropics, where values were already too short. In the region at 10 °N, where CTL exhibited long residence times, it overcompensates the bias. In the subtropics the bias is slightly improved. However, with increasing land masses, especially in the north, the bias increases again, almost doubling in the case of PTB-5\_1 at 40 °latitude.

Together, these results illustrate that moisture is exported too quickly out of the boundary layer to the free troposphere in the ICON simulations, which leaves less moisture for transport to the inner tropics. The change in  $U_{min}$  does not directly correct for this problem, but combats the consequences by artificially increasing the moisture supply to the boundary layer, specifically in low-wind regimes in the tropics. A more physically motivated correction would be to increase the transport of subtropical moisture into the tropics rather than the subtropical free troposphere. The consequent increase in inner-tropical moisture source would likewise lead to a increase in the Walker circulation. This would naturally increase the near-surface wind speeds and compensate for the missing evaporation in the tropics.

### 3.2.5 The impact of the surface wind threshold: global variables

As we have seen far-reaching impacts on the general atmospheric circulations when  $U_{min}$  is increased, we investigate the impact of the altered  $U_{min}$  on key global parameters in Fig.10. The first parameter pair is the asymmetry index  $A_p$  and symmetry index  $E_p$  of the precipitation distribution. All sensitivity experiments show a positive  $A_p$ , consistent with the observational references IMERG and GPCP, indicating that the inter-hemispheric energy imbalance is captured correctly.  $E_p$  shows negative values, starting to increase only at a  $U_{min}$  of 4 m s<sup>-1</sup>. The negative  $E_p$  for  $U_{min}$  limiters smaller than 4 m s<sup>-1</sup>, reflect the large dry bias over the Warm Pool. They indicates that the net energy input (NEI) to the atmosphere at the equator is most likely low biased. Since  $U_{min}$  preferentially effects evaporation in warm regions where evaporation is climatologically high and regions with low wind speeds, an increase in near-surface wind speed will have a more substantial effect in the inner-tropics, which

Figure 10. Simulation scores for all 40 km sensitivity experiments as a function of prescribed minimum surface wind speed. The values for a  $U_{min}$  of 1 m s<sup>-1</sup> are the CTL settings and were taken from the respective two years of the CTL simulation. Values for the asymmetric index  $A_p$ , symmetry index  $E_P$  as well as latent heat flux, near-surface specific humidity  $q_s$ , net top-of-the-atmosphere (TOA) imbalance, upwards radiative shortwave flux, outgoing long wave radiation (OLR) and global mean surface temperature  $T_S$  are shown. Each point depicts a one-year global average. For reference the corresponding values from IMERG, GPCP, ERA5 and CERES are depicted.

The simulations scores confirm that the correction in the low biased near-surface humidity with  $U_{min}$  comes at the cost of a high biased global latent heat flux compared to ERA5 (cf. Fig. 8 and Fig. C2 for spatial maps). Globally already enough moisture is supplied to the atmosphere in CTL, but there are biases in the moisture distribution associated with the global circulation.

The improvements of ITCZ bias comes at the cost of strong biases in the global year-mean net top-of-the-atmosphere (TOA) imbalance which drops by  $5\,\mathrm{W\,m^{-2}}$  for a  $U_{min}$  of  $6\,\mathrm{m\,s^{-1}}$  (Fig. 10). The main contribution to the changes in TOA imbalance arise from upward shortwave radiation which increases substantially as the cloud cover increases. Due to the specified-SST setup, the increased bias in TOA imbalance can only have small effects on the surface temperature  $T_s$  as temperature changes at the surfaces are restricted to sea ice and land only. Despite the negative TOA,  $T_s$  increases by around 1 K. This can be explained by the increase in free tropospheric moist static energy, as both specific humidity values and atmospheric temperature increase while the lapse rate over land does not change substantially (Zhang and Boos, 2023). The increase in  $T_s$  is reflected in the increase in outgoing long wave radiation. The tuning simulations 5-1t and 6-1t demonstrate that it is possible to readjust the radiative fluxes to expected values without strong deterioration in the improved precipitation distribution.

## 3.2.6 The impact of the surface wind threshold: Consequences on global dynamics

Within this section, we have answered the first part of the second and a large portion of the third research question group: The precipitation biases in coarse-scale  $40\,\mathrm{km}$  simulations can be addressed by increasing evaporation via an increase in  $U_{min}$  in the turbulence scheme. Mechanistically, the increase in  $U_{min}$  should lead to an increase in the drag on the surface winds. However, in the inner tropics, the increase in  $U_{min}$  also counterbalances the dry bias in the Warm Pool region through increased evaporation. This fuels deep convection and leads to an acceleration of the Walker circulation transporting spurious moisture over the Eastern and Central Pacific into the main convective region. The increase in the Walker circulation strength counterbalances the increases in drag on the near-surface wind speeds in the inner tropics, leading to an increase in near-surface wind speed over ocean. In the subtropics, which do not benefit from the increase in Walker circulation strength, the increased drag on the surface winds leads to a reduction in the trade wind strength.  $U_{min}$  compensates for three problems in the moisture transport:

- 1. for too strong vertical transport out of the boundary layer (cf. Fig. 9).
- 2. for the consequently too weak meridional equatorward near-surface specific humidity transport within the Trades from the subtropics (cf. Tab. 2), the main source region of inner tropical moisture (Schneider et al., 2014).
- 3. for the resulting dry bias in the tropics and too weak moisture transport by tropical circulations.

By drawing the moisture primarily from the inner tropics without fixing the problem in the vertical transport in the higher latitudes, it however acts more like a "patch" than a real fix. The improved representation of the ITCZ comes at the cost of a replacement of higher latitude (sub)tropical moisture sources by an increase in the inner-tropical moisture source, a reduction in the trade wind strength, and an increased global-mean latent heat flux bias (Fig.1c).

#### 440 3.3 Transferability to higher resolutions

465

After understanding the sensitivity to  $U_{min}$  changes in the coarser 40 km setup, we proceed to investigate whether the corresponding improvements with  $U_{min}$  also translate to higher 5 km resolution and additionally whether discarding parameterizations improves the representation of the mean atmospheric state. Due to computational constraints, we perform seasonal rather than year-long sensitivity tests. We focus on two MAM seasons, as this is the key season for the double-ITCZ bias (e.g., Si et al. (2021); Adam et al. (2018)). A comparison of the precipitation bias for the CTL and PTB5-1 simulations in 40 km, 5 km and 5 km-rParam is shown in Fig. 11. The 5 km setup with the full suite of parameterizations exhibits a less pronounced double-ITCZ feature in the East Pacific than rParam, its counterpart without gravity wave and deep convective parameterization (Fig. 11b,e; cf. Fig. 11c,f). Additionally, while the 5 km-Param setup reduces the global-mean near-surface specific humidity bias compared to the 40-km setup, the 5 km-rParam setup increases the global-mean near-surface specific humidity bias. However, especially in the MAM average, it becomes apparent that the precipitation dry bias over the Warm Pool is stronger in the parameterized simulations (Fig. 11e; cf. Fig. 11f), with an RMSE comparable to the 40 km setup. This is also true for the dry bias over the islands in the Maritime Continent, which disappears in rParam but is still apparent in the 5 km setup with the full set of parameterizations.

In both 5 km configurations, the seasonal-average double-ITCZ bias is reduced by  $U_{min}$  changes, and their RMSE in seasonal-average precipitation is less than in the comparable 40 km simulations. It can therefore be expected that the 5 km setups are likely to outperform the coarser resolution simulations in longer timeframe PTB-5\_1 simulations, as was already the case in the CTL simulations (Fig. 2 and Fig. 11 a, b, c). The positive precipitation bias appearing in the Indian Ocean with the  $U_{min}$  adjustments is not as prominent at higher resolution (Fig. 11h, i; cf. Fig. 11g). In contrast to the 40 km setup, both 5 km setups also show strong improvements in the precipitation biases over the tropical Atlantic.

Across all simulations, the  $40 \, \mathrm{km}$  and  $5 \, \mathrm{km}$ -Param setups show similar biases in the moisture distribution, but with reduced amplitude at increased resolution (Figure 12). In the CTL simulations, the near-surface dry bias is much more pronounced in  $5 \, \mathrm{km}$ -Param than in the  $40 \, \mathrm{km}$  and  $5 \, \mathrm{km}$ -Param setups. In all setups, the increase in  $U_{min}$  reduces the dry bias and changes the sign of the inner-tropical dry bias up to  $3 \, \mathrm{km}$ . In the setups with parameterizations, the  $U_{min}$  fix counterbalances the near-surface specific humidity dry bias, while leading to a slight deterioration of the free tropospheric moisture bias at higher latitudes. In the  $5 \, \mathrm{km}$ -rParam setup, the near-surface humidity dry bias can not be entirely counterbalanced by the  $U_{min}$  fix, however, the free tropospheric moist bias does not increase as much. The remaining near-surface bias in PTB-5\_1-rParam at  $5 \, \mathrm{km}$  resolution amounts to a percentage bias of less than  $3 \, \mathrm{percent}$ ,  $15 \, \mathrm{percent}$  of the background values in  $40 \, \mathrm{km}$  and  $5 \, \mathrm{km}$ . Overall, despite the differences described between the two resolutions, both  $40 \, \mathrm{km}$  and  $5 \, \mathrm{km}$  exhibit the same symptoms of a too-strong vertical moisture transport from the boundary layer into the free troposphere between  $20 \, \mathrm{to} \, 40^\circ$  latitude, which comes at the cost of reduced near-surface moisture transport into the tropics, and this remains the case in rParam.

f) and two-MAM averages from PTB-5\_1 simulations (g-i) for 40 km, 5 km and 5 km-rParam. The corresponding global precipitation RMSE is indicated below the panel for each experiment. Statistically insignificant differences between IMERG and the experiments are shown based on a two-sided z-test at  $\alpha = 0.1$ , and insignificant regions are grayed out. The global RMSE and mean error in the near-surface specific humidity, calculated with respect to the values derived from the Figure 11. Large-scale precipitation bias with respect to IMERG in: 2-yr averages from CTL simulations (a-c), two-MAM averages from CTL simulations (d-ERA5 reanalysis, are shown in the inlays.

**Figure 12.** Atmospheric specific humidity bias with respect to ERA5: all CTL for a 2 year average (a-c), all CTL for an average over two MAM seasons (d-f) and all PTB-5\_1 for an average over two MAM seasons (g-i) for 40 km, 5 km and 5 km-rParam. Statistically significant differences between ERA5 and the model data are shown, insignificant regions are hatched.

The increase in atmospheric specific humidity and deep convection in the inner tropics with modified  $U_{min}$  is also visible in the zonal-mean temperature biases in Fig.13. The tropical free troposphere warms in both resolutions as the convection ties the temperature profile to a different base-line moist adiabat with increased moisture supply. In the parameterized setups, the cold bias is overcompensated, resulting in a warm bias. The 5 km-rParam exhibited a stronger cold bias than the parameterized setups. Here, the cold bias is reduced. A possible reason for the differences in the zonal-mean temperatures could be differences in the entrainment rates into the convective cores in the parameterized and non-parameterized simulations. The more pronounced cold bias in the 5 km-rParam setups suggests that more entrainment takes place in the simulations relying on explicitly described deep convection, resulting in an overall colder tropical free troposphere.

475

**Figure 13.** Zonal mean temperature bias with respect to ERA5: all CTL for a 2 year average (a-c), all CTL for an average over two MAM seasons (d-f) and all PTB-5\_1 for an average over two MAM seasons (g-i) for 40 km, 5 km and 5 km-rParam. Statistically significant differences between ERA5 and the model data are shown, insignificant regions are hatched.

Next, we re-tune the 5 km setups with the same parameter adaptations used in 40 km to test the transferability of the tuning to higher resolution. The simulation scores from this exercise are shown in Fig.14. In many variables, 40 km and 5 km behave similarly, although it is also evident that the 5 km simulations exhibit a greater spread (i.e., variability) than the coarser 40 km counterpart for some variables. In  $A_s$  all simulations perform well compared to IMERG, although the 5 km-rParam simulations exhibit a small positive deviation. The  $E_p$  score is negatively biased in the untuned version and improves for both resolutions with the  $U_{min}$  adjustment. The improvement in near-surface specific humidity comes at the cost of an increased bias in latent heat flux. The TOA radiative imbalance shows increased variability in the 5 km simulations. The main simulation from which the sensitivity experiments were branched off shows maximum jumps in annual-mean TOA imbalance of 0.4 Wm<sup>-2</sup>, with no clear drift in the three simulated years. This suggests that newly resolved processes enhance variability, both in SW up (suggesting a role for low clouds) and OLR (suggesting a role for high clouds). The global-mean near-surface temperature  $T_s$  is the only parameter in which different setups show opposing trends with adjustment of the  $U_{min}$  and tuning: warming for parameterized setups and cooling for 5 km-rParam. For the 40 km and 5 km-rParam setups, the tuned versions are coming closer to the observational values. The 5 km simulations with parameterizations shows a deterioration in  $T_s$  bias.

Figure 14. Simulation scores for two MAM seasons of CTL, PTB-5\_1 and PTB-5\_1t sensitivity experiments in the 40 km, 5 km and 5 km-rParam setup. The 40 km scores are shown in different shades of red, the 5 km setups in shades of green and the 5 km-rParam in different shades of blue. Values for the asymmetric index  $A_p$ , symmetry index  $E_P$  as well as latent heat flux, near-surface specific humidity  $q_s$ , net top-of-the-atmosphere (TOA) imbalance, upwards radiative shortwave flux, outgoing long wave radiation (OLR) and global mean near-surface temperature  $T_S$  are shown. Each point depicts a one season average. For reference the corresponding values from IMERG, GPCP, ERA5 and CERES are listed.

Summarizing the main findings in this section, we demonstrated

- 1. that the  $U_{min}$  adjustment to counteract the development of the double ITCZ is similarly effective in 5 km simulations as in 40 km simulations and
- 2. that the 40 km tuning is transferable to the 5 km-Param simulations with sole adjustments of the orographic tuning parameters.

We also showed that the humidity and temperature fields change in a similar fashion to the adjustments of  $U_{min}$  at different resolutions.

### 4 Discussion and Conclusions

In our study, we compared the representation of the precipitation as simulated by ICON over a resolution hierarchy spanning from CMIP-resolution up to km-scale. Although some small-scale improvements, such as a better representation of precipitation over land with more pronounced orography were found, increased resolution and the discard of deep convective and non-orographic gravity wave parameterization could not resolve persistent large-scale precipitation biases. The large-scale precipitation biases found in our ICON resolution hierarchy are consistent with the double-ITCZ bias present through several CMIP generations (Tian et al., 2024). Additionally, we find a too-cold tropical free troposphere and a too-moist free troposphere in the subtropics. These biases are symptoms of a general misrepresentation of the exchange of energy and moisture between the inner- and outer tropics (Fig. 1 a, cf. Fig. 1 b).

To address the biases, we focused on the parameterizations that remain active at the km-scale, focusing on  $U_{min}$  in the bulk flux formulation of the turbulence scheme. It is known that the bulk flux formulation can lead to biased results compared to observations in both very low and very high wind regimes, producing a high bias in latent heat release (Hsu et al., 2022). Additionally, adjusting  $U_{min}$  yielded promising results in the Sapphire configuration of the ICON model (Hohenegger et al., 2022; Segura et al., 2025). The usage of increased  $U_{min}$  in the bulk flux formula resolved some of the precipitation biases. This improvement came at the cost of a global overestimation of evaporative fluxes. Furthermore, a deterioration of precipitation over the land made a separate adjustment necessary. This was not seen in Segura et al. (2025) due to their focus on the double-ITCZ feature in the Warm Pool region. Furthermore, we showed that the  $U_{min}$  adjustment led to similar changes at the km-scale as in our lower-resolution simulations, indicating that tuning lessons learned at lower resolution can be applied to km-scale simulations. Our analysis also gives new insights into the mechanisms by which  $U_{min}$  acts: By increasing evaporation, it counterbalances the low biases in the near-surface specific humidity in the tropics, enhancing deep convective activity, which increases free tropospheric temperatures and the temperature inversion in subsidence regions.

The fact that the model can not replicate the observed near-surface wind distribution hints at more overarching problems in the global near-surface moisture advection. The increase in  $U_{min}$  speed acts to address the too weak Walker circulation but does not address the lack of moisture transport from the higher (sub)tropical latitudes starting at 30 °latitude, which is likely

the origin of the tropical dry bias (Fig. 12 c). This also means that the additional supply of near-surface moisture is drawn from the wrong location, i.e. the inner tropics instead of the subtropics. Our moisture budget analysis shows an exaggerated vertical moisture transport out of the boundary layer (cf. Fig. 9), which explains the deficit in moisture available for transport into the inner tropics. The study by Lang et al. (2023) gives further evidence that exaggerated vertical moisture transport out of the boundary layer is related to turbulent transport. They also demonstrate that tropical moisture biases are greatly influenced by insufficient transport from the subtropics. In addition, they described the tendency of the TTE turbulence scheme used in this ICON configuration to act similarly to a convection scheme, also in higher atmospheric layers. Huusko et al. (2025) in turn demonstrate that there are no significant changes in the effects of turbulence boundary layer schemes with increasing resolution. Their and our findings combined can explain the positive moisture bias in the higher layers of the subtropical atmosphere - they are caused by the tendency of the turbulence scheme to transport moisture upwards out of the boundary layer like a convective scheme. This situation may be aggravated if parameterizations such as the deep convective parameterization are discarded prematurely. Another source of the bias might also be the shallow convection; however, as similar biases were also encountered in the ICON sapphire configuration without the shallow convection scheme (Segura et al., 2025), a bias in the used turbulence schemes or in the dynamical core is more likely.

In summary, the main findings of this work are:

- 1. Similar large-scale precipitation biases persist throughout the ICON resolution hierarchy up to km-scale resolution, although improvements over tropical islands in the Warm Pool region appear with increasing resolution and discard of parameterizations.
  - 2. Precipitation biases can be resolved by adapting undiscardable parameterizations but only yield an overall improvement of the precipitation fields if ocean and land are treated separately. The direction of regional changes hold over various resolutions. The main driving mechanism is a strengthening of the Walker circulation by increasing the supply of available moisture for the deep convective regions. However, the  $U_{min}$  fix proposed by (Segura et al., 2025) leads to additional drag on near-surface winds in higher latitudes. If the  $U_{min}$  limiter has to be used as a patch, it may be advisable to at least restrict it to the bulk flux formula for evaporation and not those for momentum exchange and sensible heat.
  - 3. There are strong indications that the  $U_{min}$  fix is not addressing the underlying core problem. A uniform wind threshold does not account for different regional circulation regimes and will lead to too high evaporation values in regions with low wind speed variations. Especially noticeable is also that the global latent heat flux and the  $20^{\circ}$  to  $40^{\circ}$  latitude free-tropospheric moisture bias are increased in the adjusted simulations. This demonstrates that there are problems with the source regions and transport of atmospheric moisture. The fix leads to a replacement of subtropical moisture by moisture originating from the inner tropics. Additionally, the increase in drag on the near-surface wind speeds leads to a slow down of the trade winds, such that the adjustment might compromise the representation of extra-tropical to tropical teleconnections.

- 4. Results indicate a two-fold faulty representation of the moisture transport (including by parameterizations) in ICON, and potentially in the many other models with a double-ITCZ bias:
  - In the subtropics, the vertical moisture transport out of the boundary layer is too strong, leaving too little moisture
    to be transported horizontally within the Trades into the tropics.
  - In the tropics, the lack of moisture reduces deep convective activity, which in turn reduces temperature inversions and slows down the Walker circulation. The reduced zonally anomalous circulations favors the development of a second ITCZ branch as moisture is not transported into the core regions for deep convection.

The differences in moisture and energy export found in the observations, the world simulated in the CTL ICON and the ICON with  $U_{min}$  adjustments are depicted in Fig.1.

The results of this study suggest that a two-fold climate modeling strategy would be greatly beneficial for the community: On the one side the advancement of high-resolution simulations by larger institutions with vast computational resources. On the other side, further development of necessary parameterizations and coarse-scale models. The persistence of large-scale biases across our ICON resolution hierarchy shows the potential of learning both from high- and low-resolutions. This approach was also demonstrated in the tuning of the precipitation biases in the NICAM model which specifically addressed and learned from resolution-independent biases (Takasuka et al., 2024). In the long term, scale-aware parameterizations or parameterizations with a seamless transition between convective and subgrid scale turbulent processes within one unified scheme (e.g., Tan et al. (2018)), might build a bridge between the high- and low-resolution modeling strategies.

Code and data availability. The ICON code was released in January 2024 under https://www.icon-model.org/news/news\_open\_source\_ release. The ICON XPP code release and source scripts are publicly available and published by (Müller et al., 2025b). The model data and postprocessing scripts needed to replicate the work presented in this paper will be made available upon publication in a zenodo archive. The ERA5 reanalysis data set is available from the climate data store (https://cds.climate.copernicus.eu/). IMERG is provided by NASA under https://gpm.nasa.gov/data/directory. GPCP data sets are maintained by NOAA under https://psl.noaa.gov/data/gridded/data. gpcp.html. CERES data can be obtained from NASA under https://ceres.larc.nasa.gov/data/. The OAFlux data are maintained by NCAR under https://climatedataguide.ucar.edu/climate-data/oaflux-objectively-analyzed-air-sea-fluxes-global-oceans. For postprocessing and data analysis python and CDO were used, the latter can be downloaded under https://code.mpimet.mpg.de/projects/cdo.

### Appendix A: Computation of ERA5 near-surface specific humidity

The near-surface specific humidity  $q_s$ , is not included as direct output of ERA5. We derive it via

$$q_s = \frac{0.622 \, p_v}{p_s - 0.378 \, p_v}.\tag{A1}$$

using the vapour pressure  $p_v$  in hPa, which is calculated using ERA5 output of 2-m dew point temperature  $T_D$  in Celsius and surface pressure  $p_s$  in hPa according to the Magnus formula

$$p_v = 6.112 \times 5 \exp\left(\frac{17.67 T_D}{T_D + 243.5 \,\mathrm{K}}\right).$$
 (A2)

## Appendix B: Computation of the velocity potential

The velocity potential  $\chi$  is obtained by computing the inverse Laplacian of the wind field divergence. Specifically, the wind field in the Helmholtz decomposition can be written as

$$\mathbf{u} = \nabla \times \psi - \nabla \chi,\tag{B1}$$

where  $\psi$  is the streamfunction. The divergence of the wind field is directly related to  $\chi$  via

$$\nabla \cdot \mathbf{u} = -\nabla^2 \chi,\tag{B2}$$

and  $\chi_{l,m}$  in the spectral space can be obtained by calculating the inverse Laplacian in spectral space according to

600 
$$\chi_{l,m} = -\frac{r^2 D_{l,m}}{l(l+1)},$$
 (B3)

where  $D_{l,m}$  are the spectral coefficients of the wind divergence  $\nabla \cdot \mathbf{u}$ , r is the earth's radius, l is the zonal wavenumber and m is the longitudinal wavenumber. We analyze  $\chi$  after transforming back to grid space.

# Appendix C: Near-surface humidity biases

**Figure C1.** Two-year mean large-scale surface specific humidity bias with respect to ERA5 (2004-2010) over the resolution hierarchy. Global mean root mean square errors are shown below the plots.

**Figure C2.** Two-year mean large-scale surface humdity bias with respect to ERA5 (2004-2010) for different settings of the surface wind in 40 km: : PTB-0.5, PTB-4, PTB-5\_1, PTB-5\_1t, PTB-6, PTB-6\_1 and PTB-6\_1t. Contours show the ERA5 climatology. The corresponding global RMSE is stated beneath the panel for each sensitivity experiment.

Author contributions. CAK had the idea for the study, conceptualized it, acquired computational resources, performed the analysis and wrote the paper manuscript. Model simulations and tuning were done by UN in 160 km, by AS in 80 km and CAK in 40 km and 5 km resolution. RJW provided input on many details of the analysis. LK advised on technical aspects on the model setup and provided the necessary input data. All authors contributed to the interpretation of the results and revised the paper manuscript.

Competing interests. The authors declare that they have no conflict of interest.

Acknowledgements. This research has been supported by the ETH Postdoctoral Fellowship Program (CAK), DWD's "Innovation Pro-610 gramme for Applied Researches and Developments" IAFE ICON-Seamless VH 4.7 (AS), the Swiss National Science Foundation Award PCEFP2 203376 (RJW) and the research unit FOR 2820 VolImpact (Grant 398006378) funded by the German Research Foundation (DFG) within the project VOLARC (UN). The authors specifically thank the ICON Seamless/ICON XPP Friends-circle and the XPP Precipitation task force for invaluable collaboration and support throughout the project year. This work greatly benefited from inspiring discussions with our colleagues Kristina Fröhlich, Karel Castro-Morales, Maike Ahlgrimm and Trang van Pham. It would not have been possible to run the 615 ICON XPP code on GPU without Roland Wirth, who fixed some remaining GPU porting bugs. Annika Lauber, Jonas Jucker and Michael Jähn from the C2SM team at ETH have been a great discussion contacts for technical issues encountered around ICON. We thank Doris Folini for insightful scientific discussion with the authors. Computing and data storage resources for the 40 km and 5 km simulations, were provided by the Swiss National Supercomputing Center (CSCS) in Lugano via the project: s1283 and lp67 as well as the early-testers access to the vCluster Todi on the new Alps infrastructure. William Sawyer, Christoph Müller as well as the technical staff at CSCS have been very 620 supportive of the ICON community when transitioning to the new Alps computing infrastructure. For the R2B5 simulations we thank the German Weather Services for providing computation resources on their system rts, for the R2B4 simulations, we thank the Deutsches Klima Rechenzentrum (DKRZ) for providing resources granted by its Scientific Steering Committee (WLA) under project ID bm0550. The authors thank two anonymous reviewers for constructive and detailed comments on the preprint version of this work.

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
