# Peer review of "Parameterization adaption needed to unlock the benefits of increased resolution for the ITCZ in ICON"

_EGUsphere, 2025_

## Referee Comment (RC1)

Review of

**Parameterization adaption needed to unlock the benefits of increased resolution for the ITCZ in ICON**

**Kroll et al.**

**General**

The authors examine the double intertropical convergence zone (IITCZ) bias in the ICON model across model resolutions. They find some improvement in the bias by controlling a critical velocity criteria of the turbulent scheme, which is offset by other emergent biases. But the overall conclusion is that increasing the model resolution, which allows discarding parameterizations at resolved scales, does not alleviate the IITCZ bias.

On the one hand, I support the approach of the work, given both the importance of the IITCZ bias and the increased prevalence of climate models with higher or variable grid resolutions. On the other hand, I see some critical problems with both the writing and the analysis by the authors. Overall, I would recommend publishing this work following a major and significant revision.

**General comments**

1. It is well known that the IITCZ bias is generally a property of coupled climate models, though some indications for processes leading the the bias are present in atmospheric models. It is therefore not surprising that the changes in the parameterizations and resolutions do not remedy the bias. More importantly, the authors use an atmospheric model (NWP). What are the surface boundary conditions? Clearly these would be important in diagnosing the IITCZ bias in the control runs, both in terms of the data being used, and in terms of the processes controlling surface heat fluxes. But there's no mention of this.

2. The work is riddled with inaccuracies, unclear statements, esoteric references, and Yoda-like sentences (e.g., 169–171). Lines 214 and 439 have referencing errors which should have been picked up in a reasonable proofing of the text. I urge the authors to do a better editorial job in this paper, which at its present form gives the impression of lack of attention to detail.

3. The scope of the analysis is limited. This in itself is fine, but needs to be acknowledged. Specifically, some of the hypothesized processes are discussed with no support, and are therefore speculative. In addition, a single parameter ($U_{min}$) is used as the control parameter. The strong response of the climate system to this single parameter demonstrates how complicated the task of bias reduction is, given the numerous other potential tuning parameters. Any general discussion of the 'root' cause of the IITCZ bias (the systematic variation of the resolution not withstanding) therefore in my opinion exaggerates the scope and implications of the study. My two recommendations in this regard is to either temper the speculative discussion of processes (see comments below) or provide more rigorous analysis (for example, the Seager decomposition may be helpful in the analysis of moisture transport).

**Comments by line number**

(abstract)      The double ITCZ (IITCZ) is itself not a precipitation bias. The "IITCZ bias" is a prominent tropical precipitation bias.

The 'root' cause only in the context the atmospheric model used here. Clearly, given that the IITCZ bias is a coupled model problem, and given the numerous mechanisms proposed as the cause of the IITCZ bias (e.g., cloud albedo, trickle bias, surface wind bias, etc.) the present work does not diagnose the actual 'root' cause.

biased how? Without specifying this, the following sentence is hard to interpret.

what do you mean by 'addresses'?

subtropical contribution to what?

what do you mean by endanger?

CMIP_(Tian and Dong, 2020) — similar missing space in many other places in the text.

"tendency to overestimate precipitation over ocean in the southern tropics and underestimate it at the equator" is inaccurate, unless used to describe the zonal mean precipitation. The IITCZ bias includes positive precipitation biases south of the equator in the eastern Pacific and Atlantic, underestimated precipitation in the equatorial Pacific, and positive precipitation biases in the western tropical Pacific.

please provide a citation in reference to the prominent problem.

increased wind convergence where?

caused ?by? moisture a more relevant reference in this context would be Marshall et al. (2014, "The ocean's role in setting the mean position of the ITCZ")

46&48  Not necessarily subtropical, it could be from any region outside the tropics.

The leading questions are themselves composed of questions. 1. Is actually three questions, and 2&3 are two questions each.

please explain what is the bulk-flux formulation.

$U_{min}$ is undefined please specify what is the schematic. Figure 13?

Given that this is an atmospheric model, how are ocean-atmosphere interactions represented? What are the surface boundary conditions? (prescribed SST, q-fluxes, etc.)

Section 2.3     $A_p$ was defined by Hwang and Frierson (2013) and $E_p$ was defined by Adam et al. (2016). Please reference the indices accordingly.

Figure 1        kg^-1 —> kg^{-1}   (also in all of the other figures)

              … near-surface specific humidity, calculated with respect to values derived from ERA5 reanalysis, is …

              Please refer to panel letters in the caption

?resp.?

Figure 2 and elsewhere, it would be better to describe units in square brackets, rather than following a divider, e.g., height / km  — > Height  [km]

citation error

216,220 and elsewhere,  \citet{} — > \citep{}

Figure 3       Would be interesting to check if the biases and resolution change affect the tropical cold dome (i.e., near the tropical tropopause), which is set by the ability of deep convection to penetrate higher into the upper troposphere.

229—232       Try to break this sentence schematic at the end of the text — > Figure 13

'vicious' is ambiguous

297—305:       This paragraph is speculative. What is the support for the changes in the intensity of the Walker circulation and marine inversion? What about meridional moisture transport?

what is the support for the changes in the Walker circulation strength?

what is the support for the assumed bias in atmospheric net energy input?

347—350       try to break this sentence

371—379       Again, this is highly speculative. What is the support for increased drag, or change in the Walker circulation? In relation to changes in surface winds, cite the relevant figure.

you refer to 'moisture transport' but Figure 10 only shows specific humidity. What is the support for the assumed dynamic changes?

Figure 11       color contrast is weak

$A_p$?

citation error.

---

## Referee Comment (RC2)

**Review of "Parameterization adaption needed to unlock the benefits of increased resolution for the ITCZ in ICON"**

Manuscript authors: Kroll et al

**Summary**

The authors use simulations at different resolutions and with the gustiness factor perturbed in the ICON model to examine mean precip biases. They argue that the double ITCZ bias persists at all resolutions but that increasing the minimum wind speed for the evaporation improves this, which however does not fix the larger underlying humidity biases across the tropics and subtropics.

There are some results of interest here, but I had a hard time following this manuscript and left it not especially convinced. In part this stems from issues with the writing, in part from the experimental design, and in part from the lack of uncertainty quantification.

Addressing all this adequately would take a serious amount of work, and as such I recommend rejection for eventual resubmission by the authors.

**Major comments**

**Experimental design**

It would be much cleaner if the highest resolution run didn't also have the convection and gravity wave schemes disabled as well, c.f. L120-122. As it stands, going from the 40-km model to the 5-km model, you're both increasing resolution **and** changing the model formulation. And even a 5-km grid is surely not fine enough to resolve the large number of convective updrafts that are smaller than 5x5km.

A relevant paper here is *Clark et al.* (2024), and references therein. In the GFDL AM4 model in an aquaplanet context, at high resolution the model behavior still changes dramatically depending on whether the convective parameterization is enabled or not.

This is especially concerning given that some fields such as the humidity and temperature biases are non-monotonic in resolution, with the change occurring going from the 40-km version in which the deep convective parameterization are activated to the 5-km version in which they are disabled.

**Uncertainty quantification**

C.f. L250-254, The 4, 5, and 6 experiments precip RMSE values are all within 0.03 of each other. Is that even a statistically significant separation? I worry about sampling uncertainty given the short durations of

the runs. I have the same concern about other results; apart from Fig. A1 there is very little discussion of uncertainty quantification and its implications for interpreting the results. It seems plausible that you're overinterpreting differences across simulations that aren't statistically well separated.

**Experiment names**

The results would be much easier to follow if the experiments had more descriptive names. So, instead of "R2B4," call it for example "160-km." The "PTB-X" simulation names are similarly unintuitive.

**Unclear arguments regarding model tuning**

I find the discussion of tuning peppered throughout the introduction to be frustrating. It feels speculative and almost conspiratorial, seeming to imply that the modeling groups are somehow, in the case of focusing on global-mean TOA radiative fluxes rather than regional circulation fidelity, doing something obviously wrong. In the discussion of tuning high-resolution models, L65-67, "misconception that all relevant processes are now resolved" is not justifiable, nor is it appropriate in tone. Model developers are fully aware of the physical scales of the various processes involved and how those compare to the scales resolved by their model.

The paper should incorporate *Zhao et al.* (2018a,b), who discuss how a tuning strategy targeting TOA fluxes was used to improve ITCZ simulation.

**English writing**

My impression is that English is not the lead author's native language for writing. There are quite a lot of sentences where the grammar and/or word choice are difficult to follow. In aggregate, these make for a somewhat jarring reading. At least one of the coauthors is a native English speaker, and so I know it is within the authors' collective ability to, in the revision, significantly tighten up the English writing. Here is a non-exhaustive list of sentences that I struggled with:

- L36-37
- L49
- L111-114
- L132-134
- L134-135
- L138-141
- L150-151
- L235-236
- L237-238 "vicious cycle" over the top
- L238-244 (break up this sentence)

Separate from the English usage, there were far too many typos. Please carefully proofread the revision carefully as a final step before submitting.

**Line-by-line comments**

**L13** "this could endanger the representation of the global circulation, energetic balance and teleconnections" confusing, due to the "could." Does it degrade these fields in your simulations or not?

**L15** what does "non-discardable" mean?

**L25** "bias has been central to the precipitation bias discussions" this reads funny to me; consider rephrasing

**L27-29** Correct and you should cite one or more papers that document these transient double ITCZ states, e.g. Magnusdottir and Wan 2008, `https://journals.ametsoc.org/view/journals/atsc/65/7/2007jas2518.1.xml`. And this preprint is particularly relevant: `https://essopenarchive.org/doi/full/10.22541/essoar.174017095.57302520`

**L42** Philander et al emphasize ocean-atmosphere coupling and continental geometry, making it an odd choice to cite regarding this claim about net energy imbalance; *Frierson et al.* (2013) would be more appropriate.

**L50-57** I don't find this discussion of the model tuning especially compelling, in large part because I'm struggling to follow it. Can you make your argument more precise and clear?

**L75-77** This sentence is meaningless to the reader, like me, who doesn't know what ICON XPP and ICON Sapphire are. I think you can omit this entirely, or if you want to keep it consider moving to the methods section or revising to provide more context

**L88** I would omit "the fuel for the hydrological cycle"; unneeded and too imprecise

**L94** "their implementation should receive more attention" this feels like too much of an editorializing statement to me in the context. I don't really know what 'minor-looking treatments' are beyond your summary having not read the Kawai et al paper, but based on your summary it's not obvious to me why they should indeed receive more attention.

**L111** I would omit the footnote; just include it in the parenthetical

**L118** calling 5 km convection "resolving" is a stretch…very few convective updrafts span 5x5 square km. "convection permitting" is a widely used and I think more appropriate choice.

**L165** Doesn't ERA5 directly output specific humidity?

**L214** fix the citations

**L215** A lot of typos through the end of this paragraph; feels sloppy.

**L229** What feedback loop?

**L229-232** Is this proposed feedback your idea? If yes, it feels rather speculative. If not, it needs citations.

**Fig. 6** What does "normalized" probability density function mean?

**Fig. 7** Is OAFlux ultimately a better product than ERA5 for the surface LH fluxes? If so, then why show the biases of the simulations against both? Why not just use OAFlux in the context of this whole discussion? Perhaps I missed something here.

**L342-344** I don't understand this. Why do the signs of the respective biases lead to this inference about "symptoms" vs. "root cause"?

**Fig. 8** symbols for control run are too faint

**References**

Clark, J. P., P. Lin, and S. A. Hill (2024), ITCZ Response to Disabling Parameterized Convection in Global Fixed-SST GFDL-AM4 Aquaplanet Simulations at 50 and 6 km Resolutions, *Journal of Advances in Modeling Earth Systems*, *16*(6), e2023MS003,968, doi:10.1029/2023MS003968.

Frierson, D. M. W., Y.-T. Hwang, N. S. Fučkar, R. Seager, S. M. Kang, A. Donohoe, E. A. Maroon, X. Liu, and D. S. Battisti (2013), Contribution of ocean overturning circulation to tropical rainfall peak in the Northern Hemisphere, *Nature Geoscience*, *6*(11), 940–944, doi:10.1038/ngeo1987.

Zhao, M., J.-C. Golaz, I. M. Held, H. Guo, V. Balaji, R. Benson, J.-H. Chen, X. Chen, L. J. Donner, J. P. Dunne, K. Dunne, J. Durachta, S.-M. Fan, S. M. Freidenreich, S. T. Garner, P. Ginoux, L. M. Harris, L. W. Horowitz, J. P. Krasting, A. R. Langenhorst, Z. Liang, P. Lin, S.-J. Lin, S. L. Malyshev, E. Mason, P. C. D. Milly, Y. Ming, V. Naik, F. Paulot, D. Paynter, P. Phillipps, A. Radhakrishnan, V. Ramaswamy, T. Robinson, D. Schwarzkopf, C. J. Seman, E. Shevliakova, Z. Shen, H. Shin, L. G. Silvers, J. R. Wilson, M. Winton, A. T. Wittenberg, B. Wyman, and B. Xiang (2018a), The GFDL Global Atmosphere and Land Model AM4.0/LM4.0: 1. Simulation Characteristics With Prescribed SSTs, *Journal of Advances in Modeling Earth Systems*, *10*(3), 691–734, doi:10.1002/2017MS001208.

Zhao, M., J.-C. Golaz, I. M. Held, H. Guo, V. Balaji, R. Benson, J.-H. Chen, X. Chen, L. J. Donner, J. P. Dunne, K. Dunne, J. Durachta, S.-M. Fan, S. M. Freidenreich, S. T. Garner, P. Ginoux, L. M. Harris, L. W. Horowitz, J. P. Krasting, A. R. Langenhorst, Z. Liang, P. Lin, S.-J. Lin, S. L. Malyshev, E. Mason, P. C. D. Milly, Y. Ming, V. Naik, F. Paulot, D. Paynter, P. Phillipps, A. Radhakrishnan, V. Ramaswamy, T. Robinson, D. Schwarzkopf, C. J. Seman, E. Shevliakova, Z. Shen, H. Shin, L. G. Silvers, J. R. Wilson, M. Winton, A. T. Wittenberg, B. Wyman, and B. Xiang (2018b), The GFDL Global Atmosphere and Land Model AM4.0/LM4.0: 2. Model Description, Sensitivity Studies, and Tuning Strategies, *Journal of Advances in Modeling Earth Systems*, *10*(3), 735–769, doi:10.1002/2017MS001209.

---

## Author Comment (AC1)

**Response to Review 1**

July 10, 2025

**Response to Review**

**General**

The authors examine the double intertropical convergence zone (IITCZ) bias in the ICON model across model resolutions. They find some improvement in the bias by controlling a critical velocity criteria of the turbulent scheme, which is offset by other emergent biases. But the overall conclusion is that increasing the model resolution, which allows discarding parameterizations at resolved scales, does not alleviate the IITCZ bias. On the one hand, I support the approach of the work, given both the importance of the IITCZ bias and the increased prevalence of climate models with higher or variable grid resolutions. On the other hand, I see some critical problems with both the writing and the analysis by the authors. Overall, I would recommend publishing this work following a major and significant revision.

We would like to thank Reviewer 1 for their detailed and constructive comments to our manuscript "Parameterization adaption needed to unlock the benefits of increased resolution for the ITCZ in ICON".

We address their suggestions by supporting our findings with additional analysis of the moisture budget and the general circulation, refine the methods description and reformulate sentences for clarity. Below we list the reviewer's comments and respond to them individually. The reviewer's comments are shown in black; our response is written in blue ink.

**General comments**

1. It is well known that the IITCZ bias is generally a property of coupled climate models, though some indications for processes leading the the bias are present in atmospheric models. It is therefore not surprising that the changes in the parameterizations and resolutions do not remedy the bias.

   Yes, the study uses an atmosphere model of the ICON XPP configuration [MFK+25]. The double ITCZ bias is present in both the coupled- and atmospheric-only model version, with a reduced magnitude in AMIP experiments. In our analysis, we prescribe sea surface temperatures, which allows us to isolate and clearly identify the atmospheric contribution to the bias, independent of a biased sea surface temperature or surface fluxes of a coupled model.

   The desire to address long-standing biases in climate models has motivated several researchers to increase model resolution (e.g. more recently, [MCCLT+22, MZZW23]). The existence of differing views on the effects of increased resolution or discard of parameterizations underscores the need to systematically investigate how increased resolution or the reduction of parameterizations affects model skill. Our study therefore aims to make a contribution to the ongoing discussion of how best to capitalize on the benefits of increased resolution, focusing on the double - ITCZ bias.

   More importantly, the authors use an atmospheric model (NWP). What are the surface boundary conditions? Clearly these would be important in diagnosing the IITCZ bias in the control runs, both in terms of the data being used, and in terms of the processes controlling surface heat fluxes. But there's no mention of this.

   Necessary boundary conditions like sea surface temperature (SST) and sea ice concentration (SIC) are prescribed based on 6H data interpolated from a monthly climatology. The CMIP6 forcing dataset from 1978-2020 was used to create these climatologies. By using the monthly

climatology in SST, the influence of the interannual variation of ENSO on precipitation variability is removed. This was mentioned in the previous version, but to make this clearer, we substantially refined the model and experiment section in the revised manuscript. In the course of this restructuring, we moved the description of the boundary condition to a more prominent position. It states: "In all experiments, we use prescribed climatological sea ice and sea surface temperature fields from the monthly values of the CMIP6 Forcing Datasets (input4MIPs, 1978-2020) as boundary conditions [DT18]. Prescribed climatological SSTs reduce the impact of interannual variability such as the influence of ENSO-events on the precipitation and separate the effects of model biases on SST representation and atmospheric processes."

2. The work is riddled with inaccuracies, unclear statements, esoteric references, and Yoda-like sentences (e.g., 169–171). Lines 214 and 439 have referencing errors which should have been picked up in a reasonable proofing of the text. I urge the authors to do a better editorial job in this paper, which at its present form gives the impression of lack of attention to detail.
The specific sentence in question has been revised to: "The ERA5 latent heat flux values are biased. Therefore, we additionally compare our data to the OAFlux dataset [FBH$^+$03, YW07, SDFT13, NCfARS22], which integrates satellite retrievals and three atmospheric reanalyses.". We have also corrected the references in lines 214 and 439. In order to address the general concern with respect to language and reading flow, we additionally carefully proofread the manuscript. To improve clarity and readability, we reformulated complicated passages of the manuscript. During this process, we also streamlined several sections of the text. For example, the revised manuscript now presents only the full set of results for PTB5 and shows the PTB6 in the summary tuning plot. In this way, the key results are more clearly highlighted and the readability is improved. For further examples of how we addressed language-related comments, please refer to our detailed responses under "Comments by line number."

3. The scope of the analysis is limited.
In order to address the reviewer's concern we integrated additional analysis focusing on the general circulation and moisture transport to support our argumentation chain. Specifically, as detailed below we investigate the Walker and Hadley circulation strength with a metric based on the velocity potential and perform a moisture budget decomposition to investigate the strong vertical moisture transport at 20 degree latitude.

This (the limited scope of analysis) in itself is fine, but needs to be acknowledged. Specifically, some of the hypothesized processes are discussed with no support, and are therefore speculative. In addition, a single parameter (Umin) is used as the control parameter. The strong response of the climate system to this single parameter demonstrates how complicated the task of bias reduction is, given the numerous other potential tuning parameters. Any general discussion of the 'root' cause of the IITCZ bias (the systematic variation of the resolution not withstanding) therefore in my opinion exaggerates the scope and implications of the study.
By isolating and analyzing the impact of one specific parameter, our goal is to provide a clear and focused step toward understanding and mitigating the double - ITCZ bias. The decision to focus on "Umin" in this study is based on two main considerations. First, a recent study [SBF$^+$24] proposed "Umin" as a solution to the precipitation bias in the Warm Pool. Our study complements their work by exploring the underlying global mechanism leading to an improvement in our ICON model, which employs the full suite of parametrizations. Second, a comprehensive analysis of the full spectrum of tuning parameters and their underlying mechanisms is beyond the scope of a single study. Gaining insight into why "Umin" helps reduce the double - ITCZ bias in ICON can ultimately inform more effective solutions with improved cost-benefit profiles, because, importantly, we do not advocate for the use of "Umin" due to its negative effects on atmospheric circulation, which became evident only through this parameter-specific investigation. To address the reviewer's concern regarding the discussion of root causes, we have carefully revised the manuscript to more clearly articulate the scope and limitations of our analysis. Specifically, we replaced the term "root cause" in the abstract with the phrase "important influence" in conjunction with subjunctive.
My two recommendations in this regard is to either temper the speculative discussion of processes (see comments below) or provide more rigorous analysis (for example, the Seager decomposition

may be helpful in the analysis of moisture transport).

We have addressed the reviewer's concerns regarding the discussion of results, as detailed in our responses to the individual comments below. In addition, we have substantially revised the manuscript. To further strengthen the analysis, we incorporated a more rigorous statistical evaluation and expanded the overall analysis. Specifically, in response to the concerns about moisture transport pathways, we now include additional diagnostics of the moisture budget (similar to those in Seager et al. 2010, but focused on the climatologies instead of climate change). Additionally, we analyze the strength of the Hadley and Walker circulations with a velocity potential based metric. The entire additional analysis is shown and discussed in the manuscript. Here we show two example results. First, the velocity potential, which we used to diagnose the Walker and Hadley Circulation strength. The circulation indexes for the circulation strength introduced by Tanaka et al., 2004 demonstrate that in CTL the Walker Circulation is too weak but improves in PTB-5. On the other hand, the Hadley Circulation strength is reduced in PTB-5 compared to CTL (compare Table 1). This backs up the hypothesis voiced in the previous version of the manuscript. The spatial maps of the velocity potential bias with respect to ERA5 can be seen in Figure 1.

Second, we show the difference between the moisture flux convergence above 850 hPa and "precipitation minus evaporation" (P-E) to demonstrate the bias in vertical transport out of the subtropical boundary layer that the ICON simulations exhibit. Figure 2 shows that the transport bias peaks at 20 North and 20-30 South, the regions where the specific humidity profiles show a moist bias. The wind surface limiter fix mainly improves the moisture transport in the tropical deep convective regions.

Table 1: Walker and Hadley Circulation Strengths for ERA5 as well as CTL, and PTB-5.

| Experiment | Walker Circulation Strength $\chi*_{max}$ / $10^7 \times m^2$ $s^{-1}$ | Hadley Circulation Strength $\chi_{max}$ / $10^6 \times m^2$ $s^{-1}$ |
|---|---|---|
| ERA5 | 1.16 | 2.02 |
| CTL | 0.94 | 2.17 |
| PTB-5 | 1.27 | 1.98 |

**Comments by line number**

1 (abstract) The double - ITCZ (IITCZ) is itself not a precipitation bias. The "IITCZ bias" is a prominent tropical precipitation bias.

The respective sentence now reads: "The double Inter-Tropical Convergence Zone (double-ITCZ) bias is a prominent tropical precipitation bias that has persisted over several climate model generations."

8 The 'root' cause only in the context the atmospheric model used here. Clearly, given that the IITCZ bias is a coupled model problem, and given the numerous mechanisms proposed as the cause of the IITCZ bias (e.g., cloud albedo, trickle bias, surface wind bias, etc.) the present work does not diagnose the actual 'root' cause.

The double - ITCZ is not a feature exclusively simulated by coupled models - this work (among many others) shows that the double - ITCZ bias can also occur in uncoupled model simulations. It is of course correct that the manuscript exclusively concentrates on the ICON model in an uncoupled setup; the statement with respect to the origin of the double-ITCZ bias is therefore restricted to this setup as well. We now emphasize this even more throughout the work. For example, the title of the work specifically states that the study is focusing on the ICON model, i.e. "Parameterization adaption needed to unlock the benefits of increased resolution for the ITCZ in ICON". In the abstract, it is stated that the work investigates the double-ITCZ in the ICON model, i.e. "In this work, we use the unique possibility offered by the new ICON XPP model configuration to study the double-ITCZ bias in a resolution hierarchy spanning from parameterized to resolved convection within a consistent modeling framework", before explaining the chain of biases leading to the expression of the double-ITCZ in this model. In order to address the concern of the definitiveness in the statement in the abstract, we reformulated as: "However, we highlight that an important underlying influence on the double-ITCZ

[Figure]

Figure 1: ERA5 multi-year average velocity potential $\chi$ at $200\,\mathrm{hPa}$ for the years 2004-2010. Two-year mean bias of the CTL, and PTB-5 experiment with respect to the ERA5 velocity potential. Negative values of the velocity potential are found in regions of ascent and divergent motion; positive values in region of subsidence and convergence.

[Figure]

Figure 2: Biases in the moisture transport out of the boundary layer in CTL, PTB-5 and PTB-5_1 with respect to ERA5. The two-year mean difference between vertically integrated moisture flux convergence above 850 hPa and precipitation minus evaporation (P-E) is shown.

in our uncoupled ICON seems to lie in insufficient moisture transport from the subtropics to the inner tropics."

9 biased how? Without specifying this, the following sentence is hard to interpret.
We modified the corresponding sentences to read: "However, we highlight that an important underlying influence on the double-ITCZ in ICON seems to lie in the insufficient transport of moisture from the subtropics to the inner tropics. The resulting low bias in tropical near-surface moisture substantially reduces deep convection over the Warm Pool, leading to a weakened Walker Circulation."

11 what do you mean by 'addresses'?
We replaced the word "addresses" with the word "resolves", it now states: "An increase in near-surface wind speed limiter resolves the low bias in near-surface moisture in the tropics, however it exacerbates a bias in the moisture source by increasing the inner tropical over the subtropical contribution."

12 subtropical contribution to what?
We clarified that the entire sentence is focusing on "moisture" by adding "to near-surface moisture" after the word "contribution". The revised sentence now reads: "An increase in near-surface wind speed limiter resolves the low bias in near-surface moisture in the tropics, however, it exacerbates a bias in the moisture source by increasing the inner tropical over the subtropical contribution to near-surface moisture."

13 what do you mean by endanger?
We replaced the word "endanger" with the word "deteriorate".

21 CMIP_(Tian and Dong, 2020) — similar missing space in many other places in the text.
We added in spaces where they were missing.

22 "tendency to overestimate precipitation over ocean in the southern tropics and underestimate it at the equator" is inaccurate, unless used to describe the zonal mean precipitation. The IITCZ bias includes positive precipitation biases south of the equator in the eastern Pacific and Atlantic, underestimated precipitation in the equatorial Pacific, and positive precipitation biases in the western tropical Pacific.
We deleted the original formulation and followed the reviewer's suggestion. The text now reads:

"Among them, the double - ITCZ bias (double - ITCZ) is the most prominent problem [MRB$^+$95, Lin07]. The double-IITCZ bias describes positive precipitation biases south of the equator in the eastern Pacific and Atlantic, as well as underestimated precipitation in the equatorial Pacific."

23 please provide a citation in reference to the prominent problem.
We now provide three citations for the double-ITCZ from the time span of 1995 to 2020: "Despite its importance, biases in the representation of precipitation within the Inter-Tropical Convergence Zone (ITCZ) have been a persistent challenge throughout model generations in the Coupled Model Intercomparison Project, CMIP (Tian and Dong, 2020). Among them, the double - ITCZ bias (double-ITCZ) is the most prominent problem (Mechoso et al., 1995; Lin, 2007)."

31 increased wind convergence where?
We reformulated to be more specific: "Over the Pacific, the natural double - ITCZ feature is associated with local minima in surface humidity and temperature along the equator (Zhang, 2001) and increased southeast Pacific ITCZ wind convergence (Halpern and Hung, 2001)".

36 caused ?by? moisture
We agree that the addition "caused moisture mass flux in the tropics." can lead to irritation and omitted it. We modified the sentence to read: "The net supply of moisture from higher latitudes leads to an excess of precipitation over evaporation in the tropics.".

42 a more relevant reference in this context would be Marshall et al. (2014, "The ocean's role in setting the mean position of the ITCZ")
We added the corresponding reference.

46 &48 Not necessarily subtropical, it could be from any region outside the tropics.
We now state "sub- and extra-tropical" instead of "subtropical" to account for this. The sentence now reads: "This underlines that the double-ITCZ problem cannot be investigated as an isolated tropical phenomenon: Sub- and extra-tropical biases in the energy budget can also be sources of the problem [KHFZ08, HF13], and tropical biases can likewise cause biases in the sub- and extra-tropical [DABBW22]. "

80 The leading questions are themselves composed of questions. 1. Is actually three questions, and 2 &3 are two questions each.
We reformulated the corresponding section. Specifically, we added "topic headings" for the research questions. The corresponding text now reads: "The focus of the paper lies on:

1. **Resolution and parameterization dependence of the double-ITCZ bias**: Can increased horizontal resolution and switching off deep convective and gravity wave parameterization improve the double-ITCZ bias? Are there common biases across resolutions? Where can resolution-dependent improvements be found?
   (*Addressed in Section 3.1 and 3.3*)

2. **Resolution-(in)dependent bias corrections:** To the extent that there are common (double-ITCZ) biases, how can they be addressed and can the same adjustments be applied at various resolutions?
   (*Addressed in Section 3.2 and 3.3*)

3. **Underlying mechanisms:** What are the underlying mechanisms leading to the double-ITCZ bias and how do the chosen adjustments ameliorate it?
   (*Addressed in Section 3 and Section 4, summarized in Schematic Figure 1, 5 and 14.*)"

86 please explain what is the bulk-flux formulation.
We now added more details to the bulk flux formulation for evaporation: "To address the second group of research questions, we focus on the choice of a minimum surface wind speed threshold ($U_{min}$) in the turbulence scheme of ICON XPP (see Section 2.1.2 and 3.2 for details). In the used bulk-flux formulation, the surface latent heat flux is a function of near-surface moisture and wind speeds. In this context $U_{min}$ sets the minimum wind speed for the calculation of the Richardson number, effectively

decreasing it in low wind speed regimes. Mechanistically, a higher threshold $U_{min}$, therefore, leads to an increased latent heat flux into the atmosphere under low wind conditions, and with it more available moisture for convection." The formula itself is also discussed when the bulk flux formulation for latent heat release is stated in section 3.2.

87 is undefined
We added a description of $U_{min}$, please see comment above.

104 please specify what is the schematic. Figure 13?
Yes, Figure 13 was meant. Within the restructuring process we decided to slowly build up Figure 13 and show the panels (a), (b) and (c) when they are discussed within the introduction and results section. This sentence was reformulated accordingly and now states: "The identified large-scale differences in the three scenarios will be visualized building-up on the circulation schematic (Fig. 1) and summarized in the Discussion and Conclusion section."

115 Given that this is an atmospheric model, how are ocean-atmosphere interactions represented? What are the surface boundary conditions? (prescribed SST, q-fluxes, etc.)
The model and experiment section was substantially revised to account for this comment and comments by reviewer 2. Addressing this comment, the discussion of the surface boundary condition was moved to a more prominent location in the text. It can now be found in the model and not the experiment section. The revised statement reads: "In all experiments, we use prescribed climatological sea ice and sea surface temperature fields from the monthly values of the CMIP6 Forcing Datasets (input4MIPs, 1978-2020) as boundary conditions [DT18]. Prescribed climatological SSTs reduce the impact of interannual variability such as the influence of ENSO-events on the precipitation and separate the effect of model biases in the SST representation and atmospheric processes."

Section 2.3 $A_p$ was defined by Hwang and Frierson (2013) and $E_p$ was defined by Adam et al. (2016). Please reference the indices accordingly.
For the sake of readability, we left this sentence as is. It is already clear that these indices come collectively from these two papers.

Figure 1 $kg^-1$ to $kg^{-1}$ (also in all of the other figures)
We corrected the typo in the respective figures.

Figure 1 ... near-surface specific humidity, calculated with respect to values derived from ERA5 reanalysis, is ...
We added the commas correspondingly.

Please refer to panel letters in the caption
We now refer to the panels in all captions.

203 ?resp.?
"respectively" is now spelled out.

Figure 2 and elsewhere, it would be better to describe units in square brackets, rather than following a divider, e.g., height / km — ¿ Height [km]
As the "[]" notation is not required by the ACP journals and the "/" notation is the more precise mathematical representation, we would like to refrain from changing the handling of the units. ACP journals require exponential writing of units, i.e. $W$ $m{-2}$, we therefore see no danger of confusion caused by multiple "/"s.

214 citation error
We added the missing citation.

216,220 and elsewhere, citet to citep
We made the corresponding changes.

**References**

[DABBW22] Yue Dong, Kyle C. Armour, David S. Battisti, and Edward Blanchard-Wrigglesworth. Two-Way Teleconnections between the Southern Ocean and the Tropical Pacific via a Dynamic Feedback. Journal of Climate, 35(19):6267–6282, October 2022.

[DT18] Paul J. Durack and Karl E. Taylor. PCMDI AMIP SST and sea-ice boundary conditions version 1.1.4, 2018.

[FBH+03] C. W. Fairall, E. F. Bradley, J. E. Hare, A. A. Grachev, and J. B. Edson. Bulk Parameterization of Air–Sea Fluxes: Updates and Verification for the COARE Algorithm. Journal of Climate, 16(4):571–591, February 2003.

[HF13] Yen-Ting Hwang and Dargan M. W. Frierson. Link between the double-Intertropical Convergence Zone problem and cloud biases over the Southern Ocean. Proceedings of the National Academy of Sciences, 110(13):4935–4940, March 2013.

[KHFZ08] Sarah M. Kang, Isaac M. Held, Dargan M. W. Frierson, and Ming Zhao. The Response of the ITCZ to Extratropical Thermal Forcing: Idealized Slab-Ocean Experiments with a GCM. Journal of Climate, 21(14):3521–3532, July 2008.

[Lin07] Jia-Lin Lin. The Double-ITCZ Problem in IPCC AR4 Coupled GCMs: Ocean–Atmosphere Feedback Analysis. Journal of Climate, 20(18):4497–4525, September 2007.

[MCCLT+22] E. Moreno-Chamarro, L.-P. Caron, S. Loosveldt Tomas, J. Vegas-Regidor, O. Gutjahr, M.-P. Moine, D. Putrasahan, C. D. Roberts, M. J. Roberts, R. Senan, L. Terray, E. Tourigny, and P. L. Vidale. Impact of increased resolution on long-standing biases in HighResMIP-PRIMAVERA climate models. Geoscientific Model Development, 15(1):269–289, 2022.

[MFK+25] Wolfgang A. Müller, Barbara Früh, Peter Korn, Roland Potthast, Johanna Baehr, Jean-Marie Bettems, Gergely Bölöni, Susanne Brienen, Kristina Fröhlich, Jürgen Helmert, Johann Jungclaus, Martin Köhler, Stephan Lorenz, Andrea Schneidereit, Reiner Schnur, Jan-Peter Schulz, Linda Schlemmer, Christine Sgoff, Trang V. Pham, Holger Pohlmann, Bernhard Vogel, Heike Vogel, Roland Wirth, Sönke Zaehle, Günther Zängl, Björn Stevens, and Jochem Marotzke. ICON: Towards vertically integrated model configurations for numerical weather prediction, climate predictions and projections. Bulletin of the American Meteorological Society, April 2025.

[MRB+95] C.R. Mechoso, A.W. Robertson, N. Barth, M.K. Davey, P. Delecluse, P.R. Gent, S. Ineson, B. Kirtman, M. Latif, H. Le Treut, T. Nagai, J.D. Neelin, S.G.H. Philander, J. Polcher, P.S. Schopf, T. Stockdale, M.J. Suarez, L. Terray, O. Thual, and J.J. Tribbia. The Seasonal Cycle over the Tropical Pacific in Coupled Ocean–Atmosphere General Circulation Models. Monthly Weather Review, 123(9):2825–2838, September 1995.

[MZZW23] Xinyu Ma, Shuyun Zhao, Hua Zhang, and Wuke Wang. The double-itcz problem in cmip6 and the influences of deep convection and model resolution. International Journal of Climatology, 43(5):2369–2390, 2023.

[NCfARS22] (Eds) National Center for Atmospheric Research Staff. The Climate Data Guide: OAFlux: Objectively Analyzed air-sea Fluxes for the global oceans, September 2022.

[SBF+24] Hans Segura, Clara Bayley, Romain Fiévet, Helene Gloeckner, Moritz Guenther, Lukas Kluft, Ann Kristin Naumann, Sebastian Ortega, Divya Sri Praturi, Marius Rixen, Hauke Schmidt, Winkler Marius, Cathy Hohenegger, and Stevens Bjorn. A single tropical rainbelt in global storm-resolving models: the role of surface heat fluxes over the warm pool, 2024.

[SDFT13]   David P. Schneider, Clara Deser, John Fasullo, and Kevin E. Trenberth. Climate Data Guide Spurs Discovery and Understanding. Eos, Transactions American Geophysical Union, 94(13):121–122, March 2013.

[YW07]     Lisan Yu and Robert A. Weller. Objectively Analyzed Air–Sea Heat Fluxes for the Global Ice-Free Oceans (1981–2005). Bulletin of the American Meteorological Society, 88(4):527–540, April 2007.

---

## Author Comment (AC2)

**Response to Review 2**

July 10, 2025

**Review of "Parameterization adaption needed to unlock the benefits of increased resolution for the ITCZ in ICON"**

**Manuscript authors: Kroll et al**

**Summary**

The authors use simulations at different resolutions and with the gustiness factor perturbed in the ICON model to examine mean precip biases. They argue that the double ITCZ bias persists at all resolutions but that increasing the minimum wind speed for the evaporation improves this, which however does not fix the larger underlying humidity biases across the tropics and subtropics. There are some results of interest here, but I had a hard time following this manuscript and left it not especially convinced. In part this stems from issues with the writing, in part from the experimental design, and in part from the lack of uncertainty quantification. Addressing all this adequately would take a serious amount of work, and as such I recommend rejection for eventual resubmission by the authors.

We thank the second reviewer for all their detailed and constructive comments. In response to the reviewer's main comments, we (1) include new high-resolution simulations with the full suite of parameterizations and discuss (in part of Section 3.3) how they compare to the setup with reduced parameterizations, (2) display statistical significance of the results, and (3) substantially revise and streamline the text. In subsequent paragraphs, we respond to the reviewer's detailed comments. The reviewer's comments are listed in black, and our response is listed in blue.

**Major comments**

**Experimental design**

It would be much cleaner if the highest resolution run didn't also have the convection and gravity wave schemes disabled as well, c.f. L120-122. As it stands, going from the 40km model to the 5km model, you're both increasing resolution and changing the model formulation. And even a 5km grid is surely not fine enough to resolve the large number of convective updrafts that are smaller than 5x5km. A relevant paper here is Clark et al. (2024), and references therein. In the GFDL AM4 model in an aquaplanet context, at high resolution the model behavior still changes dramatically depending on whether the convective parameterization is enabled or not. This is especially concerning given that some fields such as the humidity and temperature biases are nonmonotonic in resolution, with the change occurring going from the 40km version in which the deep convective parameterization is activated to the 5km version in which they are disabled.

We agree with the reviewer that an additional experiment in 5 km with a deep convective and gravity wave drag parametrization would benefit the study. We ran the corresponding experiment, the evaluation of which is now integrated in the manuscript. It replaces the original 5 km run without parameterization. We dedicate part of the results section 3.3 to a comparison of the 5 km setup with and without the deep convective and gravity wave parametrization. The introduction and discussion also includes a reference of Clark et al. 2024 [CLH24]: "In this context, it is important to note that

even at a horizontal resolution of 5 km, the necessity of deep convective parameterizations is still disputed with some studies showing improvements of the atmospheric respresentation with the discard of parameterizations [VTBP+20] and others showing deterioration and insufficiently resolved processes [CLH24]".

Indeed, the atmospheric representation is improved with the parameterizations; however, the double - ITCZ persists in both setups. In Figure R 1, we exemplarily show the temperature fields for the 40 km setup, the 5 km simulation including all parameterizations and the 5 km simulations with reduced parameterizations (rParam):

[Figure]

Figure 1: Zonal mean temperature bias with respect to ERA5: all CTL for a 2 year average (subfigures (a-c)), all CTL for an average over two MAM seasons (subfigures (d-f) and all PTB-5_1 for an average over two MAM seasons (subfigures (g-i)) for 40 km, 5 km and 5 km, rParam. Statistically significant differences between ERA and the model data are shown, insignificant regions are hatched. Statistical significance was tested with a two-sided ztest at $\alpha = 0.1$ after autocorrelation correction.

**Uncertainty quantification**

C.f. L250-254, The 4, 5, and 6 experiments precip RMSE values are all within 0.03 of each other. Is that even a statistically significant separation? I worry about sampling uncertainty given the short durations of the runs. I have the same concern about other results; apart from Fig. A1 there is very little discussion of uncertainty quantification and its implications for interpreting the results. It seems plausible that you're over-interpreting differences across simulations that aren't statistically well separated.

We have expanded the uncertainty quantification and marked statistically significant differences in all precipitation, temperature, and specific humidity figures. With respect to the differences in the precipitation fields: The reference is CTL or ERA5 depending on the respective quantities shown. For our research questions, the statistically significant difference between CTL and $U_{min} \in \{4,5,6\}$ is the important quantity, not the difference between $U_{min} \in \{4,5,6\}$. We can clearly show that the difference in precipitation between CTL and $U_{min} \in 4,5,6$ is statistically significant in the revised manuscript version. Also, there is a clear trend in the various tuning experiments. This shows that the change in the respective tuning parameter leads to systematic changes in the precipitation distribution and latent heat flux. In addition, we can explain the changes in model behavior based on physical arguments, again raising confidence in the results. We have chosen $U_{min} = 5ms^{-1}$ as the basis for the 5 km experiments. After demonstrating that $U_{min}$ can significantly alter the representation of the ITCZ compared to IMERG, we decided to invest our computation time in the requested simulation with parametrization rather than increasing the runtime of the $U_{min} \in \{4,5,6\}$.

Figure R 2 shows an example on how we are now restricting the analysis to statistically significant differences only.

[Figure]

Figure 2: Two-year mean large-scale precipitation bias with respect to IMERG 2004-2010 for different settings of the surface wind at 40 km resolution: PTB-0.5 (subfigure (a)), PTB-4 (subfigure (b)), PTB-5 (subfigure (c)), PTB-5_1 (subfigure (e)), PTB-5_1t (subfigure (g)), PTB-6 (subfigure (d)), PTB-6_1 (subfigure (f)) and PTB-6_1t (subfigure (h)). The corresponding global precipitation RMSE is stated beneath the panel for each sensitivity experiment. Statistically insignificant differences between IMERG and the experiments based on a two-sided z-test at $\alpha = 0.1$ are shown, and insignificant regions are grayed out. The global RMSE and mean error in near-surface specific humidity, calculated with respect to values derived from ERA5 reanalysis, is depicted in the inlays.

**Experiment names**

The results would be much easier to follow if the experiments had more descriptive names. So, instead of "R2B4," call it for example "160km." The "PTBX" simulation names are similarly unintuitive. We discarded of the ICON-specific language and only refer to the nominal resolution and experiment type, i.e. "CTL, 160 km" or "PTB-5, 160 km" to make the text easier to follow. PTB stands for

"perturbed", and the numbers refer to the wind speed limiter values. As the value of the wind speed limiter is important for context, we decided to keep this part of the nomenclature.

**Unclear arguments regarding model tuning**

I find the discussion of tuning peppered throughout the introduction to be frustrating. It feels speculative and almost conspiratorial, seeming to imply that the modeling groups are somehow, in the case of focusing on global mean TOA radiative fluxes rather than regional circulation fidelity, doing something obviously wrong.

We want to ensure the reviewer that there is definitely no intention to accuse climate modelers of doing something wrong, and some of the authors of this manuscript are modelers themselves. We are fully aware that the TOA tuning is followed by process tuning, and we intended only to highlight the potential trade-offs between TOA tuning and process tuning, where improving one might degrade the other. We modified the corresponding sentence to make this clearer. In this context, we also added references to the studies from GFDL. It now reads: "Most model evaluation workflows focus first on global mean top-of-the-atmosphere (TOA) fluxes and potentially surface energy fluxes [MSR+12], and then proceed to tune additional atmospheric fields and processes, for example, as outlined in Hourdin et al. 2017 [HMG+17]. There can be trade-offs between these different optimizations, although there are existing approaches to optimize the precipitation based on TOA flux [ZGH+18a, ZGH+18b]." Adding a caveat that model tuning/process tuning is necessary should not disregard the work of modelers. In contrast, we wish to promote their work, which is unfortunately sometimes not appreciated enough. The following sentences underline that it would be very valuable for the community to allow for more publications documenting model improvements: "However, in general, most published model evaluation workflows provide little information on regional energy budgets that are mechanistically important for the large-scale circulation and precipitation distribution. This poses a problem: Sometimes improvements in global mean energy fluxes introduce compensating errors in regional energy fluxes and lead to deterioration of the large-scale circulation and precipitation distribution."

In the discussion of tuning highresolution models, L65-67, "misconception that all relevant processes are now resolved" is not justifiable, nor is it appropriate in tone. Model developers are fully aware of the physical scales of the various processes involved and how those compare to the scales resolved by their model.

We agree with the reviewer that the model developers are fully aware that not all relevant processes are resolved. However, the authors have seen cases where this is not fully acknowledged, for example in proposals for additional high-resolution modeling efforts. We softened the sentence, and it now reads: "High computational costs, in combination with the hope that the many relevant processes are now resolved, can lead to a shortening of the model tuning process in high-resolution simulations."

The paper should incorporate Zhao et al. (2018a,b), who discuss how a tuning strategy targeting TOA fluxes was used to improve ITCZ simulation.

As stated above, the requested citations were integrated in the introduction to showcase a tuning strategy targeting the double-TCZ: "There can be trade-offs between these different optimizations, although there are existing approaches to optimize the precipitation distribution based on TOA flux [ZGH+18a, ZGH+18b].".

**English writing**

My impression is that English is not the lead author's native language for writing. There are quite a lot of sentences where the grammar and/or word choice are difficult to follow. In aggregate, these make for a somewhat jarring reading. At least one of the coauthors is a native English speaker, and so I know it is within the authors' collective ability to, in the revision, significantly tighten up the English writing. Here is a nonexhaustive list of sentences that I struggled with:

- L36-37: The sentence was reformulated as follows: "The net supply of moisture from higher latitudes leads to an excess of precipitation over evaporation in the tropics. "

- L49: We reformulated to: "It is critical to recognize the coupling between the double - ITCZ problem and biases in the energy budget in the context of model tuning, because the first step of model evaluation is in many cases the energy balance of the model [Wil20]."

- L111-114: We split this sentence in two. It now says: "ICON XPP is based on the parametrizations for radiation ecRad [HB18], cloud microphysics [Sei08], vertical diffusion (TTE) [MSZ$^+$07], convection [Tie89, BKJ$^+$08], subgrid scale orographic drag [LM97] and non-orographic gravity wave drag [OBS$^+$10]. The atmosphere is coupled to the land model JSBACH [RGG$^+$21]."

- L132-134: The corresponding sentence was simplified to: "The limiter $U_{min}$ is used to compute the wind-shear term of the near-surface moist Richardson number if the horizontal velocities meet the criterion $\boldsymbol{U}^2_{horizontal} < U^2_{min}$. "

- L134-135: This sentence was deleted.

- L138-141: The sentence was split into four for simplification: "The two $U_{min}$ settings that perform best in representing the large-scale annual-mean ITCZ, PTB-5 and PTB-6, with $U_{min}$ = 5 $ms^{-1}$ and 6 $ms^{-1}$, are further optimized. First, the wind speed is adapted for land and ocean separately. In PTB-5_1 and PTB-6_1, $U_{min}$ is set to = 1 $ms^{-1}$ over land to account for the slower near-surface wind speeds. Second, in PTB-5_1t, PTB-6_1t, the model is retuned to reestablish the CTL top-of-the-atmosphere (TOA) imbalance."

- L150-151: The sentence was split in two and rephrased to read: "For this purpose, a 1 degree horizontal resolution is chosen. This resolution is close to the tropical 1.4 degree horizontal resolution of the 160 km simulation."

- L235-236: The sentence was split apart and now reads: "We now understand the biases common to all resolutions and their potential role in the ITCZ formation. In a next step, we focus on the second group of research questions addressing parameter adjustments to counterbalance the identified issues in the atmospheric representation."

- L237-238 "vicious cycle" over the top: The word "vicious" was removed.

- L238-244 (break up this sentence): We changed the sentence and the description of $Umin$. The paragraph now reads: "Near-surface humidity is strongly influenced by evaporation. Evaporation $E$ can be tied to wind speed $U$ at the lowest atmospheric level via the bulk flux formula for evaporation

$$E = \rho_a C_E U(q_s - q_a), \tag{1}$$

where $\rho_a$ is the atmospheric density, $q_s$ near-surface specific humidity, $q_a$ atmospheric specific humidity and $C_E$ a proportionally constant. The bulk flux formulation shows that the latent heat flux is directly proportional to the near-surface wind speed. We do not change $U$ directly however, but adapt a hard-coded minor-looking treatment. In ICON-XPP's turbulence parameterisation, the surface wind used for the computation of the Richardson number has a lower prescribed threshold $U_{min}$ with $U = MAX(U_{min}, U)$. Increasing the default value of $U_{min}$ from 1 decreases the Richardson number in low-wind regimes, enhancing the proportionally constant $C_E$. This increases turbulence and, consequently, the turbulent fluxes. In total, the bulk flux formula for latent heat release thus suggests an increase in horizontal wind speed low-limiter, $U_{min}$, as a potential lever to address the dry bias over the Warm Pool and the related double - ITCZ biases. It is important to note that changes in $U_{min}$ can similarly impact the turbulent fluxes of sensible heat and momentum next to the targeted latent heat flux. "

Separate from the English usage, there were far too many typos. Please carefully proofread the revision carefully as a final step before submitting.

We carefully proofread the revision before submitting it.

**Line by line comments**

**L13** "this could endanger the representation of the global circulation, energetic balance and teleconnections" confusing, due to the "could." Does it degrade these fields in your simulations or not?

Yes, the global circulation is affected as a consequence of the increased drag on the surface winds. This is visible in the slowdown of the trade winds (Fig. 5). As intended by the fix, the latent heat flux increases. However, the latent heat flux bias shifts and increases the relative contribution of inner-tropical moisture (Fig. 7), which changes the net energy input into the atmosphere. Both changes are degrading the corresponding fields and can not be counterbalanced as they are part of the solution to the precipitation bias. The slowdown of the trade winds endangers the teleconnections related to

the wind-evaporation SST feedback. The top-of-the-atmosphere imbalance also changes considerably. However, this change can be corrected via re-tuning. We changed the wording to "can" to be more definite.

**L15** what does "nondiscardable" mean?

"Nondiscardable" refers to parametrizations which are needed at the given resolution because the corresponding processes cannot be explicitly resolved. An example would be radiation or turbulence.

**L25** "bias has been central to the precipitation bias discussions" this reads funny to me; consider rephrasing

The sentence was rephrased and now reads: "The simulated double - ITCZ results in strong local precipitation biases and influences the El Niño–Southern Oscillation and other large-scale climate phenomena [HK14, ZDCT14]. Therefore, identifying the cause of the double - ITCZ is a key step towards improving climate models.".

**L27-29** Correct and you should cite one or more papers that document these transient double ITCZ states, e.g. Magnusdottir and Wan 2008, https://journals.ametsoc.org/view/journals/atsc/65/7/2007jas2518.1.xml. And this preprint is particularly relevant:
https://essopenarchive. org/doi/full/10.22541/essoar.174017095.57302520

We now cite both the proposed paper and preprint.

**L42** Philander et al emphasize ocean atmosphere coupling and continental geometry, making it an odd choice to cite regarding this claim about net energy imbalance; Frierson et al. (2013) would be more appropriate.

Note that the "net energy input" includes the energy flux from the ocean to the atmosphere, and thus this statement is consistent with the work of Philander et al. We now additionally cite Frierson et al. (2013).

**L50-57** I don't find this discussion of the model tuning especially compelling, in large part because I'm struggling to follow it. Can you make your argument more precise and clear?

This comment was addressed in the section above titled "Unclear arguments regarding model tuning". Please refer to our previous statements and the revised manuscript.

**L75-77** This sentence is meaningless to the reader, like me, who doesn't know what ICON XPP and ICON Sapphire are. I think you can omit this entirely, or if you want to keep it consider moving to the methods section or revising to provide more context

This sentence is important to include for readers who are familiar with ICON, and we believe what is relevant to take away from it is sufficiently clear even to readers not familiar with ICON. We kept this sentence as is.

**L88** I would omit "the fuel for the hydrological cycle"; unneeded and too imprecise

Omitted.

**L94** "their implementation should receive more attention" this feels like too much of an editorializing statement to me in the context. I don't really know what 'minorlooking treatments' are beyond your summary having not read the Kawai et al paper, but based on your summary it's not obvious to me why they should indeed receive more attention.

Unlike "conventional" tuning parameters, "minor locking treatments" are hardcoded in the code and act as limiters in physical quantities. They are much less regularly revised, i.e. when the model resolution is increased. We now choose a softer formulation: "As "minor-looking treatments" can have impacts comparable to the exchange of parameterization schemes [KYKY22], their implementation merits more attention."

**L111** I would omit the footnote; just include it in the parenthetical

The footnote is removed.

**L118** calling 5 km convection "resolving" is a stretch...very few convective updrafts span 5x5 square km. "convection permitting" is a widely used and I think more appropriate choice.

We switched to convection permitting throughout.

**L165** Doesn't ERA5 directly output specific humidity?

Yes, ERA5 does output the atmospheric specific humidity, and we show the corresponding value in the zonal bias plots. However, the near-surface humidity values at $10\,m$ are not available in the archive.

**L214** fix the citations

Done.

**L215** A lot of typos through the end of this paragraph; feels sloppy.

The two typos were corrected and formulations were adjusted for an easier reading flow.

**L229** What feedback loop?

A stronger Walker circulation would lead to even more increased near-surface moisture in the Warm Pool, which constitutes a feedback. To avoid any confusion about this term, we remove the words "through a feedback loop."

**L229-232** Is this proposed feedback your idea? If yes, it feels rather speculative. If not, it needs citations.

Yes, it is our idea - based on physical arguments. We test this feedback loop in Section 3.2. In order to give the reader some guidance, we outline the hypothesis we are testing at this point. In order to account for the reviewer's concern, we also expanded the analysis and give more details on the changes in circulation and moisture transport in the revised manuscript version. The additional analysis includes an evaluation of the Walker and Hadley circulations and a moisture budget analysis.

**Fig. 6** What does "normalized" probability density function mean?

Normalized means that the area under probability density function integrates to the value of 1. This complies with the standard mathematical definition, we therefore did not add any further description to avoid suggesting otherwise.

**Fig. 7** Is OAFlux ultimately a better product than ERA5 for the surface LH fluxes? If so, then why show the biases of the simulations against both? Why not just use OAFlux in the context of this whole discussion? Perhaps I missed something here.

OAFlux does not provide all the quantities needed for the other bias plots (that is, all the temperature and humidity plots of the atmosphere). To remain in a consistent framework that is also closed energetically, we show the difference to ERA5 and only add OAFlux as complementing information.

**L342-344** I don't understand this. Why do the signs of the respective biases lead to this inference about "symptoms" vs. "root cause"?

We reformulated giving more details: "The high bias in latent heat flux suggests that enough moisture is supplied to the atmosphere without the wind speed limiter fix. The increase in positive latent heat flux bias in the inner tropics with the wind speed limiter fix shows that the inner-tropical moisture source is increasingly overrepresented at the cost of higher-latitude moisture sources within the trade wind regions. The coincidental increase in the higher-latitude moist bias in the free latitudes demonstrates that the moisture is transported vertically instead of horizontally into the inner tropics, which is likely to lead to the dry bias in the inner tropics. These three findings point to an underlying problem in moisture transport for which $U_{min}$ only combats the symptoms rather than the root cause.".

**Fig. 8** symbols for control run are too faint

We adjusted the chosen color map. The corresponding figure now appears as this Figure R 3.

[Figure]

Figure 3: Tuning scores for all 40 km sensitivity experiments as a function of prescribed minimum surface wind speed. The values for a $U_{min}$ of 1 are the CTL settings and where taken from the respective two years of the CTL simulation. Values for the asymmetric index $A_p$, symmetry index $E_P$ as well as latent heat flux, near-surface specific humidity $q_s$, net top-of-the-atmosphere (TOA) imbalance, upwards radiative shortwave flux, outgoing long wave radiation (OLR) and global mean surface temperature $T_S$ are shown. Each point depicts one global year average. For reference the corresponding values from IMERG, GPCP, ERA5 and CERES are depicted.

**References**

Clark, J. P., P. Lin, and S. A. Hill (2024), ITCZ Response to Disabling Parameterized Convection in Global FixedSST GFDLAM4 Aquaplanet Simulations at 50 and 6 km Resolutions, Journal of Advances in Modeling Earth Systems, 16(6), e2023MS003,968, doi:10.1029/2023MS003968.

Frierson, D. M. W., Y.T. Hwang, N. S. Fučkar, R. Seager, S. M. Kang, A. Donohoe, E. A. Maroon, X. Liu, and D. S. Battisti (2013), Contribution of ocean overturning circulation to tropical rainfall peak in the Northern Hemisphere, Nature Geoscience, 6(11), 940–944, doi:10.1038/ngeo1987.

Zhao, M., J.C. Golaz, I. M. Held, H. Guo, V. Balaji, R. Benson, J.H. Chen, X. Chen, L. J. Donner, J. P. Dunne, K. Dunne, J. Durachta, S.M. Fan, S. M. Freidenreich, S. T. Garner, P. Ginoux, L. M. Harris, L. W. Horowitz, J. P. Krasting, A. R. Langenhorst, Z. Liang, P. Lin, S.J. Lin, S. L. Malyshev, E. Mason, P. C. D. Milly, Y. Ming, V. Naik, F. Paulot, D. Paynter, P. Phillipps, A. Radhakrishnan, V. Ramaswamy, T. Robinson, D. Schwarzkopf, C. J. Seman, E. Shevliakova, Z. Shen, H. Shin, L. G. Silvers, J. R. Wilson, M. Winton, A. T. Wittenberg, B. Wyman, and B. Xiang (2018a), The GFDL Global Atmosphere and Land Model AM4.0/LM4.0: 1. Simulation Characteristics With Prescribed SSTs, Journal of Advances in Modeling Earth Systems, 10(3), 691–734, doi:10.1002/2017MS001208.

Zhao, M., J.C. Golaz, I. M. Held, H. Guo, V. Balaji, R. Benson, J.H. Chen, X. Chen, L. J. Donner, J. P. Dunne, K. Dunne, J. Durachta, S.M. Fan, S. M. Freidenreich, S. T. Garner, P. Ginoux, L. M. Harris, L. W. Horowitz, J. P. Krasting, A. R. Langenhorst, Z. Liang, P. Lin, S.J. Lin, S. L. Malyshev, E. Mason, P. C. D. Milly, Y. Ming, V. Naik, F. Paulot, D. Paynter, P. Phillipps, A. Radhakrishnan, V. Ramaswamy, T. Robinson, D. Schwarzkopf, C. J. Seman, E. Shevliakova, Z. Shen, H. Shin, L. G. Silvers, J. R. Wilson, M. Winton, A. T. Wittenberg, B. Wyman, and B. Xiang (2018b), The GFDL Global Atmosphere and Land Model AM4.0/LM4.0: 2. Model Description, Sensitivity Studies, and Tuning Strategies, Journal of Advances in Modeling Earth Systems, 10(3), 735–769, doi:10.1002/2017MS001209.

**References**

[BKJ+08]  Peter Bechtold, Martin Köhler, Thomas Jung, Francisco Doblas-Reyes, Martin Leutbecher, Mark J. Rodwell, Frederic Vitart, and Gianpaolo Balsamo. Advances in simulating atmospheric variability with the ECMWF model: From synoptic to decadal time-scales. Quarterly Journal of the Royal Meteorological Society, 134(634):1337–1351, 2008. _eprint: https://rmets.onlinelibrary.wiley.com/doi/pdf/10.1002/qj.289.

[CLH24]  Joseph P. Clark, Pu Lin, and Spencer A. Hill. ITCZ Response to Disabling Parameterized Convection in Global Fixed-SST GFDL-AM4 Aquaplanet Simulations at 50 and 6 km Resolutions. Journal of Advances in Modeling Earth Systems, 16(6):e2023MS003968, June 2024.

[HB18]  Robin J. Hogan and Alessio Bozzo. A Flexible and Efficient Radiation Scheme for the ECMWF Model. Journal of Advances in Modeling Earth Systems, 10(8):1990–2008, 2018. _eprint: https://agupubs.onlinelibrary.wiley.com/doi/pdf/10.1029/2018MS001364.

[HK14]  Yoo-Geun Ham and Jong-Seong Kug. Effects of Pacific Intertropical Convergence Zone precipitation bias on ENSO phase transition. Environmental Research Letters, 9(6):064008, May 2014.

[HMG+17]  Frédéric Hourdin, Thorsten Mauritsen, Andrew Gettelman, Jean-Christophe Golaz, Venkatramani Balaji, Qingyun Duan, Doris Folini, Duoying Ji, Daniel Klocke, Yun Qian, Florian Rauser, Catherine Rio, Lorenzo Tomassini, Masahiro Watanabe, and Daniel Williamson. The Art and Science of Climate Model Tuning. Bulletin of the American Meteorological Society, 98(3):589–602, March 2017.

[KYKY22]  Hideaki Kawai, Kohei Yoshida, Tsuyoshi Koshiro, and Seiji Yukimoto. Importance of Minor-Looking Treatments in Global Climate Models. Journal of Advances in Modeling Earth Systems, 14(10):e2022MS003128, October 2022.

[LM97]  François Lott and Martin J. Miller. A new subgrid-scale orographic drag parametrization: Its formulation and testing. Quarterly Journal of the Royal Meteorological Society, 123(537):101–127, 1997. _eprint: https://rmets.onlinelibrary.wiley.com/doi/pdf/10.1002/qj.49712353704.

[MSR+12]  Thorsten Mauritsen, Bjorn Stevens, Erich Roeckner, Traute Crueger, Monika Esch, Marco Giorgetta, Helmuth Haak, Johann Jungclaus, Daniel Klocke, Daniela Matei, Uwe Mikolajewicz, Dirk Notz, Robert Pincus, Hauke Schmidt, and Lorenzo Tomassini. Tuning the climate of a global model. Journal of Advances in Modeling Earth Systems, 4(3):2012MS000154, March 2012.

[MSZ+07]  Thorsten Mauritsen, Gunilla Svensson, Sergej S. Zilitinkevich, Igor Esau, Leif Enger, and Branko Grisogono. A Total Turbulent Energy Closure Model for Neutrally and Stably Stratified Atmospheric Boundary Layers. Journal of the Atmospheric Sciences, 64(11):4113–4126, November 2007.

[OBS+10] Andrew Orr, Peter Bechtold, John Scinocca, Manfred Ern, and Marta Janiskova. Improved Middle Atmosphere Climate and Forecasts in the ECMWF Model through a Nonorographic Gravity Wave Drag Parameterization. Journal of Climate, 23(22):5905 – 5926, 2010. Place: Boston MA, USA Publisher: American Meteorological Society.

[RGG+21] Christian H. Reick, Veronika Gayler, Daniel Goll, Stefan Hagemann, Marvin Heidkamp, Julia E. M. S. Nabel, Thomas Raddatz, Erich Roeckner, Reiner Schnur, and Stiig Wilkenskjeld. JSBACH 3 - The land component of the MPI Earth System Model: documentation of version 3.2. page 4990986, February 2021. Artwork Size: 4990986 Medium: application/pdf Publisher: MPI für Meteorologie Version Number: 1.

[Sei08] Axel Seifert. A revised cloud microphysical parameterization for COSMO-LME., 2008.

[Tie89] M. Tiedtke. A Comprehensive Mass Flux Scheme for Cumulus Parameterization in Large-Scale Models. Monthly Weather Review, 117(8):1779 – 1800, 1989. Place: Boston MA, USA Publisher: American Meteorological Society.

[VTBP+20] Jesús Vergara-Temprado, Nikolina Ban, Davide Panosetti, Linda Schlemmer, and Christoph Schär. Climate Models Permit Convection at Much Coarser Resolutions Than Previously Considered. Journal of Climate, 33(5):1915 – 1933, 2020.

[Wil20] Martin Wild. The global energy balance as represented in CMIP6 climate models. Climate Dynamics, 55(3-4):553–577, August 2020.

[ZDCT14] Honghai Zhang, Clara Deser, Amy Clement, and Robert Tomas. Equatorial signatures of the Pacific Meridional Modes: Dependence on mean climate state. Geophysical Research Letters, 41(2):568–574, January 2014.

[ZGH+18a] M. Zhao, J.-C. Golaz, I. M. Held, H. Guo, V. Balaji, R. Benson, J.-H. Chen, X. Chen, L. J. Donner, J. P. Dunne, K. Dunne, J. Durachta, S.-M. Fan, S. M. Freidenreich, S. T. Garner, P. Ginoux, L. M. Harris, L. W. Horowitz, J. P. Krasting, A. R. Langenhorst, Z. Liang, P. Lin, S.-J. Lin, S. L. Malyshev, E. Mason, P. C. D. Milly, Y. Ming, V. Naik, F. Paulot, D. Paynter, P. Phillipps, A. Radhakrishnan, V. Ramaswamy, T. Robinson, D. Schwarzkopf, C. J. Seman, E. Shevliakova, Z. Shen, H. Shin, L. G. Silvers, J. R. Wilson, M. Winton, A. T. Wittenberg, B. Wyman, and B. Xiang. The GFDL Global Atmosphere and Land Model AM4.0/LM4.0: 2. Model Description, Sensitivity Studies, and Tuning Strategies. Journal of Advances in Modeling Earth Systems, 10(3):735–769, March 2018.

[ZGH+18b] M. Zhao, J.-C. Golaz, I. M. Held, H. Guo, V. Balaji, R. Benson, J.-H. Chen, X. Chen, L. J. Donner, J. P. Dunne, K. Dunne, J. Durachta, S.-M. Fan, S. M. Freidenreich, S. T. Garner, P. Ginoux, L. M. Harris, L. W. Horowitz, J. P. Krasting, A. R. Langenhorst, Z. Liang, P. Lin, S.-J. Lin, S. L. Malyshev, E. Mason, P. C. D. Milly, Y. Ming, V. Naik, F. Paulot, D. Paynter, P. Phillipps, A. Radhakrishnan, V. Ramaswamy, T. Robinson, D. Schwarzkopf, C. J. Seman, E. Shevliakova, Z. Shen, H. Shin, L. G. Silvers, J. R. Wilson, M. Winton, A. T. Wittenberg, B. Wyman, and B. Xiang. The GFDL Global Atmosphere and Land Model AM4.0/LM4.0: 1. Simulation Characteristics With Prescribed SSTs. Journal of Advances in Modeling Earth Systems, 10(3):691–734, March 2018.

---

## Author Response (AR1)

**Response to Review 1**

August 8, 2025

**Response to Review**

**General**

The authors examine the double intertropical convergence zone (IITCZ) bias in the ICON model across model resolutions. They find some improvement in the bias by controlling a critical velocity criteria of the turbulent scheme, which is offset by other emergent biases. But the overall conclusion is that increasing the model resolution, which allows discarding parameterizations at resolved scales, does not alleviate the IITCZ bias. On the one hand, I support the approach of the work, given both the importance of the IITCZ bias and the increased prevalence of climate models with higher or variable grid resolutions. On the other hand, I see some critical problems with both the writing and the analysis by the authors. Overall, I would recommend publishing this work following a major and significant revision.

We would like to thank Reviewer 1 for their detailed and constructive comments to our manuscript "Parameterization adaption needed to unlock the benefits of increased resolution for the ITCZ in ICON"

We address their suggestions by supporting our findings with additional analysis of the moisture budget (residence times in boundary layer) and the general circulation, refine the methods description and reformulate sentences for clarity. Below we list the reviewer's comments and respond to them individually. The reviewer's comments are shown in black; our response is written in blue ink.

**General comments**

1. It is well known that the IITCZ bias is generally a property of coupled climate models, though some indications for processes leading the the bias are present in atmospheric models. It is therefore not surprising that the changes in the parameterizations and resolutions do not remedy the bias.

Yes, the study uses an atmosphere model of the ICON XPP configuration [MFK+25]. The double ITCZ bias is present in both the coupled- and atmospheric-only model version, with a reduced magnitude in AMIP experiments. In our analysis, we prescribe sea surface temperatures, which allows us to isolate and clearly identify the atmospheric contribution to the bias, independent of a biased sea surface temperature or surface fluxes of a coupled model.

The desire to address long-standing biases in climate models has motivated several researchers to increase model resolution (e.g. more recently, [MCCLT+22, MZZW23]). The existence of differing views on the effects of increased resolution or discard of parameterizations underscores the need to systematically investigate how increased resolution or the reduction of parameterizations affects model skill. Our study therefore aims to make a contribution to the ongoing discussion of how best to capitalize on the benefits of increased resolution, focusing on the double - ITCZ bias.

More importantly, the authors use an atmospheric model (NWP). What are the surface boundary conditions? Clearly these would be important in diagnosing the IITCZ bias in the control runs, both in terms of the data being used, and in terms of the processes controlling surface heat fluxes. But there's no mention of this.

Necessary boundary conditions like sea surface temperature (SST) and sea ice concentration (SIC) are prescribed based on 6H data interpolated from a monthly climatology. The CMIP6 forcing dataset from 1978-2020 was used to create these climatologies. By using the monthly

climatology in SST, the influence of the interannual variation of ENSO on precipitation variability is removed. This was mentioned in the previous version, but to make this clearer, we substantially refined the model and experiment section in the revised manuscript. In the course of this restructuring, we moved the description of the boundary condition to a more prominent position. It states: "In all experiments, we use prescribed 6-hourly climatological sea ice and SST fields interpolated from the monthly climatological values of the CMIP6 Forcing Datasets (input4MIPs, 1978-2020) as boundary conditions [DT18]. Prescribed climatological SSTs reduce the impact of interannual variability, such as the influence of ENSO events on precipitation. In addition, they separate the effect of model biases in the SST representation and atmospheric processes."

- 2. The work is riddled with inaccuracies, unclear statements, esoteric references, and Yoda-like sentences (e.g., 169–171). Lines 214 and 439 have referencing errors which should have been picked up in a reasonable proofing of the text. I urge the authors to do a better editorial job in this paper, which at its present form gives the impression of lack of attention to detail. The specific sentence in question has been revised to: "The ERA5 latent heat flux values are high biased [MSW+20, SND+21]. Therefore, we also compare our data to the OAFlux dataset [SDFT13, NCfARS22], which integrates satellite retrievals and three atmospheric reanalysis.". We have also corrected the references in lines 214 and 439. In order to address the general concern with respect to language and reading flow, we additionally carefully proofread the manuscript. To improve clarity and readability, we reformulated complicated passages of the manuscript. During this process, we also streamlined several sections of the text. For example, the revised manuscript now presents only the full set of results for PTB5 and shows the PTB6 in the summary tuning plot. In this way, the key results are more clearly highlighted and the readability is improved. For further examples of how we addressed language-related comments, please refer to our detailed responses under "Comments by line number" or the tracked-changed manuscript.
- 3. The scope of the analysis is limited.

In order to address the reviewer's concern we integrated additional analysis focusing on the general circulation and moisture transport to support our argumentation chain. Specifically, as detailed below we investigate the Walker and Hadley circulation strength with a metric based on the velocity potential and perform a moisture budget analysis to investigate the strong vertical moisture transport out of the boundary layer based on residence times within the boundary layer.

This (the limited scope of analysis) in itself is fine, but needs to be acknowledged. Specifically, some of the hypothesized processes are discussed with no support, and are therefore speculative. In addition, a single parameter (Umin) is used as the control parameter. The strong response of the climate system to this single parameter demonstrates how complicated the task of bias reduction is, given the numerous other potential tuning parameters. Any general discussion of the 'root' cause of the IITCZ bias (the systematic variation of the resolution not withstanding) therefore in my opinion exaggerates the scope and implications of the study.

It is our goal is to provide a clear and focused step toward understanding and mitigating the double - ITCZ bias by isolating and analyzing the impact of one specific parameter. The decision to focus on "Umin" in this study is based on two main considerations. First, a recent study [SBF+25] proposed "Umin" as a solution to the precipitation bias in the Warm Pool. Our study complements their work by exploring the underlying global mechanism leading to an improvement in our ICON model, which employs the full suite of parametrizations. Second, a comprehensive analysis of the full spectrum of tuning parameters and their underlying mechanisms is beyond the scope of a single study. Gaining insight into why "Umin" helps reduce the double - ITCZ bias in ICON can ultimately inform more effective solutions with improved cost-benefit profiles, because, importantly, we do not advocate for the use of "Umin" due to its negative effects on atmospheric circulation, which became evident only through this parameter-specific investigation. To address the reviewer's concern regarding the discussion of root causes, we have carefully revised the manuscript to more clearly articulate the scope and limitations of our analysis. Specifically, we replaced the term "root cause" in the abstract with the phrase "a key driver".

My two recommendations in this regard is to either temper the speculative discussion of processes

(see comments below) or provide more rigorous analysis (for example, the Seager decomposition may be helpful in the analysis of moisture transport).

We have addressed the reviewer's concerns regarding the discussion of results, as detailed in our responses to the individual comments below. In addition, we have substantially revised the manuscript. To further strengthen the analysis, we incorporated a more rigorous statistical evaluation and expanded the overall analysis. Specifically, in response to the concerns about moisture transport pathways, we now include additional diagnostics of the moisture budget (based on [PO83, PO92], but focusing on climatological residence times in the boundary layer). Additionally, we analyze the strength of the Hadley and Walker circulations with a velocity potential based metric. The entire additional analysis is shown and discussed in the manuscript (Section 2.4, 2.5, 3.2.2, 3.24). Here we show two example results. First, the velocity potential, which we used to diagnose the Walker and Hadley Circulation strength. The circulation indexes for the circulation strength introduced by Tanaka et al., 2004 [TIK04] demonstrate that in CTL the Walker Circulation is too weak but improves in PTB-5. On the other hand, the Hadley Circulation strength is reduced in PTB-5 compared to CTL (compare Table 1). This backs up the hypothesis voiced in the previous version of the manuscript. The spatial maps of the velocity potential bias with respect to ERA5 can be seen in Figure 1.

Second, we show the difference between the inverse residence time within the boundary layer up to 850 hPa compared to ERA5, which we compute based on a moisture budget analysis. Figure 2 shows that the residence time in the ICON simulations is shorter than in the ERA5 reference, i.e. due to too fast vertical export out of the boundary layer, a bias which is not corrected by the wind speed limiter fix. Further details and interpretation are given in the revised manuscript.

Table 1: Walker and Hadley circulation strengths for ERA5 as well as CTL, PTB-5 and PTB-5\_1. Indices are calculated following the approach by Tanaka et al. (2004).

| Experiment           | Walker circulation Strength $\chi *_{max}$ | Hadley circulation Strength $\chi_{max}$ / |
|----------------------|--------------------------------------------|--------------------------------------------|
|                      | $10^7 \text{ m}^2 \text{ s}^{-1}$          | $10^6 \ {\rm m^2 \ s^{-1}}$                |
| ERA5                 | 1.16                                       | 3.23                                       |
| $\operatorname{CTL}$ | 0.94                                       | 2.17                                       |
| PTB-5                | 1.27                                       | 1.98                                       |
| $PTB-5_1$            | 1.22                                       | 1.98                                       |

**Comments by line number**

1 (abstract) The double - ITCZ (IITCZ) is itself not a precipitation bias. The "IITCZ bias" is a prominent tropical precipitation bias.

The respective sentence now reads: "The double Inter-Tropical Convergence Zone (double-ITCZ) bias is a persistent tropical precipitation bias over many climate model generations."

8 The 'root' cause only in the context the atmospheric model used here. Clearly, given that the IITCZ bias is a coupled model problem, and given the numerous mechanisms proposed as the cause of the IITCZ bias (e.g., cloud albedo, trickle bias, surface wind bias, etc.) the present work does not diagnose the actual 'root' cause.

The double - ITCZ is not a feature exclusively simulated by coupled models - this work (among many others) shows that the double - ITCZ bias can also occur in uncoupled model simulations. It is of course correct that the manuscript exclusively concentrates on the ICON model in an uncoupled setup; the statement with respect to the origin of the double-ITCZ bias is therefore restricted to this setup as well. We now emphasize this even more throughout the work. For example, the title of the work specifically states that the study is focusing on the ICON model, i.e. "Parameterization adaption needed to unlock the benefits of increased resolution for the ITCZ in ICON". In the abstract, it is stated that the work investigates the double-ITCZ in the ICON model, i.e. "In this work, we study the double-ITCZ bias in an ICON XPP resolution hierarchy spanning from parameterized to explicitly described deep convection within a consistent framework.", before explaining the chain of biases leading to the expression of the double-ITCZ in this model. In order to address the concern of the

Figure 1: ERA5 multi-year average velocity potential  $\chi$  at 200 hPa for the years 2004-2010. Two-year mean bias of the CTL, and PTB-5 experiment with respect to the ERA5 velocity potential. Negative values of the velocity potential are found in regions of ascent and divergent motion; positive values in region of subsidence and convergence.

Figure 2: Inverse residence time  $\tau_{R_{bl}}$  within the boundary layer up to 850 hPa: (a) ERA5 for the reference years 2004-2010, and differences of two-year average of CTL (d), PTB-5 (e) and PTB-5\_1 (f) compared to ERA5. The gray lines mark the 40 and 20 latitude. (b) Difference of zonal distribution of  $\tau_R$  of CTL, PTB-5 and PTB-5\_1 to ERA5 within the latitude band between 40 S and 40 N, restricted to ocean only. (c) Probability density function of  $\tau_R$  of ERA5, CTL, PTB-5 and PTB-5\_1 within the latitude band between 40 S and 40 N, restricted to ocean only.

definitiveness in the statement in the abstract, we reformulated as: "However, we highlight that a key driver of the double-ITCZ bias in ICON seems to lie in the insufficient moisture transport from the subtropics to the inner tropics."

9 biased how? Without specifying this, the following sentence is hard to interpret.

We modified the corresponding sentences to read: "However, we highlight that a key driver of the double-ITCZ bias in ICON seems to lie in the insufficient moisture transport from the subtropics to the inner tropics. The resulting low bias in tropical near-surface moisture reduces deep convection over the Warm Pool, leading to a weakened Walker circulation. These biases ultimately culminate in the double-ITCZ feature."

**11 what do you mean by 'addresses'?**

We rephrased the corresponding sentence. It now states: "Increasing the near-surface wind speed limiter improves tropical near-surface moisture but exacerbates the bias in the moisture source, increasing the inner tropical contribution at the expense of the subtropics."

**12 subtropical contribution to what?**

We clarified that the entire sentence is focusing on "moisture" by adding "to near-surface moisture". The revised sentence now reads: "An increase in near-surface wind speed limiter resolves the low bias in near-surface moisture in the tropics, however, it exacerbates a bias in the moisture source by increasing the inner tropical over the subtropical contribution to near-surface moisture."

**13 what do you mean by endanger?**

We replaced the word "endanger" with the word "degrade".

21 CMIP\_(Tian and Dong, 2020) — similar missing space in many other places in the text. We added in spaces where they were missing.

22 "tendency to overestimate precipitation over ocean in the southern tropics and underestimate it at the equator" is inaccurate, unless used to describe the zonal mean precipitation. The IITCZ bias includes positive precipitation biases south of the equator in the eastern Pacific and Atlantic, underestimated precipitation in the equatorial Pacific, and positive precipitation biases in the western tropical

**Pacific.**

We deleted the original formulation and followed the reviewer's suggestion. The text now reads: "Among them, the double-ITCZ bias is the most prominent problem [MRB+95, Lin07]. It describes positive precipitation biases south of the equator in the eastern Pacific and Atlantic, as well as underestimated precipitation in the equatorial Pacific."

23 please provide a citation in reference to the prominent problem.

We now provide three citations for the double-ITCZ from the time span of 1995 to 2020: "Despite its importance, biases in the representation of precipitation within the Inter-Tropical Convergence Zone (ITCZ) have been a persistent challenge throughout many model generations in the Coupled Model Intercomparison Project (CMIP) (Tian and Dong, 2020). Among them, the double-ITCZ bias is the most prominent problem (Mechoso et al., 1995; Lin, 2007)."

31 increased wind convergence where?

The corresponding sentence was deleted during the restructuring of the manuscript.

36 caused ?by? moisture

The corresponding sentence was deleted during restructuring.

42 a more relevant reference in this context would be Marshall et al. (2014, "The ocean's role in setting the mean position of the ITCZ")

We added the corresponding reference.

46 & 48 Not necessarily subtropical, it could be from any region outside the tropics.

We now state "sub- and extra-tropical" instead of "subtropical" to account for this. The sentence now reads: "This underlines that the double-ITCZ problem cannot be investigated as an isolated tropical phenomenon: Sub- and extratropical biases in the energy budget can also be sources of the problem [KHFZ08, HF13, KHX+19], and tropical biases can likewise cause biases in the sub- and extratropics [HMS17, DABBW22, FDW+23]."

80 The leading questions are themselves composed of questions. 1. Is actually three questions, and 2 &3 are two questions each.

We reformulated the corresponding section. Specifically, we added "topic headings" for the research questions. The corresponding text now reads: "We focus on the following questions:

- 1. Resolution and parameterization dependence of the double-ITCZ bias: Can increased horizontal resolution and switching off deep convective and gravity wave parameterization improve the double-ITCZ bias? Are there common biases across resolutions? Where can resolution-dependent improvements be found? (Addressed in Section 3.1 and 3.3)
- 2. **Resolution-(in)dependent bias corrections:** To the extent that there are common (double-ITCZ) biases, how can they be addressed and can the same adjustments be applied at various resolutions?

(Addressed in Section 3.2 and 3.3)

3. **Underlying mechanisms:** What are the underlying mechanisms leading to the double-ITCZ bias in ICON and how do the chosen adjustments ameliorate it? (Addressed in Section 3 and Section 4, summarized in Schematic Figure 1)."

86 please explain what is the bulk-flux formulation.

We added an entirely new section (Section 2.1.2, subheading "Default and  $U_{min}$  adapted ICON") which explains the bulk flux formulation in the context of the  $U_{min}$  changes in detail and at one place. In the introduction, we now refer to this new section.

87 is undefined

We added a detailed description of  $U_{min}$  in a separate section, please see comment above.

104 please specify what is the schematic. Figure 13?

Yes, Figure 13 was meant. Within the restructuring process we decided to move this Figure to the introduction and reference it there.

115 Given that this is an atmospheric model, how are ocean-atmosphere interactions represented? What are the surface boundary conditions? (prescribed SST, q-fluxes, etc.)

The model and experiment section was substantially revised to account for this comment and comments by reviewer 2. Addressing this comment, the discussion of the surface boundary condition was moved to a more prominent location in the text. It can now be found in the model and not the experiment section. The revised statement reads: "In all experiments, we use prescribed 6-hourly climatological sea ice and SST fields interpolated from the monthly climatological values of the CMIP6 Forcing Datasets (input4MIPs, 1978-2020) as boundary conditions [DT18]. Prescribed climatological SSTs reduce the impact of interannual variability, such as the influence of ENSO events on precipitation. In addition, they separate the effect of model biases in the SST representation and atmospheric processes."

Section 2.3  $A_p$  was defined by Hwang and Frierson (2013) and  $E_p$  was defined by Adam et al. (2016). Please reference the indices accordingly.

For the sake of readability, we left this sentence as is. It is already clear that these indices come collectively from these two papers.

Figure 1  $kg^{-1}$  to  $kg^{-1}$  (also in all of the other figures) We corrected the typo in the respective figures.

Figure  $1 \dots$  near-surface specific humidity, calculated with respect to values derived from ERA5 reanalysis, is  $\dots$

We added the commas correspondingly.

Please refer to panel letters in the caption We now refer to the panels in all captions.

203 ?resp.? "respectively" is now spelled out.

Figure 2 and elsewhere, it would be better to describe units in square brackets, rather than following a divider, e.g., height / km — ; Height [km]

As the "[]" notation is not required by the ACP journals and the "/" notation is the more precise mathematical representation, we would like to refrain from changing the handling of the units. ACP journals require exponential writing of units, i.e.  $W\ m^{-2}$ , we therefore see no danger of confusion caused by multiple "/"s.

214 citation error We added the missing citation.

216,220 and elsewhere, citet to citep We made the corresponding changes.

**References**

[DABBW22] Yue Dong, Kyle C. Armour, David S. Battisti, and Edward Blanchard-Wrigglesworth. Two-Way Teleconnections between the Southern Ocean and the Tropical Pacific via a Dynamic Feedback. Journal of Climate, 35(19):6267–6282, October 2022.

[DT18] Paul J. Durack and Karl E. Taylor. PCMDI AMIP SST and sea-ice boundary conditions version 1.1.4, 2018.

- [FDW+23] Xiaofang Feng, Qinghua Ding, Liguang Wu, Charles Jones, Huijun Wang, Mitchell Bushuk, and Dániel Topál. Comprehensive Representation of Tropical–Extratropical Teleconnections Obstructed by Tropical Pacific Convection Biases in CMIP6. Journal of Climate, 36(20):7041–7059, October 2023.
- [HF13] Yen-Ting Hwang and Dargan M. W. Frierson. Link between the double-Intertropical Convergence Zone problem and cloud biases over the Southern Ocean. Proceedings of the National Academy of Sciences, 110(13):4935–4940, March 2013.
- [HMS17] Stephanie A. Henderson, Eric D. Maloney, and Seok-Woo Son. Madden–Julian Oscillation Pacific Teleconnections: The Impact of the Basic State and MJO Representation in General Circulation Models. Journal of Climate, 30(12):4567–4587, June 2017.
- [KHFZ08] Sarah M. Kang, Isaac M. Held, Dargan M. W. Frierson, and Ming Zhao. The Response of the ITCZ to Extratropical Thermal Forcing: Idealized Slab-Ocean Experiments with a GCM. Journal of Climate, 21(14):3521–3532, July 2008.
- [KHX+19] Sarah M. Kang, Matt Hawcroft, Baoqiang Xiang, Yen-Ting Hwang, Gabriel Cazes, Francis Codron, Traute Crueger, Clara Deser, Øivind Hodnebrog, Hanjun Kim, Jiyeong Kim, Yu Kosaka, Teresa Losada, Carlos R. Mechoso, Gunnar Myhre, Øyvind Seland, Bjorn Stevens, Masahiro Watanabe, and Sungduk Yu. Extratropical—Tropical Interaction Model Intercomparison Project (Etin-Mip): Protocol and Initial Results. Bulletin of the American Meteorological Society, 100(12):2589–2606, December 2019.
- [Lin07] Jia-Lin Lin. The Double-ITCZ Problem in IPCC AR4 Coupled GCMs: Ocean—Atmosphere Feedback Analysis. Journal of Climate, 20(18):4497–4525, September 2007.
- [MCCLT+22] E. Moreno-Chamarro, L.-P. Caron, S. Loosveldt Tomas, J. Vegas-Regidor, O. Gut-jahr, M.-P. Moine, D. Putrasahan, C. D. Roberts, M. J. Roberts, R. Senan, L. Terray, E. Tourigny, and P. L. Vidale. Impact of increased resolution on long-standing biases in HighResMIP-PRIMAVERA climate models. Geoscientific Model Development, 15(1):269–289, 2022.
- [MFK+25] Wolfgang A. Müller, Barbara Früh, Peter Korn, Roland Potthast, Johanna Baehr, Jean-Marie Bettems, Gergely Bölöni, Susanne Brienen, Kristina Fröhlich, Jürgen Helmert, Johann Jungclaus, Martin Köhler, Stephan Lorenz, Andrea Schneidereit, Reiner Schnur, Jan-Peter Schulz, Linda Schlemmer, Christine Sgoff, Trang V. Pham, Holger Pohlmann, Bernhard Vogel, Heike Vogel, Roland Wirth, Sönke Zaehle, Günther Zängl, Björn Stevens, and Jochem Marotzke. ICON: Towards vertically integrated model configurations for numerical weather prediction, climate predictions and projections. Bulletin of the American Meteorological Society, April 2025.
- [MRB+95] C.R. Mechoso, A.W. Robertson, N. Barth, M.K. Davey, P. Delecluse, P.R. Gent, S. Ineson, B. Kirtman, M. Latif, H. Le Treut, T. Nagai, J.D. Neelin, S.G.H. Philander, J. Polcher, P.S. Schopf, T. Stockdale, M.J. Suarez, L. Terray, O. Thual, and J.J. Tribbia. The Seasonal Cycle over the Tropical Pacific in Coupled Ocean—Atmosphere General Circulation Models. Monthly Weather Review, 123(9):2825–2838, September 1995.
- [MSW+20] Brecht Martens, Dominik L. Schumacher, Hendrik Wouters, Joaquín Muñoz-Sabater, Niko E. C. Verhoest, and Diego G. Miralles. Evaluating the land-surface energy partitioning in ERA5. Geoscientific Model Development, 13(9):4159–4181, September 2020.
- [MZZW23] Xinyu Ma, Shuyun Zhao, Hua Zhang, and Wuke Wang. The double -ITCZ problem in CMIP6 and the influences of deep convection and model resolution. International Journal of Climatology, 43(5):2369–2390, April 2023.
- [NCfARS22] (Eds) National Center for Atmospheric Research Staff. The Climate Data Guide: OAFlux: Objectively Analyzed air-sea Fluxes for the global oceans, September 2022.

- [PO83] José P. Peixóto and Abraham H. Oort. The Atmospheric Branch Of The Hydrological Cycle And Climate. In Alayne Street-Perrott, Max Beran, and Robert Ratcliffe, editors, Variations in the Global Water Budget, pages 5–65. Springer Netherlands, Dordrecht, 1983.
- [PO92] J. P. Peixoto and A. H. Oort. Physics of Climate Peixoto, J. P., & Oort, A. H. (1992). Physics of Climate. American Institute of Physics, 1992.
- [SBF+25] H. Segura, C. Bayley, R. Fievét, H. Glöckner, M. Günther, L. Kluft, A. K. Naumann, S. Ortega, D. S. Praturi, M. Rixen, H. Schmidt, M. Winkler, C. Hohenegger, and B. Stevens. A Single Tropical Rainbelt in Global Storm-Resolving Models: The Role of Surface Heat Fluxes Over the Warm Pool. Journal of Advances in Modeling Earth Systems, 17(7):e2024MS004897, July 2025.
- [SDFT13] David P. Schneider, Clara Deser, John Fasullo, and Kevin E. Trenberth. Climate Data Guide Spurs Discovery and Understanding. Eos, Transactions American Geophysical Union, 94(13):121–122, March 2013.
- [SND+21] Xiangzhou Song, Chunlin Ning, Yongliang Duan, Huiwu Wang, Chao Li, Yang Yang, Jianjun Liu, and Weidong Yu. Observed Extreme Air—Sea Heat Flux Variations during Three Tropical Cyclones in the Tropical Southeastern Indian Ocean. Journal of Climate, 34(9):3683–3705, May 2021.
- [TIK04] H. L. Tanaka, Noriko Ishizaki, and Akio Kitoh. Trend and interannual variability of Walker, monsoon and Hadley circulations defined by velocity potential in the upper troposphere. Tellus A: Dynamic Meteorology and Oceanography, 56(3):250, January 2004.

**Response to Review 2**

August 8, 2025

**Review of "Parameterization adaption needed to unlock the benefits of increased resolution for the ITCZ in ICON"**

Manuscript authors: Kroll et al

**Summary**

The authors use simulations at different resolutions and with the gustiness factor perturbed in the ICON model to examine mean precip biases. They argue that the double ITCZ bias persists at all resolutions but that increasing the minimum wind speed for the evaporation improves this, which however does not fix the larger underlying humidity biases across the tropics and subtropics. There are some results of interest here, but I had a hard time following this manuscript and left it not especially convinced. In part this stems from issues with the writing, in part from the experimental design, and in part from the lack of uncertainty quantification. Addressing all this adequately would take a serious amount of work, and as such I recommend rejection for eventual resubmission by the authors.

We thank the second reviewer for all their detailed and constructive comments. In response to the reviewer's main comments, we (1) include new high-resolution simulations with the full suite of parameterizations and discuss (in part of Section 3.3) how they compare to the setup with reduced parameterizations, (2) display statistical significance of the results, and (3) substantially revise and streamline the text. In subsequent paragraphs, we respond to the reviewer's detailed comments. The reviewer's comments are listed in black, and our response is listed in blue.

**Major comments**

**Experimental design**

It would be much cleaner if the highest resolution run didn't also have the convection and gravity wave schemes disabled as well, c.f. L120-122. As it stands, going from the 40km model to the 5km model, you're both increasing resolution and changing the model formulation. And even a 5km grid is surely not fine enough to resolve the large number of convective updrafts that are smaller than 5x5km. A relevant paper here is Clark et al. (2024), and references therein. In the GFDL AM4 model in an aquaplanet context, at high resolution the model behavior still changes dramatically depending on whether the convective parameterization is enabled or not. This is especially concerning given that some fields such as the humidity and temperature biases are nonmonotonic in resolution, with the change occurring going from the 40km version in which the deep convective parameterization is activated to the 5km version in which they are disabled.

We agree with the reviewer that an additional experiment in 5 km with a deep convective and gravity wave drag parametrization would benefit the study. We ran the corresponding experiment, the evaluation of which is now integrated in the manuscript. It replaces the original 5 km run without parameterization. We dedicate part of the results section 3.3 to a comparison of the 5 km setup with and without the deep convective and gravity wave parametrization. The introduction and discussion also includes a reference of Clark et al. 2024 [CLH24]: "In this context, it is important to note that

even at a horizontal resolution of 5 km, the necessity of deep convective parameterizations is still disputed, with some studies showing improvements in atmospheric representation with the elimination of parameterizations [VTBP+20] and others showing deterioration and insufficiently resolved processes [CLH24].".

Indeed, the atmospheric representation is improved with the parameterizations; however, the double - ITCZ persists in both setups. In Figure R 1, we exemplarily show the temperature fields for the 40 km setup, the 5 km simulation including all parameterizations and the 5 km simulations with reduced parameterizations (rParam):

Figure 1: Zonal mean temperature bias with respect to ERA5: all CTL for a 2 year average (a-c), all CTL for an average over two MAM seasons (d-f) and all PTB-5\_1 for an average over two MAM seasons (g-i) for  $40 \, \mathrm{km}$ ,  $5 \, \mathrm{km}$  and  $5 \, \mathrm{km}$ , rParam. Statistically significant differences between ERA and the model data are shown, insignificant regions are hatched. Statistical significance was tested with a two-sided ztest at  $\alpha = 0.1$  after autocorrelation correction.

**Uncertainty quantification**

C.f. L250-254, The 4, 5, and 6 experiments precip RMSE values are all within 0.03 of each other. Is that even a statistically significant separation? I worry about sampling uncertainty given the short durations of the runs. I have the same concern about other results; apart from Fig. A1 there is very little discussion of uncertainty quantification and its implications for interpreting the results. It seems plausible that you're over-interpreting differences across simulations that aren't statistically well separated.

We have expanded the uncertainty quantification and marked statistically significant differences in all precipitation, temperature, and specific humidity figures. With respect to the differences in the precipitation fields: The reference is CTL or ERA5 depending on the respective quantities shown. For our research questions, the statistically significant difference between CTL and  $U_{min} \in \{4,5,6\}$  is the important quantity, not the difference between  $U_{min} \in \{4,5,6\}$ . We can clearly show that the difference in precipitation between CTL and  $U_{min} \in \{4,5,6\}$  is statistically significant in the revised manuscript version. Also, there is a clear trend in the various tuning experiments. This shows that the change in the respective tuning parameter leads to systematic changes in the precipitation distribution and latent heat flux. In addition, we can explain the changes in model behavior based on physical arguments, again raising confidence in the results. We have chosen  $U_{min} = 5ms^{-1}$  as the basis for the 5 km experiments. After demonstrating that  $U_{min}$  can significantly alter the representation of the ITCZ compared to IMERG, we decided to invest our computation time in the requested simulation with parametrization rather than increasing the runtime of the  $U_{min} \in \{4,5,6\}$ .

Figure R 2 shows an example on how we are now restricting the analysis to statistically significant differences only.

Figure 2: Two-year mean large-scale precipitation bias with respect to IMERG 2004-2010 for different settings of the surface wind at 40 km resolution: PTB-0.5 (a), PTB-4 (b), PTB-5 (c), PTB-5\_1 (e), PTB-5\_1t (g), PTB-6 (d), PTB-6\_1 (f) and PTB-6\_1t (h). The corresponding global precipitation RMSE is stated beneath the panel for each sensitivity experiment. Statistically insignificant differences between IMERG and the experiments based on a two-sided z-test at  $\alpha=0.1$  are shown, and insignificant regions are grayed out. The global RMSE and mean error in near-surface specific humidity, calculated with respect to values derived from ERA5 reanalysis, is depicted in the inlays.

**Experiment names**

The results would be much easier to follow if the experiments had more descriptive names. So, instead of "R2B4," call it for example "160km." The "PTBX" simulation names are similarly unintuitive. We discarded of the ICON-specific language and only refer to the nominal resolution and experiment type, i.e. "CTL, 160 km" or "PTB-5, 160 km" to make the text easier to follow. PTB stands for "perturbed", and the numbers refer to the wind speed limiter values. As the value of the wind speed limiter is important for context, we decided to keep this part of the nomenclature.

**Unclear arguments regarding model tuning**

I find the discussion of tuning peppered throughout the introduction to be frustrating. It feels speculative and almost conspiratorial, seeming to imply that the modeling groups are somehow, in the case of focusing on global mean TOA radiative fluxes rather than regional circulation fidelity, doing something obviously wrong.

We want to ensure the reviewer that there is definitely no intention to accuse climate modelers of doing something wrong, and some of the authors of this manuscript are modelers themselves. We are fully aware that the TOA tuning is followed by process tuning, and we intended only to highlight the potential trade-offs between TOA tuning and process tuning, where improving one might degrade the other. We modified the corresponding sentence to make this clearer. In this context, we also added references to the studies from GFDL. It now reads: "Most model evaluation workflows focus first on global mean top-of-the-atmosphere (TOA) fluxes and potentially surface energy fluxes [MSR+12], and then proceed to tune additional atmospheric fields and processes, for example, as outlined in [HMG+17]. This is done with the knowledge that there can be trade-offs between these different optimizations, making dedicated approaches to optimize across multiple processes necessary, for example to focus on the precipitation distribution and TOA fluxes within one framework [ZGH+18a, ZGH+18b]." Adding a caveat that model tuning/process tuning is necessary should not disregard the work of modelers. In contrast, we wish to promote their work, which is unfortunately sometimes not appreciated enough. The following sentences underline that it would be very valuable for the community to allow for more publications documenting model improvements: "However, in general, most published model evaluation workflows provide little information on regional energy budgets that are mechanistically important for the large-scale circulation and precipitation distribution. This poses a problem: Sometimes improvements in global mean energy fluxes introduce compensating errors in regional energy fluxes and lead to deterioration of the large-scale circulation and precipitation distribution."

In the discussion of tuning high resolution models, L65-67, "misconception that all relevant processes are now resolved" is not justifiable, nor is it appropriate in tone. Model developers are fully aware of the physical scales of the various processes involved and how those compare to the scales resolved by their model.

We agree with the reviewer that the model developers are fully aware that not all relevant processes are resolved. However, the authors have seen cases where this is not fully acknowledged, for example in proposals for additional high-resolution modeling efforts. We softened the sentence, and it now reads: "High computational costs, in combination with the hope that the most relevant processes are now resolved, can then lead to a shortening of the model tuning process in high-resolution simulations."

The paper should incorporate Zhao et al. (2018a,b), who discuss how a tuning strategy targeting TOA fluxes was used to improve ITCZ simulation.

As stated above, the requested citations were integrated in the introduction to showcase a tuning strategy targeting the double-TCZ: "This is done with the knowledge that there can be trade-offs between these different optimizations, making dedicated approaches to optimize across multiple processes necessary, for example to focus on the precipitation distribution and TOA fluxes within one framework [ZGH+18a, ZGH+18b].".

**English writing**

My impression is that English is not the lead author's native language for writing. There are quite a lot of sentences where the grammar and/or word choice are difficult to follow. In aggregate, these make for a somewhat jarring reading. At least one of the coauthors is a native English speaker, and so I know it is within the authors' collective ability to, in the revision, significantly tighten up the English writing. Here is a nonexhaustive list of sentences that I struggled with:

- L36-37: The corresponding sentence was deleted during restructuring.
- L49: We reformulated to: "It is critical to recognize the coupling between the double-ITCZ problem and biases in the energy budget also in the context of model tuning, because the first step of model evaluation is in many cases the energy balance of the model [Wil20]."

- L111-114: We split this sentence in two. It now says: "ICON XPP uses parameterizations for radiation [HB18], cloud microphysics [Sei08], vertical diffusion [MSZ+07], convection [Tie89, BKJ+08], subgrid scale orographic drag [LM97] and non-orographic gravity wave drag [OBS+10]. The atmosphere is coupled to the land model JSBACH [RGG+21]."
- L132-134: The corresponding sentence was deleted during the restructuring of the manuscript.
- L134-135: This sentence was deleted.
- L138-141: The sentence was split into four for simplification: "The two  $U_{min}$  settings that perform the best in representing the large-scale annual-mean precipitation, PTB-5 and PTB-6, with  $U_{min} = 5 \text{ m s}^{-1}$  and 6 m s-1, are chosen for further optimization. First, the wind speed is adapted for land and ocean separately. In PTB-5\_1 and PTB-6\_1,  $U_{min}$  is set to = 1 m s-1 over land to account for the slower near-surface wind speeds. Second, in PTB-5\_1t, PTB-6\_1t, the model is retuned to reestablish a similar top-of-the-atmosphere (TOA) imbalance to CTL."
- L150-151: The sentence was split in two and rephrased to read: "For this purpose, a 1 horizontal resolution is chosen. This resolution is close to the tropical 1.4 horizontal resolution of the 160-km simulation."
- L235-236: The corresponding paragraph was reformulated it now says: "In this section, we investigate the efficacy of a parameter adjustment in addressing the double-ITCZ bias across resolutions in ICON. We test the  $U_{min}$  parameter adjustments (Sec. 2.1.2) to address the near-surface dry bias over the Warm Pool region with the goal of interrupting the circle of biases outlined in the previous paragraph. With an increase in  $U_{min}$ , we aim to increase evaporation and near-surface specific humidity. We expect that the increase in near-surface specific humidity will improve the Warm Pool precipitation and the double-ITCZ feature, through the mechanisms shown in Fig. 1."
- L237-238 "vicious cycle" over the top: The word "vicious" was removed.
- L238-244 (break up this sentence): We changed the sentence and the description of Umin. The paragraph now reads: "In this framework, evaporation E is tied to wind speed U at the lowest atmospheric level via the bulk flux formula for evaporation

$$E = \rho_a C_E U(q_s - q_a), \tag{1}$$

where  $\rho_a$  is the atmospheric density,  $q_s$  surface specific humidity,  $q_a$  near-surface atmospheric specific humidity, and the bulk transfer coefficient for latent heat  $C_E$ , which is inversely proportional to the Richardson number. The bulk flux formulation shows that the latent heat flux is directly proportional to the near-surface wind speed. The fix suggested by segurasingle2025doesnotchangeUdirectly, butadam coded lower limit for the near-surface wind speeds. In ICONXPP's turbulence parameterization, the surface wind University of the property ofwith  $U = MAX(U_{min}, U)$ . This lower limiter is used to account for the influence of subgridscale turbulence with the goal of increasing the turbulent fluxes in low-wind regimes. Increasing the default value of  $U_{min}$  from 1 m s-1 decreases the Richardson number in low-wind regimes, e.g., in the Warm Pool, and leads to increased evaporation. The influence of changes in  $U_{min}$ at various resolutions, including in simulations with the full set of parameterizations, has not been tested before. It is important to note that changes in  $U_{min}$  can similarly impact turbulent fluxes of sensible heat and momentum in addition to the targeted latent heat flux. Potential consequences of the resulting changes in the momentum budget were not considered in  $segura_single_2025. Concretely, this means that the suggested increase in U_{min}$  will increase the drag on near-surface winds. It is our aim to investigate whether there are associated negative influences on the circulation next to the positive effect for precipitation over the Warm Pool."

Separate from the English usage, there were far too many typos. Please carefully proofread the revision carefully as a final step before submitting.

We carefully proofread the revision before submitting it.

**Line by line comments**

**L13** "this could endanger the representation of the global circulation, energetic balance and teleconnections" confusing, due to the "could." Does it degrade these fields in your simulations or not?

Yes, the global circulation is affected as a consequence of the increased drag on the surface winds. This is visible in the slowdown of the trade winds (Fig. 7). As intended by the fix, the latent heat flux increases. However, the latent heat flux bias shifts and increases the relative contribution of inner-tropical moisture (Fig. 8), which changes the net energy input into the atmosphere. Both changes are degrading the corresponding fields and can not be counterbalanced as they are part of the solution to the precipitation bias. The slowdown of the trade winds endangers the teleconnections related to the wind-evaporation SST feedback. The top-of-the-atmosphere imbalance also changes considerably. However, this change can be corrected via re-tuning. We the sentence to "This degrades the representation of the global circulation, energy balance, and teleconnections." to be more definite.

**L15** what does "nondiscardable" mean?**

"Non-discardable" refers to parametrizations which are needed at the given resolution because the corresponding processes cannot be explicitly resolved. An example would be radiation or turbulence.

L25 "bias has been central to the precipitation bias discussions" this reads funny to me; consider rephrasing

The sentence was rephrased and now reads: "In addition to its influence on regional precipitation biases, the double ITCZ can influence large-scale climate phenomena such as the El Niño-Southern Oscillation [HK14, ZDCT14]. Therefore, identifying the cause of the double ITCZ is a key step towards improving climate models".

**L27-29** Correct and you should cite one or more papers that document these transient double ITCZ states, e.g. Magnusdottir and Wan 2008, https://journals.ametsoc.org/view/journals/atsc/65/7/2007jas2518.1.xml. And this preprint is particularly relevant:

 $https://essopenarchive.\ org/doi/full/10.22541/essoar.174017095.57302520$

We now cite both the proposed paper and preprint.

L42 Philander et al emphasize ocean atmosphere coupling and continental geometry, making it an odd choice to cite regarding this claim about net energy imbalance; Frierson et al. (2013) would be more appropriate.

Note that the "net energy input" includes the energy flux from the ocean to the atmosphere, and thus this statement is consistent with the work of Philander et al. We now additionally cite Frierson et al. (2013).

**L50-57** I don't find this discussion of the model tuning especially compelling, in large part because I'm struggling to follow it. Can you make your argument more precise and clear?

This comment was addressed in the section above titled "Unclear arguments regarding model tuning". Please refer to our previous statements and the revised manuscript.

L75-77 This sentence is meaningless to the reader, like me, who doesn't know what ICON XPP and ICON Sapphire are. I think you can omit this entirely, or if you want to keep it consider moving to the methods section or revising to provide more context

This sentence is important to include for readers who are familiar with ICON, and we believe what is relevant to take away from it is sufficiently clear even to readers not familiar with ICON. We kept this sentence as is.

L88 I would omit "the fuel for the hydrological cycle"; unneeded and too imprecise Omitted.

L94 "their implementation should receive more attention" this feels like too much of an editorializing statement to me in the context. I don't really know what 'minorlooking treatments' are beyond your summary having not read the Kawai et al paper, but based on your summary it's not obvious to me why they should indeed receive more attention.

The corresponding statement was deleted.

L111 I would omit the footnote; just include it in the parenthetical The footnote is removed.

L118 calling 5 km convection "resolving" is a stretch...very few convective updrafts span 5x5 square km. "convection permitting" is a widely used and I think more appropriate choice. We switched to convection permitting throughout.

**L165 Doesn't ERA5 directly output specific humidity?**

Yes, ERA5 does output the atmospheric specific humidity, and we show the corresponding value in the zonal bias plots. However, the near-surface humidity values at 10 m are not available in the archive.

**L214** fix the citations**

Done.

**L215** A lot of typos through the end of this paragraph; feels sloppy.

The two typos were corrected and formulations were adjusted for an easier reading flow.

**L229** What feedback loop?**

A stronger Walker circulation would lead to even more increased near-surface moisture in the Warm Pool, which constitutes a feedback. To avoid any confusion about this term, we remove the words "through a feedback loop."

L229-232 Is this proposed feedback your idea? If yes, it feels rather speculative. If not, it needs citations.

Yes, it is our idea - based on physical arguments. We test this feedback loop in Section 3.2. In order to give the reader some guidance, we outline the hypothesis we are testing at this point. In order to account for the reviewer's concern, we also expanded the analysis and give more details on the changes in circulation and moisture transport in the revised manuscript version. The additional analysis includes an evaluation of the Walker and Hadley circulations and a moisture budget analysis.

**Fig. 6 What does "normalized" probability density function mean?**

Normalized means that the area under probability density function integrates to the value of 1. This complies with the standard mathematical definition, we therefore did not add any further description to avoid suggesting otherwise.

**Fig. 7** Is OAFlux ultimately a better product than ERA5 for the surface LH fluxes? If so, then why show the biases of the simulations against both? Why not just use OAFlux in the context of this whole discussion? Perhaps I missed something here.

OAFlux does not provide all the quantities needed for the other bias plots (that is, all the temperature and humidity plots of the atmosphere). To remain in a consistent framework that is also closed energetically, we show the difference to ERA5 and only add OAFlux as complementing information.

L342-344 I don't understand this. Why do the signs of the respective biases lead to this inference about "symptoms" vs. "root cause"?

The corresponding paragraph was deleted during the restructuring process.

**Fig. 8 symbols for control run are too faint**

We adjusted the chosen color map. The corresponding figure now appears as this Figure R 3.

Figure 3: Tuning scores for all  $40 \,\mathrm{km}$  sensitivity experiments as a function of prescribed minimum surface wind speed. The values for a  $U_{min}$  of 1 are the CTL settings and where taken from the respective two years of the CTL simulation. Values for the asymmetric index  $A_p$ , symmetry index  $E_P$  as well as latent heat flux, near-surface specific humidity  $q_s$ , net top-of-the-atmosphere (TOA) imbalance, upwards radiative shortwave flux, outgoing long wave radiation (OLR) and global mean surface temperature  $T_S$  are shown. Each point depicts one global year average. For reference the corresponding values from IMERG, GPCP, ERA5 and CERES are depicted.

**References**

Clark, J. P., P. Lin, and S. A. Hill (2024), ITCZ Response to Disabling Parameterized Convection in Global FixedSST GFDLAM4 Aquaplanet Simulations at 50 and 6 km Resolutions, Journal of Ad vances in Modeling Earth Systems, 16(6), e2023MS003,968, doi:10.1029/2023MS003968.

Frierson, D. M. W., Y.T. Hwang, N. S. Fučkar, R. Seager, S. M. Kang, A. Donohoe, E. A. Maroon, X. Liu, and D. S. Battisti (2013), Contribution of ocean overturning circulation to tropical rainfall peak in the Northern Hemisphere, Nature Geoscience, 6(11), 940–944, doi:10.1038/ngeo1987.

Zhao, M., J.C. Golaz, I. M. Held, H. Guo, V. Balaji, R. Benson, J.H. Chen, X. Chen, L. J. Donner, J. P. Dunne, K. Dunne, J. Durachta, S.M. Fan, S. M. Freidenreich, S. T. Garner, P. Ginoux, L. M. Harris, L. W. Horowitz, J. P. Krasting, A. R. Langenhorst, Z. Liang, P. Lin, S.J. Lin, S. L. Malyshev, E. Mason, P. C. D. Milly, Y. Ming, V. Naik, F. Paulot, D. Paynter, P. Phillipps, A. Radhakrishnan, V. Ramaswamy, T. Robinson, D. Schwarzkopf, C. J. Seman, E. Shevliakova, Z. Shen, H. Shin, L. G. Silvers, J. R. Wilson, M. Winton, A. T. Wittenberg, B. Wyman, and B. Xiang (2018a), The GFDL Global Atmosphere and Land Model AM4.0/LM4.0: 1. Simulation Characteristics With Prescribed SSTs, Journal of Advances in Modeling Earth Systems, 10(3), 691–734, doi:10.1002/2017MS001208.

Zhao, M., J.C. Golaz, I. M. Held, H. Guo, V. Balaji, R. Benson, J.H. Chen, X. Chen, L. J. Donner, J. P. Dunne, K. Dunne, J. Durachta, S.M. Fan, S. M. Freidenreich, S. T. Garner, P. Ginoux, L. M. Harris, L. W. Horowitz, J. P. Krasting, A. R. Langenhorst, Z. Liang, P. Lin, S.J. Lin, S. L. Malyshev, E. Mason, P. C. D. Milly, Y. Ming, V. Naik, F. Paulot, D. Paynter, P. Phillipps, A. Radhakrishnan, V. Ramaswamy, T. Robinson, D. Schwarzkopf, C. J. Seman, E. Shevliakova, Z. Shen, H. Shin, L. G. Silvers, J. R. Wilson, M. Winton, A. T. Wittenberg, B. Wyman, and B. Xiang (2018b), The GFDL Global Atmosphere and Land Model AM4.0/LM4.0: 2. Model Description, Sensitivity Studies, and Tuning Strategies, Journal of Advances in Modeling Earth Systems, 10(3), 735–769, doi:10.1002/2017MS001209.

**References**

- [BKJ+08] Peter Bechtold, Martin Köhler, Thomas Jung, Francisco Doblas-Reyes, Martin Leutbecher, Mark J. Rodwell, Frederic Vitart, and Gianpaolo Balsamo. Advances in simulating atmospheric variability with the ECMWF model: From synoptic to decadal time-scales.

  Quarterly Journal of the Royal Meteorological Society, 134(634):1337–1351, 2008. Leprint: https://rmets.onlinelibrary.wiley.com/doi/pdf/10.1002/qj.289.
- [CLH24] Joseph P. Clark, Pu Lin, and Spencer A. Hill. ITCZ Response to Disabling Parameterized Convection in Global Fixed-SST GFDL-AM4 Aquaplanet Simulations at 50 and 6 km Resolutions. Journal of Advances in Modeling Earth Systems, 16(6):e2023MS003968, June 2024.
- [HB18] Robin J. Hogan and Alessio Bozzo. A Flexible and Efficient Radiation Scheme for the ECMWF Model. Journal of Advances in Modeling Earth Systems, 10(8):1990–2008, 2018. -eprint: https://agupubs.onlinelibrary.wiley.com/doi/pdf/10.1029/2018MS001364.
- [HK14] Yoo-Geun Ham and Jong-Seong Kug. Effects of Pacific Intertropical Convergence Zone precipitation bias on ENSO phase transition. Environmental Research Letters, 9(6):064008, May 2014.
- [HMG+17] Frédéric Hourdin, Thorsten Mauritsen, Andrew Gettelman, Jean-Christophe Golaz, Venkatramani Balaji, Qingyun Duan, Doris Folini, Duoying Ji, Daniel Klocke, Yun Qian, Florian Rauser, Catherine Rio, Lorenzo Tomassini, Masahiro Watanabe, and Daniel Williamson. The Art and Science of Climate Model Tuning. Bulletin of the American Meteorological Society, 98(3):589–602, March 2017.
- [LM97] François Lott and Martin J. Miller. A new subgrid-scale orographic drag parametrization: Its formulation and testing. Quarterly Journal Meteorological Society, Royal 123(537):101–127, 1997. \_eprint: https://rmets.onlinelibrary.wiley.com/doi/pdf/10.1002/qj.49712353704.
- [MSR+12] Thorsten Mauritsen, Bjorn Stevens, Erich Roeckner, Traute Crueger, Monika Esch, Marco Giorgetta, Helmuth Haak, Johann Jungclaus, Daniel Klocke, Daniela Matei, Uwe Mikolajewicz, Dirk Notz, Robert Pincus, Hauke Schmidt, and Lorenzo Tomassini. Tuning the climate of a global model. Journal of Advances in Modeling Earth Systems, 4(3):2012MS000154, March 2012.
- [MSZ+07] Thorsten Mauritsen, Gunilla Svensson, Sergej S. Zilitinkevich, Igor Esau, Leif Enger, and Branko Grisogono. A Total Turbulent Energy Closure Model for Neutrally and Stably Stratified Atmospheric Boundary Layers. Journal of the Atmospheric Sciences, 64(11):4113–4126, November 2007.
- [OBS+10] Andrew Orr, Peter Bechtold, John Scinocca, Manfred Ern, and Marta Janiskova. Improved Middle Atmosphere Climate and Forecasts in the ECMWF Model through a Nonorographic Gravity Wave Drag Parameterization. Journal of Climate, 23(22):5905 5926, 2010. Place: Boston MA, USA Publisher: American Meteorological Society.

- [RGG+21] Christian H. Reick, Veronika Gayler, Daniel Goll, Stefan Hagemann, Marvin Heidkamp, Julia E. M. S. Nabel, Thomas Raddatz, Erich Roeckner, Reiner Schnur, and Stiig Wilkenskjeld. JSBACH 3 The land component of the MPI Earth System Model: documentation of version 3.2. page 4990986, February 2021. Artwork Size: 4990986 Medium: application/pdf Publisher: MPI für Meteorologie Version Number: 1.
- [Sei08] Axel Seifert. A revised cloud microphysical parameterization for COSMO-LME., 2008.
- [Tie89] M. Tiedtke. A Comprehensive Mass Flux Scheme for Cumulus Parameterization in Large-Scale Models. Monthly Weather Review, 117(8):1779 1800, 1989. Place: Boston MA, USA Publisher: American Meteorological Society.
- [VTBP+20] Jesús Vergara-Temprado, Nikolina Ban, Davide Panosetti, Linda Schlemmer, and Christoph Schär. Climate Models Permit Convection at Much Coarser Resolutions Than Previously Considered. Journal of Climate, 33(5):1915 – 1933, 2020.
- [Wil20] Martin Wild. The global energy balance as represented in CMIP6 climate models. Climate Dynamics, 55(3-4):553–577, August 2020.
- [ZDCT14] Honghai Zhang, Clara Deser, Amy Clement, and Robert Tomas. Equatorial signatures of the Pacific Meridional Modes: Dependence on mean climate state. Geophysical Research Letters, 41(2):568–574, January 2014.
- [ZGH+18a] M. Zhao, J.-C. Golaz, I. M. Held, H. Guo, V. Balaji, R. Benson, J.-H. Chen, X. Chen, L. J. Donner, J. P. Dunne, K. Dunne, J. Durachta, S.-M. Fan, S. M. Freidenreich, S. T. Garner, P. Ginoux, L. M. Harris, L. W. Horowitz, J. P. Krasting, A. R. Langenhorst, Z. Liang, P. Lin, S.-J. Lin, S. L. Malyshev, E. Mason, P. C. D. Milly, Y. Ming, V. Naik, F. Paulot, D. Paynter, P. Phillipps, A. Radhakrishnan, V. Ramaswamy, T. Robinson, D. Schwarzkopf, C. J. Seman, E. Shevliakova, Z. Shen, H. Shin, L. G. Silvers, J. R. Wilson, M. Winton, A. T. Wittenberg, B. Wyman, and B. Xiang. The GFDL Global Atmosphere and Land Model AM4.0/LM4.0: 2. Model Description, Sensitivity Studies, and Tuning Strategies. Journal of Advances in Modeling Earth Systems, 10(3):735-769, March 2018.
- [ZGH+18b] M. Zhao, J.-C. Golaz, I. M. Held, H. Guo, V. Balaji, R. Benson, J.-H. Chen, X. Chen, L. J. Donner, J. P. Dunne, K. Dunne, J. Durachta, S.-M. Fan, S. M. Freidenreich, S. T. Garner, P. Ginoux, L. M. Harris, L. W. Horowitz, J. P. Krasting, A. R. Langenhorst, Z. Liang, P. Lin, S.-J. Lin, S. L. Malyshev, E. Mason, P. C. D. Milly, Y. Ming, V. Naik, F. Paulot, D. Paynter, P. Phillipps, A. Radhakrishnan, V. Ramaswamy, T. Robinson, D. Schwarzkopf, C. J. Seman, E. Shevliakova, Z. Shen, H. Shin, L. G. Silvers, J. R. Wilson, M. Winton, A. T. Wittenberg, B. Wyman, and B. Xiang. The GFDL Global Atmosphere and Land Model AM4.0/LM4.0: 1. Simulation Characteristics With Prescribed SSTs. Journal of Advances in Modeling Earth Systems, 10(3):691–734, March 2018.

---

## Referee Report (RR1)

**Review of**

**Parameterization adaption needed to unlock the benefits of increased resolution for the ITCZ in ICON**

**Kroll et al.**

**General**

This is my second review of this work, where the authors examine the double intertropical convergence zone (IITCZ) bias in ICON XPP across model resolutions. A turbulent threshold wind parameter is used as a tuning parameter. Increasing the parameter enhances evaporation, which compensates for the weak bias in moisture transport into the deep tropics, but also influences drag and convection invigoration. The paper is much improved and the authors have adequately addressed my comments. I therefore recommend accepting the paper following some minor editorial comments.

**Comments by line number**

- 9 (abstract) "low bias" is ambiguous (it can interpreted as a weak bias). I suggest "dry bias"
- 15 (abstract) non-discardable
- 31 why "return" flow? "poleward flow" seems more appropriate
- 38 I would add Adam et al. 2018 who explicitly show the energetic contributions to the bias from various factors
- 85 ICON or ICON XPP? Are the results generalizable to other ICON versions?
- 101 what do you mean by "real world atmospheric representation"? Do you mean observational data?

Figure 1 walker -> Walker

145  $U \rightarrow U$

188 Here again "high biased" is ambiguous. Do you mean highly biased or too high?

Eq. 3 No need to specify  $0^{\circ}N$  and  $0^{\circ}S$ , you can simply use  $0^{\circ}$

243  $20^{\circ}N$  and  $20^{\circ}S$

251-252 This explanation is speculative and not convincing. If this were true, we would see a contrast between rising and descending regions (in the lower troposphere), which I don't see. Another possibility is that energy input into the lower atmosphere is similar in all cases, and therefore it translates to higher temperatures for the drier atmosphere. Showing zonal mean moist static energy biases would help in that regard.

291 remove comma

Figure 10 caption Inconsistent slanting of variables

I think this is a critical point worth elaborating on. Specifically, beyond the surface heat fluxes examined in this work, changes in surface wind stress critically affect ocean dynamics and vertical structure (e.g., the depth of the mixed layer) which would then influence the climate system through multiple mechanisms.

---

## Author Response (AR2)

**Response to Review 1**

October 17, 2025

**Review of "Parameterization adaptation needed to unlock the benefits of increased resolution for the ITCZ in ICON"**

Manuscript authors: Kroll et al

**General**

This is my second review of this work, where the authors examine the double intertropical convergence zone (IITCZ) bias in ICON XPP across model resolutions. A turbulent threshold wind parameter is used as a tuning parameter. Increasing the parameter enhances evaporation, which compensates for the weak bias in moisture transport into the deep tropics, but also influences drag and convection invigoration. The paper is much improved and the authors have adequately addressed my comments. I therefore recommend accepting the paper following some minor editorial comments.

We would like to thank Reviewer 1 very much for taking the time to review our manuscript "Parameterization adaption needed to unlock the benefits of increased resolution for the ITCZ in ICON" twice. Their thoughtful and detailed comments were very helpful in improving the first version and enhancing the clarity and accuracy of the manuscript. Below we list the reviewer's comments and respond to them individually. The reviewer's comments are shown in black; our response is written in blue ink.

**Comments by line number**

9 (abstract) "low bias" is ambiguous (it can interpreted as a weak bias). I suggest "dry bias" We adopted the improved formulation.

15 (abstract) non-discardable We corrected the typo.

31 why "return" flow? "poleward flow" seems more appropriate
We modified - it now reads "poleward flow" instead of "poleward return flow".

38 I would add Adam et al. 2018 who explicitly show the energetic contributions to the bias from various factors

Adam et al. 2018 are now included in the discussion: "This underlines that the double-ITCZ problem cannot be investigated as an isolated tropical phenomenon: Sub- and extratropical biases in the energy budget can also be sources of the problem (Kang et al., 2008; Hwang and Frierson, 2013; Adam et al., 2018; Kang et al., 2019), and tropical biases can likewise cause biases in the sub- and extratropics (Henderson et al., 2017; Dong et al., 2022; Feng et al., 2023)."

85 ICON or ICON XPP? Are the results generalizable to other ICON versions?

The results have been tested in different ICON configurations, but as we only investigate ICON-XPP we changed the formulation to: "What are the underlying mechanisms leading to the double-ITCZ and associated large-scale climate biases in ICON XPP and how do the chosen adjustments ameliorate it?".

101 what do you mean by "real world atmospheric representation"? Do you mean observational data? Yes, observational data/observations are meant. We rephrased to: "For this purpose, we compare the reference observations, the ICON simulation with a double ITCZ, and the ICON simulation with tuning adjustments."

Figure 1 walker — Walker We corrected the notation.

145 U — *U*

We corrected the notation.

188 Here again "high biased" is ambiguous. Do you mean highly biased or too high? We clarified that "too high" was meant. The corresponding sentence now reads: "The ERA5 latent heat flux values are thought to have a positive bias (Martens et al., 2020; Song et al., 2021)."

Eq. 3 No need to specify  $0^{\circ}$  N and  $0^{\circ}$  S, you can simply use  $0^{\circ}$  We removed the "S" and "N" for  $0^{\circ}$ .

 $243~20^{\circ}$  N and  $20^{\circ}$  S

We changed the notation accordingly.

251-252 This explanation is speculative and not convincing. If this were true, we would see a contrast between rising and descending regions (in the lower troposphere), which I don't see. Another possibility is that energy input into the lower atmosphere is similar in all cases, and therefore it translates to higher temperatures for the drier atmosphere. Showing zonal mean moist static energy biases would help in that regard.

We now follow the reviewer's formulation and also state the explanation as a hypothesis: "The near-surface warm bias could result from the dry bias in the near-surface layer. If the energy input into the lower atmosphere is similar in the CTL and PTB simulations, the same amount of energy would produce a warmer atmosphere under drier conditions.". For the sake of brevity, we chose not to include an additional figure.

**291 remove comma**

We removed the comma. The sentence now reads: "In a next step, we address concerns that increased  $U_{min}$  might adversely affect the velocity distribution by increasing the surface drag coefficient on the near-surface velocities and correspondingly slowing down the circulation.".

Figure 10 caption Inconsistent slanting of variables We corrected the inconsistent slanting of variables.

559-561 I think this is a critical point worth elaborating on. Specifically, beyond the surface heat fluxes examined in this work, changes in surface wind stress critically affect ocean dynamics and vertical structure (e.g., the depth of the mixed layer) which would then influence the climate system through multiple mechanisms.

We agree that the Umin fix would most likely have even more pronounced negative effects in a coupled setup. Following the reviewer's suggestion we added the following formulation: "In a coupled climate simulation, changes in  $U_{min}$  could also critically influence ocean dynamics and vertical structure (e.g., lead to substantial cooling), potentially introducing severe large-scale biases."

**Response to Review 2**

October 17, 2025

**Review of "Parameterization adaption needed to unlock the benefits of increased resolution for the ITCZ in ICON"**

Manuscript authors: Kroll et al

We would like to thank Reviewer 2 very much for reviewing our manuscript "Parameterization adaption needed to unlock the benefits of increased resolution for the ITCZ in ICON". Below we list the reviewer's comments and respond to them individually. The reviewer's comments are shown in black; our response is written in blue ink.

This study investigated the double-ITCZ bias in the AMIP simulations of ICON at different horizontal resolutions and explored the impacts of the minimum gustiness parameter. They showed that the double-ITCZ bias still exists with 5km horizontal resolution. They found that while increasing Umin over ocean reduces the tropical precipitation bias to some extent, it has some adverse impacts on the large-scale circulation and evaporation and does not fix the biases in humidity and its source.

While the manuscript presents a lot of results which are worthwhile to be published, the manuscript is not clearly structured (partly because there are too many results) and the writing is sometimes too casual for publication standard. Also, I feel the manuscript lacks a clear focus. I would recommend a major revision so the authors can improve their presentation and sort out their focus.

All authors have reread the manuscript, with particular attention to shortening the text. In this process, we also removed phrasings that appeared too casual, while acknowledging that some choices of phrasing may reflect stylistic preferences. Regarding the focus of the manuscript, we worked on making the structure even clearer. A key change in this regard was to align the research questions directly with the results sections. Previously the research questions were answered over multiple section, which made it far harder for the reader to follow.

There are a lot of experiments with different values in the perturbing parameters. While I understand the workflow, I think it will be much clearer if the authors can focus on just one or two experiments. For example, they may focus on PTB- $5_1$  and remove the results of PTB-5. It makes sense to me to keep Umin =1m/s over land.

We agree that the paper contains a number of experiments and analyses. In the previous review round, this was explicitly requested (i.e., an additional 5 km simulation and analysis for Fig. 7 and 9). To accommodate these, we already discarded showing most retuning experiments after the wind speed limiter adjustments (PTB-5\_1t and PTB-6\_1t) in the last revision cycle. We agree that—should one decide to use the Umin approach—the PTB-5\_1 experiment would be superior to PTB-5. However, as [SBF+24] used a uniform Umin, we think it is important to point out that using separate values over land and ocean yields better results and is physically motivated. Therefore, we see a clear motivation to keep some of the discussion about PTB-5 versus PTB-5\_1. To reduce the amount of data presented, we now only show the PTB-5\_1 results where they are absolutely needed to make this statement (precipitation, wind fields, and circulation), while we omit PTB-5\_1 in the figures for latent heat flux and residence time.

**Line by line comments**

Figures: for unit in the colorbar, one may use "[]" instead of "/" which often means "divided by". Both "[]" and "/" are commonly used in the literature. As both notations are acceptable, we would

prefer to retain our current style.

Figure 3,4: Are these biases in humidity and temperature shared by CMIP models?

Yes, similar biases exist in CMIP models. We have now added a corresponding comment in the discussion: "Again, similar temperature and humidity biases can be found in the CMIP models ([TFK+13, TFT24]), suggesting that the described bias in moisture export from the subtropical boundary layer is not unique to the ICON model."

Figure 5: Since the different treatment for land and ocean, why not show the RMSE for land and ocean separately?

Following the reviewer's comment above, we are trying to reduce the complexity and amount of data shown, specifically with respect to PTB-5\_1 and PTB-5. We therefore concentrate on optimizing the global RMSE rather than two separate quantities.

Figure 5: better to move the box of RMSE and mean error out of the figure.

We tested moving the RMSE and mean error out of the figure (cf. Fig. 1), but decided to keep the former layout as it appears less busy while still showing the relevant tropical precipitation data.

For 2d spatial map, is it necessary to show the global map? Maybe focus on the tropical region since the paper is about tropical precipitation bias.

We would like to keep the global maps, as the Umin threshold is applied globally, and the discussed consequences also arise in the sub- and extra tropics, as seen, for example, in the newly introduced near-surface wind speed biases over extra tropical land. We however now removed more of the white space between the individual plots, which allowed saving space to display more details.

Figure 11: Is it necessary to rotate it? We removed the rotation.

Line 204: in the origin of

We suspect that Line 304 was meant and corrected accordingly: "Section 3.2.3 explores how  $U_{min}$  leads to an unintended shift in the origin of the inner tropical moisture."

Line 509: Here, it is more related to the double-ITCZ bias in AMIP (instead of CMIP), which exists without the bias in SST.

We now make a clear distinction between our AMIP-type setup, possible reasons for the bias therein, and then compare it to the CMIP setups: "The precipitation biases in our ICON resolution hierarchy arise from both atmospheric contributions and evaporation biases linked to a mismatch between prescribed SSTs and atmospheric conditions. Although our model setup is AMIP-type, it also expresses a double-ITCZ bias as seen across several CMIP generations ([TFT24]).".

Line 512: It is better to avoid using two "focus on".

We reformulated the sentence to: "To address the biases, we concentrated on the parameterizations that remain active at the km-scale, focusing on  $U_{min}$  in the bulk flux formulation of the turbulence scheme."

Given the results presented here, what are the implications for the double-ITCZ bias in AMIP-style simulations (in the CMIP project).

We added our suggestion for ICON and AMIP-style simulations in the CMIP project in the Discussion: "The results of this study suggest that a physically motivated correction of the double-ITCZ bias in ICON (and potentially other AMIP-style CMIP simulations) should focus on reducing exaggerated vertical moisture fluxes in the subtropics."

Figure 1: Two-year mean large-scale precipitation bias with respect to IMERG averaged over 2004-2010 for different settings of the surface wind at 40 km resolution: PTB-0.5 (a), PTB-4 (b), PTB-5 (c), PTB-5\_1 (e), PTB-5\_1 (g), PTB-6 (d), PTB-6\_1 (f) and PTB-6\_1 (h). The corresponding global precipitation RMSE is stated beneath the panel for each sensitivity experiment. Statistically insignificant differences between IMERG and the experiments based on a two-sided z-test at  $\alpha=0.1$  are grayed out. The global RMSE and mean error in near-surface specific humidity, calculated with respect to values derived from ERA5 reanalysis, is depicted in the inlays.

**References**

- [SBF+24] Hans Segura, Clara Bayley, Romain Fi´evet, Helene Gloeckner, Moritz Guenther, Lukas Kluft, Ann Kristin Naumann, Sebastian Ortega, Divya Sri Praturi, Marius Rixen, Hauke Schmidt, Winkler Marius, Cathy Hohenegger, and Stevens Bjorn. A single tropical rainbelt in global storm-resolving models: the role of surface heat fluxes over the warm pool, 2024.
- [TFK+13] Baijun Tian, Eric J. Fetzer, Brian H. Kahn, Joao Teixeira, Evan Manning, and Thomas Hearty. Evaluating CMIP5 models using AIRS tropospheric air temperature and specific humidity climatology. Journal of Geophysical Research: Atmospheres, 118(1):114–134, January 2013.
- [TFT24] Baijun Tian, Eric J. Fetzer, and Joao Teixeira. Assessing the Tropospheric Temperature and Humidity Simulations in CMIP3/5/6 Models Using the AIRS Obs4MIPs V2.1 Data. Journal of Geophysical Research: Atmospheres, 129(15):e2023JD040536, August 2024.